TOPICAL REVIEW

# Heksor: the central nervous system substrate of an adaptive behaviour

Jonathan R. Wolpaw[1,2] and Adam Kamesar[3]

[1]*National Centre for Adaptive Neurotechnologies, Albany Stratton VA Medical Centre, Albany, NY, USA*
[2]*Department of Biomedical Sciences, State University of New York, Albany, NY, USA*
[3]*Hebrew Union College, Cincinnati, OH, USA*

Handling Editors: Ian Forsythe & Richard Carson

The peer review history is available in the Supporting Information section of this article (https://doi.org/10.1113/JP283291#support-information-section).

**Abstract**   Over the past half-century, the largely hardwired central nervous system (CNS) of 1970 has become the ubiquitously plastic CNS of today, in which change is the rule not the exception. This transformation complicates a central question in neuroscience: how are adaptive behaviours – behaviours that serve the needs of the individual – acquired and maintained through life? It poses a more basic question: how do many adaptive behaviours share the ubiquitously plastic CNS? This question compels neuroscience to adopt a new paradigm. The core of this paradigm is a CNS entity with unique properties, here given the name *heksor* from the Greek *hexis*. A heksor is a distributed network of neurons and synapses that changes itself as needed

**Jonathan R. Wolpaw** is Director of the NIH-funded National Center for Adaptive Neurotechnologies at the Stratton VA Medical Center in Albany, New York and a Professor of Biomedical Sciences at the State University of New York in Albany. He is a neurologist who has been engaged in basic and clinical neuroscience research for 50 years. He and his colleagues developed and are using operant conditioning of spinal reflexes to explore the plasticity underlying learning and to enhance functional recovery for people with spinal cord injuries and other disorders. They have also led development of electroencephalography-based brain–computer interfaces (BCIs) and are exploring BCI-based methods for enhancing neurorehabilitation. His research group has been funded for many years by NIH, other federal agencies and private foundations, and has received numerous national and international awards. **Adam Kamesar** is a classical philologist who was trained at a number of European universities. His primary research interests and publications have to do with the history of ancient Greek and Latin scholarship, especially in the areas of grammar and language, exegesis, literary criticism, and philosophy. In attempting to provide assistance to Dr Wolpaw in finding a new designation for the CNS substrate of an adaptive behaviour, he was able to rely on his experience with the ancient sources in these sectors. He has also published translations from Latin, Greek, Hebrew and Italian. He is the American correspondent for Adamantius (Bologna), and serves on the editorial boards of Henoch (Turin) and Vigiliae Christianae (Leiden).

to maintain the key features of an adaptive behaviour, the features that make the behaviour satisfactory. Through their concurrent changes, the numerous heksors that share the CNS negotiate the properties of the neurons and synapses that they all use. Heksors keep the CNS in a state of *negotiated equilibrium* that enables each heksor to maintain the key features of its behaviour. The new paradigm based on heksors and the negotiated equilibrium they create is supported by animal and human studies of interactions among new and old adaptive behaviours, explains otherwise inexplicable results, and underlies promising new approaches to restoring behaviours impaired by injury or disease. Furthermore, the paradigm offers new and potentially important answers to extant questions, such as the generation and function of spontaneous neuronal activity, the aetiology of muscle synergies, and the control of homeostatic plasticity.

(Received 10 May 2022; accepted after revision 20 June 2022; first published online 30 June 2022)

**Corresponding author** J. R. Wolpaw: National Centre for Adaptive Neurotechnologies, Albany Stratton VA Medical Centre, 113 Holland Ave., Albany, NY 12208, USA. Email: wolpaw@neurotechcenter.org

**Abstract figure legend** The first image on the left is a naïve CNS. The triangles are neurons or synapses; their properties can be modified by experience. Early in life, experience creates the substrate (the heksor) for flexion-withdrawal behaviour. Its heksor (green) comprises a network of neurons and synapses that changes itself as needed to maintain satisfactory flexion-withdrawal. Later on, experience creates the heksor for locomotion (red); it changes itself as needed to maintain satisfactory locomotion. Because the two heksors overlap, their concurrent changes are a negotiation. They negotiate the properties of their neurons and synapses so that each heksor maintains satisfactory performance of its behaviour. Still later, athletic training creates the heksor for throwing the discus (orange). This expands the negotiation. Each of the three heksors may affect the neurons or synapses in any one of them. Together, the heksors keep the CNS in a state of negotiated equilibrium that ensures satisfactory performance of all their behaviours.

## Introduction

In the past half-century, the CNS of 1970, which could change in only a few places and in only a few ways, has gradually become the ubiquitously plastic CNS of today, in which change is the rule rather than the exception (Bertrand & Cazalets, 2013; Chorghay et al., 2018; Crosson et al., 2017; De Zeeuw et al., 2021; Fields, 2015; Frisén, 2016; Hagenston et al., 2020; Martins-Pinge, 2011; Mendell, 1984; Pi et al., 2019; Pierrot-Deseilligny & Burke, 2012; Wolpaw, 2010; Wolpaw & Tennissen, 2001). This transformation complicates a central question in neuroscience: how are adaptive behaviours – behaviours that serve the needs of the individual – acquired and maintained through life? It poses a more basic question: how do many adaptive behaviours share a ubiquitously plastic CNS? This article explicates and begins to answer this critical question. The article's central thesis is that the question compels neuroscience to adopt a new paradigm for how adaptive behaviours are acquired and maintained.

The core of the new paradigm is a CNS entity with unique properties that enable it to produce and maintain an adaptive behaviour in the ubiquitously plastic CNS. The entity is here given the name *heksor*. The first section describes how the recognition that the CNS remains plastic through life changes the problem of adaptive behaviours and, in doing so, reveals the unique entity to which we are applying the name *heksor*. The second section describes the process that arrived at *heksor* as an appropriate name, defines and explicates the new term,

and states its fundamental differences from existing terms. The third reviews the evidence for the existence of heksors, describes their role in acquisition and maintenance of adaptive behaviours, and illustrates the explanatory power of the new concept. The final section describes strategies for testing the applicability of the heksor concept to a broad range of adaptive behaviours and for exploring its ability to answer important questions in neuroscience.

The central message of the presentation is that the progress of the past 50 years compels neuroscience to develop a new paradigm for understanding how adaptive behaviours are acquired and maintained. The old paradigm – with its assumptions and its terminology – was adequate for a largely hardwired CNS but is not adequate for what we now understand to be a ubiquitously plastic CNS. By introducing the concept of the *heksor* and linking it to the complementary concept of the *negotiated equilibrium* that heksors create, this article introduces, formalizes, and begins to explore a new conceptual paradigm that is driven by and encompasses the recent recognition that the CNS remains ubiquitously plastic through life.

### The problem of adaptive behaviours

Adaptive behaviours are behaviours that are useful, that fulfil the needs of the individual. They enable the individual to adapt to the environment, to function effectively in the environment. A flexion-withdrawal reflex

is an adaptive behaviour: it removes the fingertip from the hot stove. Locomotion is another: it moves the person across the room. Adaptive behaviours are normative: each has a set of attributes, or key features, by which it can be judged to be good or bad, satisfactory or unsatisfactory. A flexion-withdrawal reflex has one key feature: it is good if it removes the fingertip from the hot stove very quickly; it is bad if it removes it slowly (or not at all). Locomotion has multiple key features: it is good if it is accurate, stable, symmetrical, energy efficient, etc.; it is bad if it lacks these key features.

Most adaptive behaviours are acquired through experience (i.e. through interactions with the environment). This is true even for simple behaviours (e.g. Chen & Wolpaw, 1995; Evatt et al., 1989; Eyre, 2014; Eyre et al., 2001; Granmo et al., 2008; Hawkins et al., 2006; Martin et al., 2004, 2009; Myklebust et al., 1982, 1986; Schouenborg, 2008; Thompson & Wolpaw, 2014; Wolpaw et al., 1983). For example, flexion-withdrawal reflexes, previously considered quintessential hardwired behaviours, are now known to be acquired through interactions with the environment that occur *in utero* and early in postnatal life (Granmo et al., 2008; Schouenborg, 2008). Figure 1*A* illustrates the essential role of appropriate activity-dependent change (i.e. plasticity) in creating satisfactory flexion-withdrawal reflexes by showing the consequences of preventing this plasticity. Neonatal transection of the rat spinal cord stops the activity-dependent plasticity that begins *in utero* and creates effective flexion-withdrawal reflexes. As a result, when these animals become adults, their limbs often move toward rather than away from a noxious stimulus.

Adaptive behaviours result from widespread CNS plasticity. For example, mastery of a finger-flexion sequence behaviour depends on changes in cortex, subcortical structures such as basal ganglia and cerebellum, and spinal cord (Fig. 1*B*) (Vahdat et al., 2015). Neuronal and/or synaptic changes in each of these regions contribute to the behaviour. Similarly, operant conditioning of the spinal stretch reflex (the knee-jerk reflex), or its electrical analogue the H-reflex, depends on changes from cerebellum to cortex to spinal cord (Fig. 1*C*); the changes in the brain induce, and may be needed to maintain, the changes in the spinal cord that directly underlie the behaviour (for review, Thompson & Wolpaw, 2014; Wolpaw, 2018). In sum, an adaptive behaviour depends on plasticity in a network of neurons and synapses that can extend from cortex to spinal cord.

**The problem 50 years ago.** Fifty years ago, the goal of research in this area was to discover where and how the CNS changed to produce a new adaptive behaviour. This goal was based on the assumption that the CNS was mostly hardware, that it consisted mainly of neurons and synapses that did not change after the completion of pre- and post-natal development. This assumption ignored impressive evidence that supposedly hardwired neurons and synapses could change later in life (e.g. DiGiorgio, 1929; Shurrager & Dykman, 1951). Nevertheless, it was the prevailing view. As a result, researchers sought to discover the special sites responsible for acquiring new adaptive behaviours. In the mid-20th century, this focus was evident in the extensive efforts of many prominent research groups to find modifiable synapses in the spinal cord, in brain regions such as hippocampus and cerebellum, and in a variety of invertebrate preparations, and to connect their modifications to changes in behaviour (reviewed in Wolpaw & Carp, 2006).

**The problem now.** The progress of the past 50 years has shown that the CNS is largely plastic; it has little or no completely fixed hardware (Bertrand & Cazalets, 2013; Chorghay et al., 2018; Crosson et al., 2017; De Zeeuw et al., 2021; Fields, 2015; Frisén, 2016; Hagenston et al., 2020; Martins-Pinge, 2011; Mendell, 1984; Pi et al., 2019; Pierrot-Deseilligny & Burke, 2012; Wolpaw, 2010; Wolpaw & Tennissen, 2001). While plasticity is most prominent during development, it continues through later life as well. It has many mechanisms and operates on many time scales, from synaptic changes that occur in minutes, to changes in cortical representation that develop over weeks, to reflex changes that evolve over decades (e.g. Koceja et al., 1995; Nudo et al., 1996; Sun et al., 2021).

The recognition of ubiquitous CNS plasticity profoundly changes the problem of how adaptive behaviours are acquired and maintained, and for a very simple reason. The CNS is a multi-user system; many different adaptive behaviours share its neurons and synapses. An individual neuron or synapse in cortex, cerebellum, basal ganglia, or spinal cord is not dedicated to one behaviour; it may participate in flexion-withdrawal reflexes, locomotion, reach-and-grasp, pole-vaulting, playing the piano, and/or a host of other behaviours. As a result, when experience modifies neurons and synapses to create a new adaptive behaviour, it inevitably disturbs existing (i.e. old) adaptive behaviours that use these same neurons and synapses. If an old behaviour is to remain satisfactory, the widely distributed neurons and synapses that produce it need to change. Thus, the goal of research is no longer simply to discover where and how the CNS changes to produce the new adaptive behaviour. Now, the goal of research is to discover where and how the CNS changes to acquire the new adaptive behaviour *and* to maintain the old adaptive behaviours affected by acquisition of the new behaviour. Furthermore, the changes that maintain the old behaviours may sometimes be far more extensive than those that underlie the new behaviour (see below). This second part of the goal,

determining how old behaviours are maintained as new behaviours are acquired – that is, how behaviours share the CNS – is the focus of this paper.

The difficulty that a widely plastic system faces in acquiring a new behaviour and simultaneously maintaining old ones is illustrated by the issue of 'catastrophic interference' now prominent in the artificial neural network (ANN) literature (and originally described by Grossberg (1980) as the 'stability–plasticity dilemma').

For an ANN that has acquired one behaviour, acquiring a second behaviour may destroy its ability to produce the first one. The development of satisfactory solutions to this problem is the focus of much current ANN research (e.g. Velez & Clune, 2017; Zhang et al., 2021; for review see Parisi et al., 2019). For the CNS, the need to maintain existing behaviours despite new plasticity is ongoing; it is not limited simply to the acquisition of new adaptive behaviours. It arises also due to growth and ageing, trauma

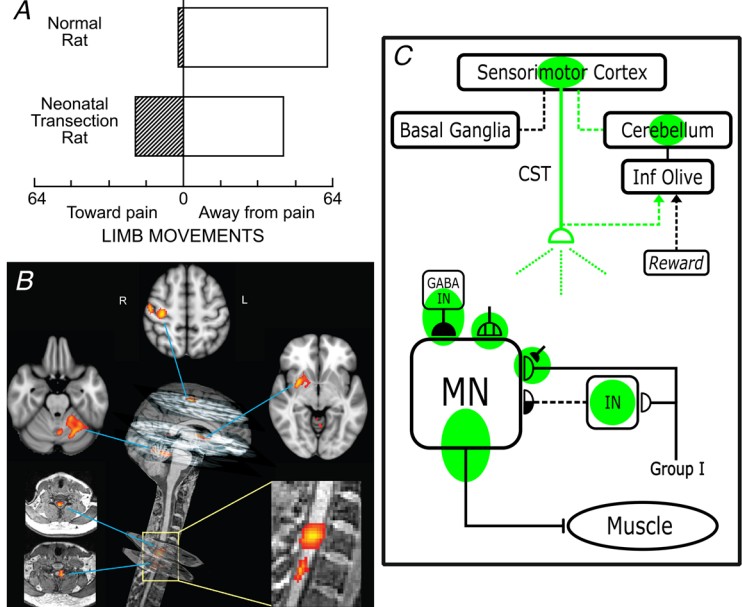

**Figure 1. Recent insights about adaptive behaviours**

*A*, Even the simplest adaptive behaviours are shaped by experience. The direction of limb movement produced by flexion-withdrawal responses to a nociceptive stimulus in normal adult rats and in adult rats in which the spinal cord had been transected just after birth. The direction is appropriate (i.e. away from the stimulus) in normal adults but is often inappropriate in transected adults. Neonatal transection prevents continuation of the normal activity-dependent shaping of flexion-withdrawal reflexes that begins *in utero*. (Modified from Levinsson et al. (1999). Copyright (1999) Society for Neuroscience.) *B* and *C*, behaviors are produced by widely distributed networks of neurons and synapses. *B*, the CNS substrate of a motor sequence behaviour (i.e. a specific sequence of individual finger flexions) revealed by fMRI during the behaviour. Distinct cortical, subcortical, and spinal areas are active (i.e. yellow/red areas). Activation in each area correlates positively with performance speed. Cortical activation is in contralateral sensorimotor cortex. Subcortical activation is in contralateral putamen and ipsilateral cerebellar lobules V–VI. Spinal activation is focused in spinal segments C7–C8. The cortical, subcortical, and spinal activations make both overlapping and independent contributions to performance. (From Vahdat et al. (2015).) *C*, the CNS substrate of operantly conditioned change in the size of the H-reflex (electrical analog of the spinal stretch reflex). The green ovals indicate the spinal and supraspinal sites of definite or probable CNS plasticity associated with H-reflex operant conditioning. CST, main corticospinal tract; GABA IN, GABAergic spinal interneuron; IN, spinal interneuron; MN, spinal motoneuron. Dashed pathways imply the possibility of intervening spinal inter-neurons. Open synaptic terminals are excitatory; filled ones are inhibitory; mixed may be either. The monosynaptic and probably oligosynaptic H-reflex pathway from the sensory afferents to the motoneuron is shown. Definite or probable sites of plasticity include the following: the motoneuron membrane (i.e. firing threshold and axonal conduction velocity); GABAergic interneurons; GABAergic terminals and C-terminals (the segmented terminals) on the motoneuron; monosynaptic afferent terminals on the motoneuron and/or their presynaptic contacts; terminals conveying oligosynaptic afferent inhibition or excitation to the motoneuron; sensorimotor cortex; and cerebellum. The data support the hypothesis that the reward contingency acts through the inferior olive to guide and maintain plasticity in the cerebellum that guides and maintains plasticity in sensorimotor cortex that (via the CST) guides and maintains plasticity in the spinal cord that is directly responsible for H-reflex change. (For review: Thompson & Wolpaw, 2014.) (Updated from Wolpaw (2010) and Wolpaw & Tennissen (2001).)

and disease, and other life events that affect the CNS itself and the peripheral sensory and motor structures through which it interacts with the body and the world.

In sum, the lifelong maintenance of an adaptive behaviour indicates that the widely distributed neurons and synapses that produce the behaviour are continually changing in response to new plasticity. Even before the life-long plasticity of the CNS was recognized, Bernstein (1967) noted that the muscle activations and kinematics of an adaptive behaviour change; he described these behaviours as 'biodynamic structures [that] live and develop'. As many research groups have demonstrated, the ongoing changes in the neurons and synapses that produce an adaptive behaviour are guided by feedback during the behaviour and by its outcome; these continual changes reduce errors and improve performance (Adams, 1987; Criscimagna-Hemminger et al., 2010; Kim et al., 2019; Krakauer & Mazzoni, 2011; Pacheco et al., 2019; Roemmich et al., 2016). This process shapes the behaviour during its original acquisition and it maintains the behaviour through life.

**A negotiated equilibrium.** It is now evident that adaptive behaviours are acquired and maintained in a ubiquitously plastic CNS. The mechanisms of this plasticity are numerous and local; and each is usually separated from its ultimate behavioural impact by multiple neurons and their synaptic connections, which are themselves plastic. Thus, these mechanisms are controlled imperfectly – if at all – by their behavioural impact. In addition, each behaviour is produced by a widely distributed network of neurons and synapses. These networks overlap; individual neurons and synapses contribute to multiple behaviours. Furthermore, each network is continually changing to maintain its own behaviour despite acquisition of new behaviours, despite concurrent changes in the overlapping networks that produce other behaviours, and despite growth, ageing, and other life events. This looks like a recipe for chaos, but it is not. The question is: why isn't it? How are adaptive behaviours maintained throughout life? What organizes the many concurrent changes involving many different sites and kinds of plasticity so that all the numerous adaptive behaviours that the CNS produces are consistently satisfactory? In short, how is the problem of acquiring and maintaining many adaptive behaviours in a ubiquitously plastic CNS solved?

For example, when acquisition of a new adaptive behaviour changes the properties of the neurons and synapses comprising the new network that produces the new behaviour, how are existing (i.e. old) behaviours that use these same neurons and synapses maintained? This question is really two questions. First, how is an individual adaptive behaviour maintained? Second, how does it come about that all adaptive behaviours

are maintained simultaneously? The answer to the first question seems clear. As described above, the network of neurons and synapses underlying an adaptive behaviour changes continually to reduce errors and improve performance; these changes are guided by feedback during the behaviour and by its outcome (Adams, 1987; Criscimagna-Hemminger et al., 2010; Kim et al., 2019; Krakauer & Mazzoni, 2011; Pacheco et al., 2019; Roemmich et al., 2016). Thus, an existing network will respond to the errors in its behaviour resulting from the plasticity associated with acquisition of a new behaviour (or associated with growth, ageing, or other life events) by changing (i.e. by undergoing further plasticity) to reduce the errors. This then is how the network of a single existing behaviour responds to the problem caused by a new network. How then are the efforts of all the networks organized so that they are all successful in maintaining their behaviours?

In 2010, the term *negotiated equilibrium* was introduced to acknowledge and begin to explore the composite process that enables the many widely distributed, overlapping, and continually changing networks of neurons and synapses underlying adaptive behaviours to maintain their behaviours when a new network is created (or when growth, ageing, or other life events produce CNS plasticity) (Wolpaw, 2010, 2018). *Negotiated equilibrium* is both a name for this necessary but as yet mysterious process and a conception of its nature. The concept hypothesizes that the process is organized by the networks themselves. It proposes that their concurrent individual efforts to maintain their own behaviours constitute a negotiation among them; they negotiate the properties of the CNS neurons and synapses that they all use. Through this process, they establish and maintain an equilibrium satisfactory to all of them. While this process may change the electromyographic (EMG) and kinematic details of an adaptive behaviour, it maintains the behaviour's key features, the features that indicate the behaviour is satisfactory. For example, the key features of locomotion include attributes such as right–left step symmetry, upright posture, adequate balance, and acceptable metabolic cost; they do not include a defined sequence of muscle contractions or fully symmetrical joint angle changes.

This separation of behavioural measures into those that are tightly controlled and those that are not, first emphasized by Bernstein (Bernstein, 1967; Latash, 2020) in the mid-20th century, is now embodied in the *uncontrolled manifold* analysis (Chang et al., 2009; Latash et al., 2007; Scholz & Schöner, 1999). The key features of the behaviour, which determine whether the behaviour is satisfactory, are tightly controlled. For finger withdrawal from a hot stove, the key feature is the height of the finger-tip relative to the stove top; this height must increase very quickly. Other measures, such

as the particular elbow, wrist, and palm locations that combine to determine finger-tip height, are not key features; they comprise the uncontrolled manifold and are not tightly controlled. By this analysis, maintaining a behaviour means maintaining its key features. This is the function of the continually changing network of neurons and synapses underlying the behaviour; it changes as needed to maintain the key features of the behaviour.

The negotiated equilibrium described here is similar to a Nash equilibrium as defined in game theory (Davis, 1983; Fudenberg & Tirole, 2005; Gibbons, 1992). The players (i.e. the current repertoire of adaptive behaviours) negotiate an equilibrium in which none can better its performance by changing further. In the healthy CNS, this negotiated equilibrium enables each behaviour to be satisfactory (i.e. 'good enough'; Loeb, 2012). By disturbing this equilibrium, new adaptive behaviours (new players) and other life events (e.g. growth, ageing) cause further negotiation that leads to a new negotiated equilibrium. This negotiated equilibrium is similar to an open-system non-equilibrium steady state in thermodynamics (Qian, 2006).

The negotiated equilibrium concept appears to be the most obvious answer to the problem of how it is that many adaptive behaviours are concurrently maintained in a ubiquitously plastic CNS. It simply starts from what many studies have already established: that the network of neurons and synapses underlying an adaptive behaviour changes continually to maintain the key features of its behaviour (Adams, 1987; Criscimagna-Hemminger et al., 2010; Kim et al., 2019; Krakauer & Mazzoni, 2011; Pacheco et al., 2019; Roemmich et al., 2016). The negotiated equilibrium concept then hypothesizes that these concurrent individual efforts constitute a negotiation among the networks that creates a CNS state satisfactory to them all. The mechanisms of the negotiation remain to be defined. This hypothesis is simpler than the alternative hypothesis that the CNS has a central executive that makes appropriate changes in the network of each old adaptive behaviour that is affected by a new behaviour, changes that are somehow coordinated so that every one of many old behaviours remains satisfactory. This alternative central executive hypothesis lacks any supporting evidence. In contrast, the negotiated equilibrium concept is based on experimental evidence indicating how an individual adaptive behaviour is maintained.

If the negotiated equilibrium concept is correct, the network underlying an adaptive behaviour has unique properties that support its internal operation and its inter-actions with other networks. It has properties that enable it to retain the key features of its behaviour and to produce this behaviour through life; and it has properties that enable it to join with other networks in establishing what is essentially a Nash equilibrium. If these networks do have

these unique properties, the non-specific phrase 'a widely distributed continually changing network of neurons and synapses' is not adequate for the conceptual and experimental endeavours needed to explore the networks and the negotiated equilibrium that they are hypothesized to create; and it will become increasingly inadequate as research in this area advances. As the mechanisms that operate within and among these neuronal and synaptic networks are illuminated, the networks' status as a unique and uniquely important CNS entity is likely to become more and more evident. Thus, these networks need a name, a simple, easy-to-use name that reflects their plastic properties and their active role in CNS function.

A comparable situation arose in the 1890s, when the neuron doctrine was firmly established and the connection between neurons was first recognized as a unique and important structure that needed a name. After dismissing generic terms such as 'junction' or 'conjunction', Sherrington introduced the new word *synapse* at the suggestion of the Classical scholar A. W. Verrall (Foster & Sherrington, 1897; Tansey, 1997). We tried to follow this admirable example to find an appropriate name for the CNS substrate of an adaptive behaviour. What we found was remarkable. Aristotle and the Stoic philosophers who followed him had a name for an entity very similar to what we would now call 'the widely distributed and continually changing network of neurons and synapses underlying an adaptive behaviour'. They defined its properties and described its substance as best they could. Now, 2200 years later, we are able to go considerably further in exploring and describing this entity; and we need the name they left for us.

## An ancient concept and a new word

*Heksor* is the name we are giving to this newly recognized entity. The process of creating this new English word had two parts, one philosophical and one linguistic. The philosophical part was finding an appropriate classical term to serve as the root for the new word, a root that embodied the properties and function of the entity we were naming. The linguistic part was deriving from that term a simple and distinctive English word that implies what we believe to be the most important properties of the entity.

In the following discussion, ancient sources are cited in parentheses, according to author, work, and standard book and chapter number, or in the case of single-book works, by chapter number, or according to other standard reference units. These divisions and units are reproduced in nearly all editions of high quality. The collection of Stoic fragments, abbreviated *SVF = Stoicorum veterum fragmenta*, vols. I–III, ed. H. von Arnim, B.G. Teubner, Leipzig (1903–1905), is cited by volume and fragment number.

**Philosophical background.** The term *hexis* (in the Greek alphabet ἕξις), from which we have derived the word *heksor*, comes from the realm of ancient Greek philosophy. It is best known in the sense in which it was used by Aristotle, but it is the Stoic usage that has proved most significant for our proposal.

In his ethical theory, Aristotle used the term *hexis* to describe what he meant by moral virtue, and it has often been translated as 'state of character'. For Aristotle, the practical component in ethics was very important. Socrates had claimed that virtue is knowledge, but Aristotle sought to develop this idea. As he saw it, what we know is important, but in the acquisition of moral excellence, this has to be translated from potential into action. Accordingly, it is the doing and repeating of virtuous acts, let us say for example, temperate or just acts, that causes a person to become temperate or just. The doing and habitual practice of the act lead to a *hexis* or 'state of character' as their result. The person then has in his/her stable possession the virtue of temperance, justice, etc., and is able with greater ease to continue to perform virtuous actions (Reale, 1989).

However, Aristotle saw the development of the virtues as *hexeis* (plural of *hexis*) in the same way that he saw the development of the skills or arts, in Greek, *technai* (plural of *technē*). That is, one acquires a certain skill or art by doing it. To use his examples, a person becomes a lyre-player by playing the lyre, and a builder by building. And one becomes a good lyre-player by playing the lyre well, and a good builder by building well. Consequently, we would also consider the arts as *hexeis* (Aristotle, *Nic. Eth.* 2.1, with Reale, 1987).

It is this latter circumstance that has led us to seek a variation of this term to describe the CNS substrate of an adaptive behaviour. For it is easier and more suitable for us to equate the arts or skills with 'useful behaviours' than it is to enter into a discussion about the nature of the virtues and other more complicated moral issues. And indeed, it just so happens the Stoics discarded Aristotle's use of the term *hexis* for the virtues, but they retained it for the arts. They preferred to characterize the virtues as *diatheseis*, that is, 'dispositions'. According to the language of our ancient texts, the Stoics classed the virtues as 'dispositions', while the 'pursuits' (Greek, *epitēdeumata*), such as (the art of?) divination they reckoned as *hexeis* (*SVF* III.104—5). While this formulation seems unusual to us, we can determine from other sources that under the more general rubric of 'pursuits' they also placed the arts in general (*technai*), and specifically the liberal arts, which included rhetoric, music, and astronomy (*SVF* III.294). The implication of this would be that the arts are to be considered *hexeis*. And indeed, we know that two of the early Stoics, Zeno and Cleanthes, defined *technē*, that is, art or technique, as *hexis* (*SVF* I.72, 490). Also under 'pursuits', and therefore, among the *hexeis*, are what

we might call hobbies, such as 'love of music', 'love of horseback riding', 'love of hunting' (*SVF* III.294). It is not entirely certain that these hobbies are to be differentiated, in this context, from the arts (cf. Giusta, 1967).

The basis for the distinction that the Stoics made between the virtues, as 'dispositions', and the arts, as *hexeis*, is also relevant to our choice of a derivative of the latter term to indicate the CNS substrate of an adaptive behaviour. For the Stoics thought that the virtues were more stable than the *hexeis*. This meant for them that the virtues, once acquired, are not subject to 'intensification or relaxation'. The *hexeis*, by contrast, and specifically the arts, are subject to these kinds of changes (*SVF* II.393). To say this in another way, once one has become brave or temperate, one remains so. By contrast, one's skill in music, arithmetic or horseback riding may increase or decrease. In view of the more advanced conception of the plasticity of the CNS, this Stoic characterization of the *hexis* has been pertinent in our considerations.

Another significant component of the Stoic understanding of the *hexis* relates to its material nature. Plato had viewed the soul, and especially its rational component, the mind, as something immaterial or incorporeal. Indeed, it was largely on this basis that he was a proponent of the immortality of the soul or mind. In his famous dialogue about the immortality of the soul, the *Phaedo*, Plato speaks of the soul as 'invisible' and 'intelligible' (79b, 80b). However, he implies its incorporeality in *Sophist* 247b–d, and uses the word 'incorporeal' (Greek, *asōmatos*) with reference to the soul in *Epinomis* 981b, although this latter dialogue may have been written by a pupil. In any case, it was a point that later Platonists took for granted (Alcinous, *Didaskalikos* 25.1). By contrast, the Stoics, in accord with the tendencies of Hellenistic thought in general, conceived of the soul and the mind as material. Zeno, the founder of Stoicism, defined the soul as *pneuma*, or spirit (*SVF* I.135). Now, we often associate this word with the immaterial, and it may be acknowledged that this association goes back to ancient times. But it came about as a result of a kind of 'dematerialization' of *pneuma* in Greek linguistic usage, and that development took place only at the close of the Hellenistic period, when Platonistic tendencies again became dominant (Rüsche, 1933). For the Stoics, *pneuma* was a material substance, a kind of hot breath, and the soul was, as they put it, 'a body' (*SVF* II.773). The mind, in as much as it is part of the soul, was also defined as *pneuma* (Philo, *On Flight* 134; Seneca, *Epistle* 50.6). In an ancient source, we find this form of *pneuma* described as *leptomeres*, i.e. 'made up of fine particles' (*SVF* II.785), and some moderns have spoken of a 'fine-textured material' (Graver & Long, 2015). Accordingly, in the present context, we should probably speak of the 'fine matter' of the brain.

The result of this view of the mind was that many psychic and mental states and functions came to be

conceived in strictly material terms. They were under-stood as different configurations of the substance or *pneuma* of the mind, or, to use the Stoic way of speaking, 'the (*pneuma* of the) mind as configured in a certain way' (Pohlenz, 1965). The psychic and mental states and functions that they viewed in this manner include the virtues, the vices, the emotions, and most importantly for our purposes, the *technai* or skills/arts. In speaking of the first category, Seneca says, 'virtue is nothing other than the mind configured in a certain manner'. And he goes on to speak of the arts as similar phenomena (*Epistle* 113.2−3). That the arts were conceived in this material way can be confirmed from the testimony of Plutarch (*On Common Conceptions* 45). We see then that within the context of their more general conception of mental states and processes, the Stoics did not make a distinction between the virtues, as 'dispositions', and the arts and pursuits, as *hexeis*.

When the Stoics speak of mental processes as 'the mind as configured in a certain way' they are relying on the third of the four so-called categories they employed in their description of reality, that of 'being in a certain state'. This third category was used to describe not the permanent qualities of any given entity, but rather, as A.A. Long has put it, 'what it is about some individual [entity] which permits us to describe it as being somewhere, being at some time, acting, having a certain size, being coloured and so forth' (Long, 1986). It is within this broader conception of a material reality subject to change and alteration that we may appreciate their understanding of mental states and processes, and specifically, *hexeis*.

In conclusion, we may say that there are three reasons for our choice of a variation of the word *hexis* to describe the CNS substrate of an adaptive behaviour. This term was used by the ancients to describe the mental aspect of, among other things, skills and 'pursuits'. And, in the view of the Stoics specifically, a *hexis* was seen to be apt to change, and to be of a material nature.

**Linguistic considerations.** In theory, it would have been possible to simply use the older term *hexis* in this new context, namely, to indicate the CNS substrate of an adaptive behaviour. However, the word has been used for centuries in philosophical literature. Moreover, in the philosophical discussions, the term is usually employed in the Aristotelian sense, and the Stoic connotations, particularly important for our specific purposes, have been left behind. Accordingly, it seemed best to offer a new coinage, while maintaining the connection with the old word.

In proposing a new word, we needed to take into account various factors. In the first place, since the term will be used mostly in scientific circles, we thought it prudent to avoid the form *hex-*, since this is often used in the (originally Greek) sense of 'six'. Therefore, we have replaced this formation with *heks-*. This is a completely justifiable substitution, because according to the ancient Greek authorities, the consonant *xi* (Greek *ξ*) is a 'double' consonant, standing for *kappa* (*κ*) plus *sigma* (*σ*) (Dionysius Thrax, *Ars gramm.* 6; Dionysius of Halicarnassus, *Lit. Comp.* 14). Indeed, it is to be noted that the adjective derived from the noun *hexis*, with the meaning 'habitual', has the form *hektikos* (ἑκτικός).

Moreover, since the newer conception of the CNS sub-strate of an adaptive behaviour entails an active role for that entity, we thought it best to add a suffix of agency, namely, *-(s)or*. This suffix is commonly employed in English, e.g. in the word 'actor', in the anatomical terms, 'incisor', 'detrusor', 'flexor', and in the general-purpose biomedical term 'effector'. However, its roots go well back in the history of the Indo-European languages. It may well have been an original *\*-tor-*, which in Latin, for example, became *-tor* or *-sor* (Buck, 1933; Sihler, 1995). When we add this suffix *-(s)or* to *heks-*, we emerge with the form *heksor*.

**Heksor: definition, justification, and explication.** A *heksor is a widely distributed network of neurons and synapses that produces an adaptive behaviour and changes itself as needed in order to maintain the key features of the behaviour, the attributes that make the behaviour satisfactory*. The plural is *heksors*; the adjective is *heksoric*.

*Heksor* is both a word and a concept. Justifying the word *heksor* is straightforward. The word simply names the distributed continually changing network of neurons and synapses that underlies an adaptive behaviour (e.g. Fig. 1*B* and *C*). In contrast, justifying the *heksor* concept is complicated and potentially controversial. First, it requires clear definition of what is a heksor and what is not. Second, it requires clear distinction between the heksor concept and existing concepts related to adaptive behaviours. Third, it requires experimental evidence for heksors. Fourth, it requires realistic strategies for testing the applicability of the heksor concept to the entire range of adaptive behaviours. Justification of the heksor concept begins here with basic explication and occupies the rest of this paper.

As indicated above, an adaptive behaviour has attributes, or key features, by which it can be judged satisfactory or unsatisfactory. The goal of its heksor is to maintain these key features. If the heksor is successful, the behaviour is satisfactory. To maintain these key features, the heksor may change the properties of its neurons and synapses (e.g. synaptic strengths, neuronal firing thresholds; Carp & Wolpaw, 1994; Hawkins et al., 1993). The heksor may also change the set of neurons and synapses that comprise it. Additional neurons and

synapses may be incorporated into it; others may be eliminated from it.

The distinction between non-adaptive behaviours, which are not produced by heksors, and adaptive behaviours, which are produced by heksors, is illustrated by a simple example. The simplest behaviour of the mammalian CNS is the largely monosynaptic stretch reflex, also known as the spinal stretch reflex (SSR) or 'knee-jerk' reflex (Pierrot-Deseilligny & Burke, 2012). Sudden muscle stretch excites primary afferent fibres that excite the spinal motoneurons and cause the muscle to contract, producing the SSR. The main attribute of the SSR is its size, which may be measured as the force it produces, or (more easily) as its EMG amplitude. Normally, in the healthy CNS, the SSR itself is not an adaptive behaviour. It contributes to adaptive behaviours such as locomotion (e.g. by regulating muscle stiffness; Carter et al., 1990; Cronin et al., 2011; Nichols & Houk, 1976; Shadmehr & Arbib, 1992; Sinkjaer et al., 1988). However, the SSR does not by itself alone serve a specific need of the individual. Thus, it is not normative; its size is neither good nor bad, satisfactory nor unsatisfactory. And, in fact, SSR size does vary greatly across people who are entirely intact neuro-logically: the knee-jerk reflex may be barely detectable in one person and large in another. Both are acceptable.

Nevertheless, the SSR can be made into an adaptive behaviour. This was initially accomplished for the SSR of the monkey biceps muscle (Wolpaw et al., 1983). An operant conditioning protocol gave a reward that was contingent solely on SSR size: the up-conditioning protocol rewarded SSR size that was above a criterion value; the down-conditioning protocol rewarded SSR size that was below a criterion. In animals exposed to the up-conditioning protocol, SSR size gradually increased; in those exposed to the down-conditioning protocol, it gradually decreased. The protocol made the biceps SSR into an adaptive behaviour with one key feature: it was satisfactory (i.e. it produced reward) when its size was larger for up-conditioning (or smaller for down-conditioning) than a criterion. In sum, by making reward contingent on this single feature – biceps SSR size – the operant conditioning protocol gradually creates, from among all the neurons and synapses that might affect SSR size, a network that henceforth changes itself as needed to maintain this key feature. The distributed network of neurons and synapses that retains this key feature and produces an SSR that has this key feature is a one-feature heksor. Figure 1*C* illustrates current knowledge of this heksor based on further operant conditioning studies of the SSR and its electrical analogue the H-reflex (for review, Thompson & Wolpaw, 2014; Wolpaw, 2018). By making the SSR or the H-reflex into a behaviour that rewards the individual when its key feature is achieved, the operant conditioning protocol changes it from a non-adaptive behaviour into an adaptive behaviour,

and creates the heksor that henceforth maintains the behaviour's key feature: satisfactory size. SSR and/or H-reflex size has now been operantly conditioned in upper and/or lower extremity muscles of monkeys, rats, mice, and humans. The acquisition, magnitude, persistence, and other characteristics of this adaptive behaviour are similar across species; and physiological and anatomical studies indicate that its CNS substrate – the heksor that maintains the behaviour – is also similar across species (Fig. 1*C*; for review, Thompson & Wolpaw, 2014; Wolpaw, 2018).

Just as the key feature created by the conditioning protocol converts the SSR or H-reflex into an adaptive behaviour, the key features of a more complex adaptive behaviour define it as a single adaptive behaviour, rather than several, and the heksor that produces and maintains it as a single heksor, rather than several. The locomotion heksor produces and maintains concurrent movements of the left and right sides of the body. Why then is locomotion not the product of two heksors, one controlling the right side and one controlling the left? Certainly, the movement of either side has key features that need to be satisfied for locomotion to be satisfactory. And satisfying the key features of the right side and those of the left will produce right-side movement and left-side movement appropriate for locomotion. However, neither of these alone is an adaptive behaviour, nor will one plus the other produce good locomotion. Additional key features must be satisfied; these features define the essential right–left symmetries (e.g. right–left symmetry in step timing and length). These key features, essential for normal locomotion, are features of the single adaptive behaviour of locomotion, and thus part of the mission of the single locomotion heksor. Neither right nor left movement alone provides these symmetries, they are properties of the two sides in combination, and thus of a single locomotion heksor.

The specification of what constitutes a single heksor rather than multiple heksors may not always be as obvious as it is for locomotion. It may become more complicated as individual adaptive behaviours are combined into more complex multistage adaptive behaviours. Or an experimental protocol that imposes a complex force field on standard movements such as reaching may raise uncertainty as to whether it creates a new heksor (e.g. Crevecoeur et al., 2020; Lackner & DiZio, 1994). Nevertheless, the principle is clear: an individual heksor is recognized and identified by the entire set of the key features of the behaviour that it produces and maintains. This is illustrated in the next section with the two adaptive behaviours of locomotion and ballet.

The entity described by the new term *heksor* is distinctly and importantly different from the entity described by existing terms often applied to the substrates of adaptive behaviours, such as *memory* and similar terms such as *engram* or *internal representation* (Dudai, 2002;

Eichenbaum, 2016; Josselyn et al., 2017; Kandel et al., 2021; Semon, 1921; Tonegawa et al., 2015). Like memories, heksors are created by experience; but there the similarity ends. The difference is simple: memories are passive; heksors are active. The CNS uses memories to guide behaviour; they are accessed when they are needed. Heksors use the CNS to produce their behaviours; they modify their neurons and synapses as needed to ensure that their behaviours are satisfactory. Since many heksors share the CNS, their concurrent changes are a negotiation among them; this negotiation ensures the maintenance of all their behaviours. The old term memory was suitable for the largely hardwired CNS of 1970; the new term heksor is suitable for the ubiquitously plastic CNS of today.

It should be noted that the heksor *vs.* memory comparison made here is confined to procedural memory, or memory for skills (i.e. adaptive behaviours); it does not address declarative memory (Dudai 2002). The heksor concept does extend to declarative memory, but that exposition will require a paper of its own.

The heksor and negotiated equilibrium concepts are compatible with models that seek to explain how the CNS produces individual adaptive behaviours (e.g. Feldman, 1966; Ingram et al., 2017). The new concepts are not an alternative to, nor do they conflict with these models. The heksor and negotiated equilibrium concepts address a different problem: they seek to explain how numerous adaptive behaviours share the ubiquitously plastic CNS. Furthermore, assuming that through their negotiations heksors accommodate each other for the overall benefit of the individual, these new concepts are compatible with models that define the terms of this accommodation (e.g. Friston, 2010). The specific contribution of the heksor and negotiated equilibrium concepts is that they begin to explain how numerous adaptive behaviours are acquired and maintained in a ubiquitously plastic CNS. Figure 2*A* illustrates the process through which heksors are hypothesized to create a negotiated equilibrium of neuronal and synaptic properties. As described below, studies in animals and humans already support this hypothesis. Further studies that test it more comprehensively are essential, and many questions invite exploration.

### Present evidence for heksors and the negotiated equilibrium they create

**Imaging studies.** Over the last several decades, many studies have used non-invasive imaging technologies, particularly functional magnetic resonance imaging (fMRI), to localize the CNS activity underlying a newly acquired behaviour (e.g. Fig. 1*B*). A few of these studies have also examined the impact of the new behaviour on the activity underlying an old (i.e. pre-existing) behaviour that shares neurons and synapses with the new behaviour.

Their results indicate that the plasticity associated with a new behaviour goes substantially beyond that directly responsible for the new behaviour; it includes changes in the activity responsible for the old behaviour as well. The extensive plasticity associated with acquisition of the new behaviour is readily explicable as the outcome of an essential and ultimately successful negotiation between the new and old heksors. This outcome ensures that both the new and old heksors maintain their key features.

In one such study, Sacco et al. (2009): (1) taught people a simple foot plantarflexion/dorsiflexion behaviour; (2) recorded fMRI while they performed it; (3) trained them for a week on a new dance style; and (4) again recorded fMRI while they performed the same simple foot plantarflexion/dorsiflexion behaviour. The investigators then compared the fMRI results before the dance training to the results after the training. Figure 2*B* summarizes their findings. The cortical and subcortical activity associated with the old foot plantarflexion/dorsiflexion behaviour is different after the new dance behaviour is acquired. Activity differs significantly in several frontal and parietal cortical areas, in putamen, and in cerebellum. Furthermore, as Fig. 2*B* also shows, functional connectivity among active cortical areas is increased. These results indicate that acquisition of the new dance heksor has changed the old foot plantarflexion/dorsiflexion heksor; the old behaviour is maintained, but the CNS activity that produces it has changed.

Parker Jones et al. (2012) compared in monolingual and bilingual people the brain activity associated with naming pictures or reading aloud in either language. As Fig. 2*C* shows, a second language changes the activity that produces the first language. In bilingual people who are speaking either one of their two languages, several cortical areas participate much more than they do in people who can speak only one language. The need for two languages to share brain areas poses difficulties for each of them (e.g. using or understanding words in the two languages that sound similar but have different meanings, or sound different but have the same meaning; ensuring language-specific pronunciation and interpretation of pronunciation; parsing language-specific sentence structures). Such requirements may explain why the heksor responsible for a second language is more extensive than that responsible for the one language of a monolingual person, and why the creation of the new language heksor is associated with similar expansion of the old (i.e. native) language heksor.

Zou et al. (2012) examined the change in the brain activity associated with a language when people learned to sign in the same language. This is a particularly interesting situation. In the new behaviour, the same language is now linked to and dependent on a different sensory modality (vision *vs.* audition) and a different

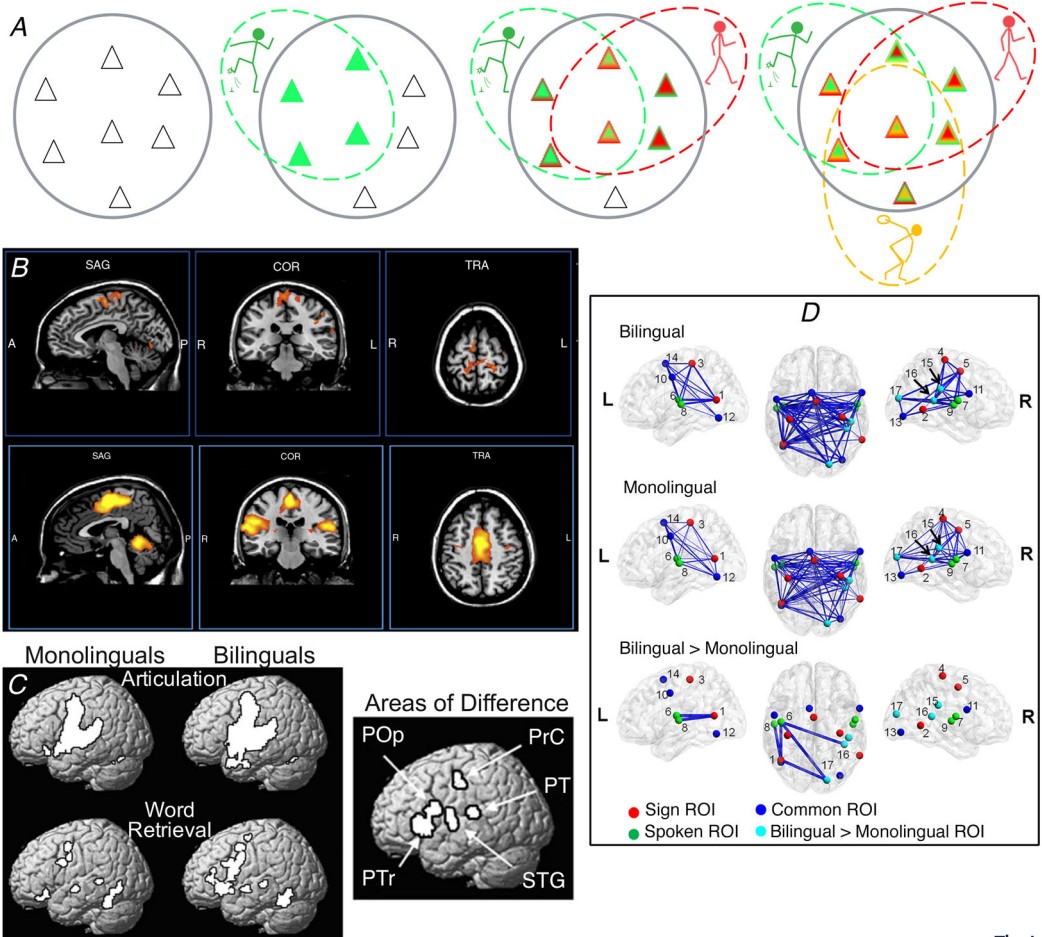

The Journal of **Physiology**

**Figure 2. Creation of a new heksor may change old heksors**

*A*, Illustration of heksor negotiation through life. The first image shows a naïve CNS. Each triangle represents one of the many CNS neurons or synapses; their properties can undergo activity-dependent plasticity. In the second image, in the prenatal and early postnatal periods, flexion-withdrawal reflex behaviour is acquired. Its heksor comprises a set of neurons and synapses (green) with properties that have changed to produce the new behaviour. In the third image, early in life, the behaviour of locomotion is acquired. Its heksor (red) overlaps the flexion-withdrawal heksor. The two heksors negotiate the properties of the neurons and synapses that they share so that each heksor can maintain the key features of its behaviour. This negotiation necessarily involves the properties of their unshared neurons and synapses as well. Thus, the state of each neuron or synapse in either heksor may be affected by both of them. In the fourth image, later in life, the athletic behaviour of throwing the discus is acquired. Its heksor (orange) overlaps the other two. The three heksors negotiate the properties of the neurons and synapses they share, and this also affects those that they do not share. Thus, the state of each neuron or synapse in each of the three heksors may be affected by all three of them. Together, the heksors keep the CNS in a state of negotiated equilibrium that maintains the key features of the behaviour produced by each heksor. *B*, 1 week of dance training changes the cortical activity underlying a simple foot plantarflexion/dorsiflexion behaviour as measured by fMRI. The top orthogonal images show areas of differential activation: post-training condition minus pre-training condition. The bottom images show areas of differential connectivity: post-training condition minus pre-training condition. The new heksor responsible for the new dance behaviour leads to changes in the old heksor responsible for the old foot plantarflexion/dorsiflexion behaviour. The brain activity that produces the old behaviour changes; the key features of the old behaviour are maintained. (Reprinted from Sacco et al. (2009), with permission from Elsevier.) *C*, In bilingual people fluent in English and another language, the brain activity (measured by fMRI) during use of either language differs significantly from the brain activity during use of English by people who speak only English. Left: White indicates areas active during articulation or word retrieval. Activation is more extensive in people who are bilingual than in those who are monolingual. Right: Areas that show greater activation in bilingual people (Pop: pars opercularis; PrC: dorsal precentral gyrus; PT: planum temporale; PTr: ventral pars triangularis; STG: superior temporal gyrus). (From Parker Jones et al. (2012). Reprinted by permission of Oxford University Press.) *D*, Functional connectivities for bilingual people (people who could both speak and sign Chinese) and

monolingual people (people who could only speak Chinese) during spoken language production. Top: Significant functional connectivities for bilingual people across all regions of interest (ROIs). Middle: Significant functional connectivities for monolingual people across all ROIs. Bottom: Increased functional connectivities for bilingual relative to monolingual people. Thicker lines indicate greater correlation coefficients. ROI numbers: 1, left (L) middle temporal gyrus (MTG); 2, right (R) MTG; 3, L precentral gyrus (PG); 4, R PG; 5, R middle cingulate gyrus; 6, L Rolandic; 7, R Rolandic; 8, L superior temporal gyrus (STG); 9, R STG; 10, L PG; 11, R PG; 12, L inferior occipital gyrus (IOG); 13, R IOG; 14, L supplementary motor area; 15, R supramarginal gyrus; 16, R STG; 17, R superior occipital gyrus. (Reprinted from Zou et al. (2012), with permission from Elsevier.)

motor output (hand control *vs.* speech). Thus, the two heksors overlap considerably in the CNS areas underlying language comprehension and generation (overtly cognitive functions), while they overlap minimally in the CNS areas directly connected to language-related sensory inputs or motor outputs (overtly sensorimotor functions). When speaking the language, people who could also sign in the language showed greater activation in several cortical areas than people who could only speak the language. Furthermore, as Fig. 2*D* illustrates, people who could both speak and sign showed additional functional connections among cortical areas that were not present in those who could only speak. Once again, it appears that creation of the new sign-language heksor was associated with changes in, including expansion of, the old spoken-language heksor.

In addition to these imaging studies, highly specialized physical training provides further examples of the impact of a new behaviour on the plasticity responsible for an old one. One striking example is that of ballet dancers. As they gradually acquire the new ballet behaviour over prolonged training, the key features of their locomotion are maintained, but the responsible muscle activity and kinematics change. They walk so much differently from non-dancers that they are easily recognized on the street (e.g. Kilgannon (1996)). Distinctive changes in spinal reflex pathways comprise part of the plasticity associated with acquisition of the new behaviour (Nielsen et al., 1993; Perez et al., 2007). Among these changes is marked reduction in reciprocal inhibition between antagonist muscles. This reduction probably contributes to the agonist–antagonist co-contraction essential in ballet. The new ballet heksor and the old locomotion heksor overlap in both the spinal cord and the brain; they accommodate each other so that the key features of both are achieved. At the same time, the key features of ballet differ in important respects from those of locomotion (e.g. in joint angle symmetry requirements, postural requirements, required rhythmicity with auditory input (e.g. music), metabolic criteria, toleration of musculoskeletal trauma). Thus, they remain two different heksors. A dancer might simultaneously perform good locomotion but bad ballet.

The recent imaging studies reviewed above show the impact of a variety of new behaviours on old behaviours that use the same CNS regions. Their results are broadly consistent with the new complementary concepts of

heksors and the negotiated equilibrium that heksors create. At the same time, the behaviours they study are in general too complex for a detailed evaluation of the credibility and usefulness of these two new concepts. For example, fMRI lacks the resolution needed to verify that the overlapping activity associated with two different language heksors is occurring in the same population of neurons and synapses, rather than in two separate populations that are simply intermingled with each other.

Beginning in the 1980s, such detailed evaluation has come from studies exploring the impact of H-reflex operant conditioning in normal animals and in animals and humans with CNS injury (for review, Thompson & Wolpaw, 2014; Wolpaw, 2018). These studies have used the H-reflex (the electrical analogue of the SSR) rather than the SSR itself because the H-reflex greatly facilitates experiments; and, as indicated above, all data to date indicate that SSR and H-reflex operant conditioning are very similar phenomena. The startling results in the earliest of these studies were the first impetus toward the heksor and negotiated equilibrium concepts.

This still-ongoing work takes advantage of the spinal cord's accessibility, simplicity, and well-defined connections to the brain. Most important, it takes advantage of the spinal cord's role as the final common pathway for behaviours and its immediate proximity to behaviour. The spinal plasticity associated with a new behaviour is likely to affect old behaviours because they too use the spinal cord. And, because the spinal cord is directly connected to behaviour, these effects on old behaviours are clear, their aetiologies can be discerned, and their mechanisms can be more readily explored. In this body of work, the new behaviour is a larger or smaller soleus H-reflex acquired through operant conditioning (Carp et al., 2006; Chen & Wolpaw, 1995; Thompson et al., 2009; Wolpaw, 1987), or a larger tibialis anterior motor evoked potential (MEP) (evoked by transcranial magnetic stimulation; TMS) also acquired through operant conditioning (Thompson & Sinkjær, 2020; Thompson et al., 2018a, 2018b, 2019). The old (i.e. pre-existing) behaviour is locomotion.

We emphasize that the principal purpose of these still ongoing studies (and of this paper) is not to further illuminate the mechanisms of the several behaviours involved, such as locomotion. Locomotion has been studied for decades and much is known

(Cote et al., 2018; Croft et al., 2019; Ferreira-Pinto et al., 2018; Frigon, 2017; Wyart, 2018). Extending or modifying current understanding of how locomotion is produced is not the purpose of the studies discussed below. Moreover, their purpose is not to determine the contributions that the spinal pathway of the H-reflex or the corticospinal pathway of the MEP makes to the adaptive behaviours in which these pathways normally participate (e.g. by regulating muscle stiffness; Carter et al., 1990; Cronin et al., 2011; Nichols & Houk, 1976; Shadmehr & Arbib, 1992; Sinkjaer et al., 1988). Rather, the purpose of the studies discussed below is to determine how several adaptive behaviours share the ubiquitously plastic CNS. Their results enable us to evaluate the explanatory power of the heksor and negotiated equilibrium concepts, particularly their capacity to explain or predict results that would be otherwise inexplicable.

For this purpose, locomotion, the operantly conditioned H-reflex, and the operantly conditioned corticospinal MEP have important experimental advantages: (1) they are each produced by a distributed continually changing network of neurons and synapses (i.e. a heksor) that extends from cortex to spinal cord (e.g. Fig. 1*C*); (2) their heksors overlap each other in the spinal cord; and (3) this spinal cord overlap ensures that the impact of a new heksor on an old one will translate directly to readily measurable EMG and kinematic effects. These advantages enabled the studies discussed here to provide new insight into how these behaviours, and by implication other adaptive behaviours, share the ubiquitously plastic CNS. As previously noted, this problem is the focus of the present paper.

**The first impetus toward the heksor and negotiated equilibrium concepts: unexpected complexity.** The H-reflex, the electrical analogue of the spinal stretch reflex (e.g. the knee-jerk reflex), is elicited by weak electrical stimulation of the peripheral nerve and is produced mainly by a two-neuron monosynaptic pathway comprised of the Ia sensory afferent neuron, its synapse on the spinal motoneuron, and the motoneuron itself. As indicated in Fig. 3*A*, this pathway is influenced by descending activity from the brain; it is this descending activity that is shaped by the H-reflex operant conditioning protocol (for review, Thompson & Wolpaw, 2014; Wolpaw, 2018). As Fig. 1*C* shows, the distributed network, or heksor, that is created by the protocol and is responsible for the smaller or larger H-reflex, extends from cortical and subcortical areas to spinal cord (for review, Thompson & Wolpaw, 2014; Wolpaw, 2018). This plasticity is produced by operant conditioning of the H-reflex in one leg only. Its behavioural impact is similarly unilateral; the H-reflex in the other leg usually changes little or not at all (Wolpaw et al., 1993). Thus, judging

purely from the new behaviour, it appears that the other side of the spinal cord is largely unaffected. However, more comprehensive study reveals that the reality is very different: acquisition of this simple unilateral behaviour produces bilateral plasticity (Wolpaw & Lee, 1987, 1989). This surprising finding was the first of the data that continued to accumulate over the next several decades and led to the heksor and negotiated equilibrium concepts. As described below, these data show that the plasticity associated with acquisition of a new behaviour comprises more than the plasticity that produces the new behaviour.

As noted above, down-conditioning of the H-reflex in one leg does not change the H-reflex in the other leg – when it is measured in the awake behaving animal (Wolpaw & Lee, 1987, 1989; Wolpaw et al., 1993). Figure 3*B* shows this result and compares it to what is found when the animal is anaesthetized and the lumbosacral spinal cord is isolated from the brain by transection at the thoracic level. The reflex asymmetry created by conditioning persists after transection; this was the expectation at the time. But something else, a complete surprise at the time, is also apparent: the reflexes on both sides are much larger than those found in a naïve (i.e. unconditioned) animal under the same circumstances of anaesthesia and transection (indicated by the 100% line). Thus, even though the H-reflex is conditioned in only one leg, both sides of the spinal cord change.

This bilateral increase in the reflex on both sides of the isolated spinal cord was totally unexpected at the time. Now, it is readily explained by the heksor and negotiated equilibrium concepts. Findings of this kind – plasticity that does not contribute to the operantly conditioned decrease in the H-reflex – are in fact predicted by the concepts. Creation of the new H-reflex heksor on the trained side of the spinal cord affects behaviours, such as locomotion, that use both sides. Thus, the new heksor compels these old heksors to change in order to maintain the key features of their bilateral behaviours. As a result, while the contralateral H-reflex of the awake behaving animal does not change in size, it is produced differently. This becomes evident only when the spinal cord is studied in isolation. In terms of the new concepts, these changes result from negotiation among the new H-reflex heksor and the old heksors that underlie bilateral behaviours. Together, these heksors create a new negotiated equilibrium that incorporates the new H-reflex heksor and at the same time maintains the key features of the old heksors. This is reflected in the unchanged contralateral H-reflex found in the behaving animal. In the behaving animal, the brain and spinal changes resulting from the negotiation combine to produce a smaller H-reflex on the conditioned side and an unchanged H-reflex on the other side. When the brain's influence is removed, the newly modified spinal cord alone determines behaviour, and the picture shown

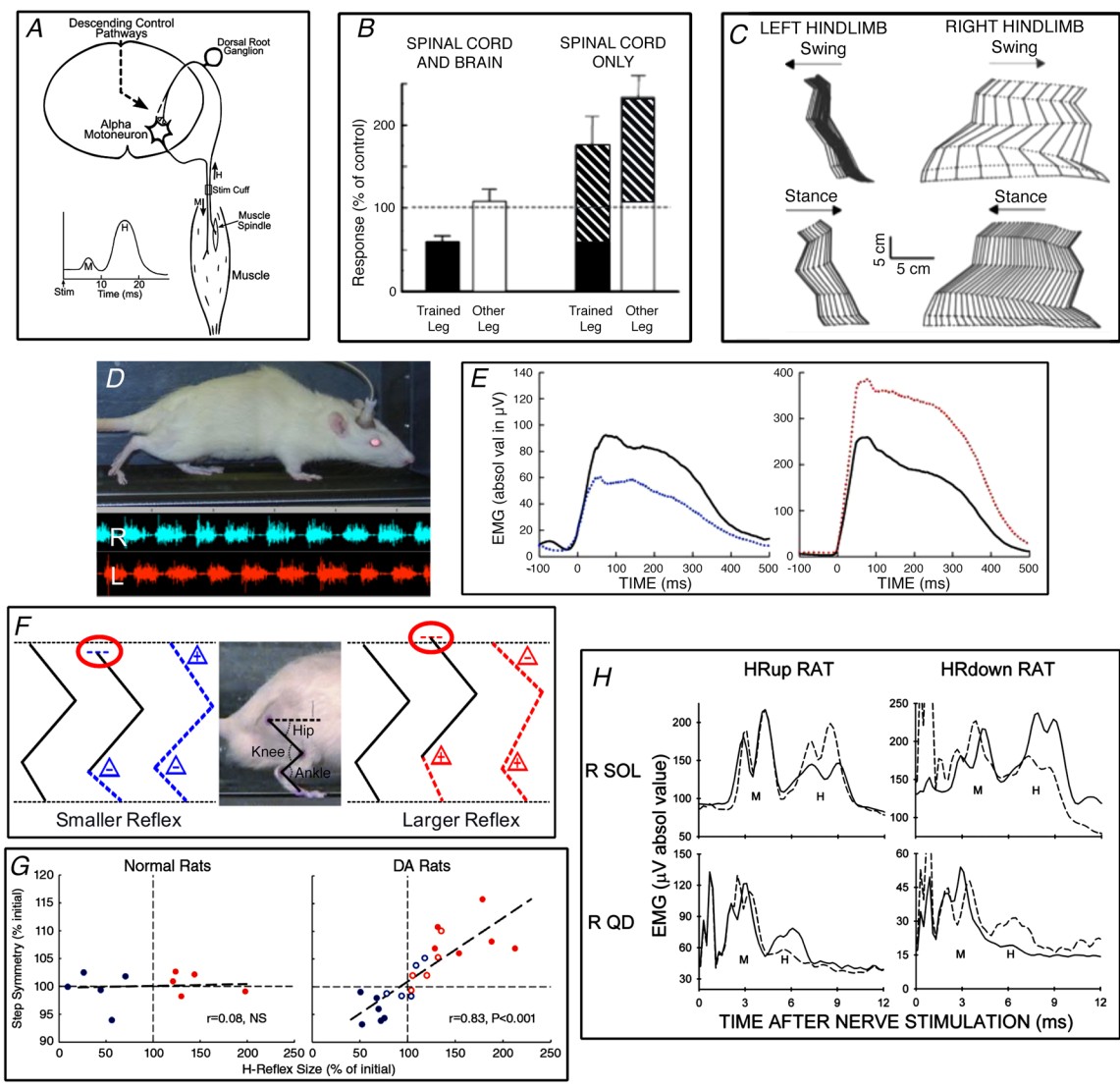

**Figure 3. Negotiation among heksors**

*A*, The H-reflex. The H-reflex is the electrical analogue of the spinal stretch reflex. It is elicited by electrical stimulation (via a nerve cuff in animals, or cutaneous electrodes in humans) of the largest (i.e. Group 1) afferents from the muscle. This afferent input excites the motoneuron monosynaptically (and probably oligosynaptically) and leads to muscle contraction. This wholly spinal pathway is affected by descending activity from the brain. The operant conditioning protocol modifies this descending activity to change H-reflex size so as to increase reward probability. The nerve-cuff stimulus also excites the largest efferents, which elicit a direct muscle response (M-wave) that precedes the H-reflex. The inset at lower left shows the timing of the M-wave and H-reflex in the rat soleus muscle. The computer automatically adjusts stimulus amplitude to maintain a target M-wave size over the weeks of H-reflex conditioning. This ensures that the effective strength of the stimulus (i.e. the numbers of efferents and afferents that it excites) remains stable. (See Wolpaw (1987) and Chen and Wolpaw (1995) for full methods presentation.) *B*, Early evidence that a new heksor changes an old one. Reflex responses from monkeys after H-reflex down-conditioning. Left: Average triceps surae H-reflexes of the awake behaving monkeys in the down-conditioned (trained) leg and the other leg. The H-reflex in the trained leg is much smaller than its control (i.e. original) value, while the H-reflex in the other leg is not changed from its control. Right: Average maximum reflex responses to dorsal root stimulation from the same monkeys under anaesthesia and after mid-thoracic spinal cord transection prior to euthanasia. The reflex asymmetry created by down-conditioning is still present, but the reflexes in both legs are much larger than the average reflexes in naïve (i.e. unconditioned) animals under the same circumstances of anaesthesia and spinal cord transection (indicated by the 100% line). The creation of the new H-reflex heksor does not change the H-reflex in the other leg as measured in the awake behaving animal; but it

does change the spinal neurons and synapses that combine with neurons and synapses in the brain to produce the unchanged H-reflex in the other leg. This becomes evident when the spinal cord is isolated from the brain. The hatched segments indicate the decreases produced by the brain. Stated simply, new spinal properties provide the appropriate reflex asymmetry (i.e. the reflex is smaller in the down-trained leg); new brain properties insure that the only effect of conditioning evident in the awake behaving animal is the smaller H-reflex in the trained leg. (Modified from Wolpaw (2018); data from Wolpaw and Lee (1987, 1989) and Wolpaw et al. (1993).) C, Nerve transection changes the locomotion heksor in both spinal cord and brain. Joint angles of left and right hindlimbs during swing and stance phases of clonidine-induced locomotion 72 days after spinalization in a cat in which left lateral gastrocnemius and soleus (LGS) muscles had been denervated 49 days before spinalization. During the 49 days before spinalization, treadmill training had enabled locomotion to recover from the impairment produced by the denervation. To prevent overlap in the display, individual stick figures are displaced by an amount equal to the displacement of the foot along the horizontal axis. Left and right hindlimbs are shown from left and right sides, respectively. Horizontal arrows indicate the direction of hindlimb movement during stance and swing phases. After spinal transection, the isolated spinal cord does not respond to locomotor training as would the isolated spinal cord of a cat that had not undergone previous left LGS denervation and recovery. Spinal locomotion is greatly impaired on the left side (i.e. the denervated side). This indicates that both the spinal cord and the brain components of the locomotion heksor had changed in the process that restored locomotion after the left LGS denervation. The result is that, after transection, the isolated spinal cord does not achieve spinal locomotion. (Reprinted from Frigon & Rossignol (2009), with permission from Elsevier.) D, A rat chronically implanted with soleus EMG electrodes walks on a treadmill. The traces show the right and left soleus locomotor EMG bursts that support the right and left stance phases of the step cycle. E and F, Soleus H-reflex conditioning affects how locomotion is produced. E, H-reflex conditioning affects locomotor EMG. Average absolute value of right soleus locomotor EMG bursts before (continuous black) and after (dotted blue or red) right soleus H-reflex conditioning in a down-conditioned rat (left, blue) and an up-conditioned rat (right, red). Because the H-reflex pathway contributes to the locomotor burst, the burst is smaller after down-conditioning and larger after up-conditioning. (Modified from Chen et al. (2005). Copyright (2005) Society for Neuroscience.) F, The EMG changes affect locomotor joint angles, but locomotion is not impaired. Centre: Right stance-phase anterior-ankle, posterior-knee, and anterior-hip joint angles as a rat walks on the treadmill. Left: Left to right, the first image shows the angles before down-conditioning of the right soleus H-reflex. The second shows that the weaker soleus burst (e.g. E, left-side) decreases ankle angle. This alone would lower the right hip and cause right–left asymmetry in hip height (i.e. the animal would tilt to the right). However, as the third image shows, a concurrent increase in hip angle makes up for the decrease in ankle angle, so that right hip height does not change. The joint angles change, but the key feature, right–left symmetry in hip height, is maintained. Right: Left to right, the first image shows the angles before up-conditioning of the right soleus H-reflex. The second shows that the stronger soleus burst (e.g. E, right-side) increases ankle angle. This alone would raise the right hip and cause right–left asymmetry in hip height (i.e. the animal would tilt to the left). However, as the third image shows, a concurrent decrease in hip angle makes up for the increase in ankle angle so that right hip height does not change. The joint angles change, but the key feature, right–left symmetry in hip height, is maintained. (Modified from Chen et al. (2011).) G, Sensory feedback guides plasticity in the locomotion heksor. Step symmetry *versus* final H-reflex size after up-conditioning (red) or down-conditioning (blue) for normal rats (left) and for DA rats (right), in which transection of the dorsal ascending (DA) tract has removed most proprioceptive sensory feedback. In normal rats, H-reflex up-conditioning or down-conditioning does not affect step symmetry (i.e. 100% indicates original (pre-conditioning) right–left ratio in step length). In contrast, in DA rats, conditioning creates asymmetries that correlate with its direction and magnitude. In the DA rats, the supraspinal component of the locomotion heksor lacks the sensory feedback it needs to guide changes that prevent the new H-reflex heksor from causing right–left asymmetry in step length. (From Chen et al. (2017), which provides full description.) H, Soleus H-reflex conditioning oppositely affects the quadriceps H-reflex. Average right soleus (R SOL) and right quadriceps (R QD) poststimulus EMG for a control day (continuous line) and for a day near the end of conditioning (dashed line) for an up-conditioned (HRup) rat and a down-conditioned (HRdown) rat. After up-conditioning the R SOL H-reflex is larger and the R QD H-reflex is smaller, while after down-conditioning the R SOL H-reflex is smaller and the R QD H-reflex is larger. Background EMG (EMG at time 0) and M-waves do not change. The effects on the R QD may result from the changes in the locomotion heksor that maintain the key features of locomotion despite the plasticity that creates the new H-reflex heksor (see text). Peaks in the first 1–2 ms after stimulation are stimulus artifacts. (From Chen et al. (2011), which provides full description.)

on the right in Fig. 3*B* emerges. This understanding of an initially inexplicable finding is a first indication of the explanatory power of the new concepts.

Subsequent studies in cats by Rossignol and colleagues provide another example of how brain and spinal changes combine to keep a behaviour unchanged. Carrier et al. (1997) compared the locomotor effect of spinal cord transection followed by unilateral denervation of ankle flexor muscles to the effect of unilateral denervation followed by spinal cord transection. When the sequence is:

$$\text{spinal transection} \rightarrow \text{treadmill training} \rightarrow$$
$$\text{flexor denervation} \rightarrow \text{more treadmill training,}$$

the spinal locomotion produced by the treadmill training recovers almost completely from the subsequent denervation. Greater hip and knee flexion make up for the

reduced ankle flexion caused by denervation. In contrast, when the sequence is:

fiexor denervation → treadmill training →

spinal transection → more treadmill training,

spinal locomotion does not develop as it would in a cat that has not undergone pre-transection denervation and training. The greater hip and knee flexion originally caused by denervation increase still further and other major abnormalities in muscle activations appear. In a subsequent study, Frigon & Rossignol (2009) extended this work. These studies showed that the deleterious effect of prior denervation on locomotor recovery after subsequent spinal transection reflects the fact that the initial recovery from the pre-transection denervation is due to plasticity in both spinal cord and brain. After the modified spinal cord is isolated by subsequent spinal transection, it does not respond to treadmill training as would a similarly isolated naïve spinal cord. Thus, locomotion remains very abnormal (i.e. Fig. 3*C*). As with the contralateral impact of unilateral H-reflex conditioning (Fig. 3*B*), the locomotor recovery after denervation alone is attributable to heksor modifications that change both brain and spinal cord. In both situations, these modifications become apparent only when the spinal cord operates in isolation. Once again, the new concepts account for unexpected results.

**The impact of a new behaviour on a pre-existing behaviour in health and disease.** Since the 1989 study that first revealed the startling bilateral impact of unilateral H-reflex conditioning (Wolpaw & Lee, 1989), a lengthy series of experiments have explored the impact of unilateral H-reflex conditioning on key features of locomotion and on locomotor EMG activity and kinematics. Because unilateral H-reflex conditioning has asymmetrical effects on the spinal cord, these studies focus on key features of locomotion that are related to right–left symmetry, specifically symmetry in step timing, step length, and hip height. Loss of these symmetries causes a limp and a tilt during walking. In these experiments, locomotion is assessed by measuring locomotor EMG activity and kinematics (e.g. joint angles, hip height, step length). One set of experiments studies the impact of unilateral H-reflex conditioning in intact animals. Another set studies its impact in animals and people whose locomotion is already asymmetrical due to an incomplete spinal cord injury (iSCI).

In an intact rat, creation of the new unilateral H-reflex heksor by the H-reflex operant conditioning protocol does not make locomotion asymmetrical. Right-left step and hip symmetries are maintained – the animals do not limp or tilt when they walk. However, the operant conditioning protocol does change locomotor EMG activity and joint angles (Chen et al., 2005, 2011). Figures 3*D–F* illustrate the results relevant to the maintenance of symmetry in hip height. As Fig. 3*E* shows, H-reflex conditioning affects locomotor EMG: the right soleus locomotor burst is larger after H-reflex up-conditioning and smaller after down-conditioning. This affects locomotor kinematics (Fig. 3*F*): the right ankle angle is larger (i.e. more plantarflexed) after H-reflex up-conditioning and smaller after down-conditioning. If this were the only kinematic effect, right hip height would change and the rat would tilt when it walked (i.e. to the left after up-conditioning and the right after down-conditioning). But this does not happen because the direct kinematic impact of the plasticity responsible for the new H-reflex behaviour – change in right ankle angle during locomotion due to the change in right soleus muscle activity – is balanced by an opposite change in right hip angle. The result is that hip height does not change, and the rat does not tilt when it walks.

In present terms, it appears that the new H-reflex heksor and the old locomotion heksor have negotiated a new equilibrium that satisfies both of them. The old heksor maintains its key feature of symmetry in hip height by modifying proximal locomotor muscle activity and kinematics on the conditioned (i.e. right) side to compensate for the unilateral change in strength of the soleus H-reflex pathway. That the features maintained include right–left symmetry in hip height and step length (Fig. 3*G*, left side) (rather than, for example, right–left symmetry in ankle angle changes during locomotion) makes sense in terms of what is normally considered satisfactory locomotion. More importantly, the right–left hip and step symmetries that are maintained are adaptive: they are likely to serve locomotor speed and stability, to increase metabolic efficiency, and to reduce musculoskeletal trauma. It is presumably these adaptive properties that guide the locomotor training experiences that define the key features of locomotion and create the locomotion heksor that maintains them.

Two subsequent studies add to understanding of these results and support their interpretation in heksor terms. One of these studies illuminates the process that guides the plasticity that preserves right–left locomotor symmetry (Chen et al., 2017). It shows that the heksor changes that preserve this symmetry are not produced by the spinal cord alone. These changes involve the brain; they require that the brain receive sensory input indicating the deleterious impact on symmetry of the plasticity responsible for the new unilateral H-reflex behaviour (Chen et al., 2017). In rats in which mid-thoracic transection of the dorsal ascending columns has abolished most proprioceptive sensory input to the supraspinal components of the old locomotion heksor, that heksor lacks the error-based guidance it needs to maintain its key features despite the creation of the new H-reflex heksor. Moreover, the new heksor does not need to change in

response to changes produced by the old locomotion heksor; the new and old heksors do not negotiate. The result is that the plasticity responsible for the new behaviour – the H-reflex heksor – does cause locomotor asymmetry. The rats now limp and tilt when they walk because the necessary negotiation has not occurred (e.g. Fig. 3*G*).

The second study appears to give some initial insight into the nature of the plasticity that preserves the locomotor symmetries (Chen et al., 2011). As illustrated in Fig. 3*H*, when the right soleus H-reflex is increased by up-conditioning, the right quadriceps H-reflex usually decreases; and when the right soleus H-reflex is decreased by down-conditioning, the right quadriceps H-reflex usually increases. The conditioning protocol does not base reward on the size of the quadriceps H-reflex. Thus, the consistent changes in quadriceps H-reflex size were a surprise. They may result from the changes in the locomotion heksor that maintain the key features of locomotion despite the plasticity that creates the new H-reflex heksor. If that is correct, soleus H-reflex conditioning should not change the quadriceps H-reflex in rats in which proprioceptive sensory input has been removed (i.e. the DA rats of Fig. 3*G*). That experiment is yet to be performed.

In sum, H-reflex conditioning does not impair the key features of locomotion in uninjured rats with sensory feedback to the brain intact; rather, it changes how these key features are achieved (Figs. 3*E* and *F*; Chen et al., 2005, 2011). In contrast, when an iSCI has already caused asymmetrical locomotion, the impact of H-reflex conditioning can be very different (Chen et al., 2006, 2014a, 2014b; Manella et al., 2013; Thompson & Wolpaw, 2021; Thompson et al., 2013). In animals and humans with asymmetrical locomotion, the impact of H-reflex conditioning is determined by whether the plasticity responsible for the new behaviour (i.e. the H-reflex heksor) is beneficial or detrimental for locomotion.

If the plasticity that changes H-reflex size (i.e. the new H-reflex heksor) restores locomotor symmetry, the existing locomotion heksor does not change to prevent this, and locomotor symmetry is restored (Chen et al., 2006; Thompson et al., 2013). Figures 4*A* and *B* illustrate this result in rats and in people. Here, the plasticity created by operant conditioning of the H-reflex in the appropriate direction restores locomotor symmetry. In rats, the deficit is weak stance, so H-reflex up-conditioning is appropriate because it increases the soleus locomotor burst. In humans, the impairment is spasticity, so down-conditioning is appropriate. In both species, appropriate H-reflex conditioning restores locomotor symmetry. When the new H-reflex heksor restores the key features of the old locomotion heksor, the old heksor accepts the change and locomotion improves.

The design of the human H-reflex conditioning protocol makes it possible to divide the total H-reflex change into a brain component (i.e. the change due to descending influence from the brain) and a spinal component (i.e. the change due to the spinal cord plasticity that this brain influence gradually produces over the course of the conditioning sessions) (Thompson et al., 2009). This analysis provides a striking illustration of the explanatory power of the new concepts (Thompson et al., 2013). The total H-reflex decrease produced by down-conditioning in people with iSCI was identical to that produced in people without iSCI (Thompson et al., 2009, 2013). However, as Fig. 4*C* shows, the two groups differed substantially (and significantly; $P < 0.01$) in the relative sizes of their brain and spinal components (Thompson et al., 2013). In people without iSCI, the brain and spinal components were nearly equal; each accounted for about 50% of the H-reflex decrease. In people with iSCI, the brain component of the decrease was very small (23%) and the spinal component was very large (77%). The small brain component was not surprising: iSCI presumably reduced the brain's influence. However, the large spinal component was surprising. Since the brain's influence produces and maintains the spinal component (Thompson & Wolpaw, 2014; Thompson et al., 2013; Wolpaw & Chen, 2006), the spinal component would be expected to be reduced as well in people with iSCI, and the total decrease would be considerably less than in people without iSCI. But this did not happen. The brain's influence was weaker than in people without iSCI, but the spinal cord plasticity associated with it was much greater.

This otherwise inexplicable result is consistent with the heksor and negotiated equilibrium concepts. As Fig. 1*C* shows, the new H-reflex heksor created by down-conditioning includes spinal neurons and synapses that are already parts of the locomotion heksor, and other old heksors as well. In people without iSCI, these old heksors have achieved a satisfactory negotiated equilibrium prior to H-reflex down-conditioning. By disturbing this equilibrium, the spinal plasticity produced by H-reflex down-conditioning is likely to impair old behaviours; their heksors will respond by changing so as to reduce this plasticity or counteract its functional impact. This adversarial negotiation between the new heksor and the old heksors shapes the new heksor so as to minimize its spinal component and maximize its brain component. In contrast, in people with iSCI, old heksors are not in a satisfactory negotiated equilibrium prior to H-reflex down-conditioning. Locomotion and other old behaviours are impaired. By reducing spasticity, the spinal plasticity produced by H-reflex down-conditioning improves the impaired behaviours; their heksors accept this beneficial plasticity and may even augment it through their own influence on the spinal cord (see below). This synergistic negotiation between the new heksor and the

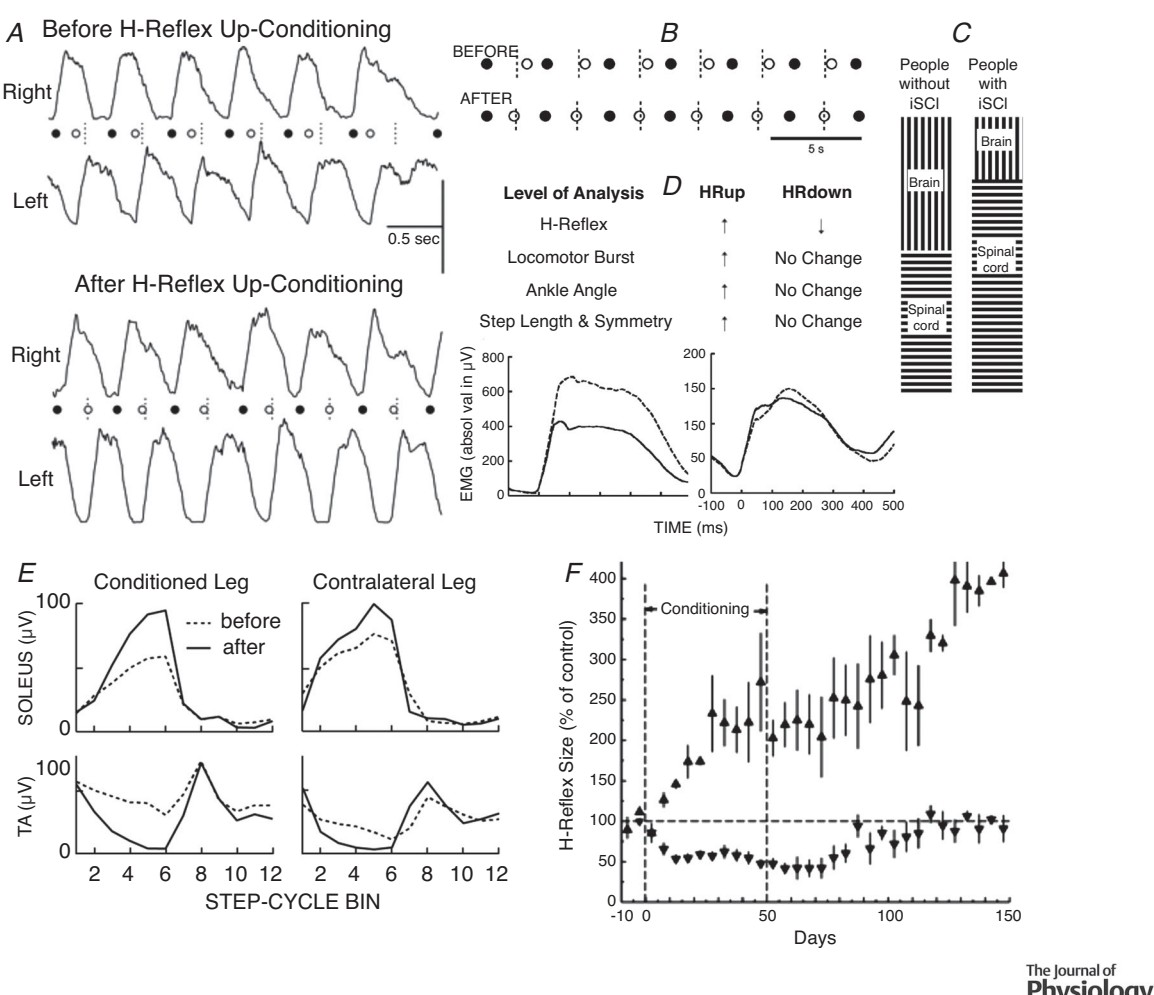

**Figure 4. A new heksor can help an old heksor that has been impaired by CNS trauma**

*A*, H-reflex up-conditioning can restore locomotor symmetry in a rat with weak stance. The traces show right and left soleus bursts (rectified EMG activity) during treadmill locomotion from a rat with an incomplete spinal cord injury, before and after up-conditioning of the right soleus H-reflex. Right and left soleus burst onsets (RBOs and LBOs) are indicated by ● and ○, respectively. Short vertical dashed lines mark the midpoints between RBOs, which is when LBOs should occur. Calibration: horizontal bar, 0.5 s; vertical bar, 100 and 150 μV for the right and left bursts, respectively. Before up-conditioning, the right burst is weak and brief; thus, LBO occurs too early and the rat limps. Up-conditioning of the right soleus H-reflex strengthens the right soleus burst so that LBO occurs on time. Right-left step symmetry is restored and the limp is gone. (From Chen et al. (2006). Copyright (2006) Society for Neuroscience.) *B*, H-reflex down-conditioning can restore locomotor symmetry in a person with spasticity. Right and left foot contacts (RFC and LFC; indicated by ● and ○, respectively) during treadmill locomotion in a person with an incomplete spinal cord injury, before and after down-conditioning of the right soleus H-reflex. The short vertical dashed lines mark the midpoints between RFCs, which is when LFCs should occur. Calibration: horizontal bar, 0.5 s. Before H-reflex down-conditioning, LFC occurs too late; the person limps. After down-conditioning, LFC occurs on time; right–left step symmetry is restored and the limp is gone. (From Thompson et al. (2013).) *C*, People with and without incomplete spinal cord injury (iSCI) differ in the relative sizes of the brain and spinal components of operantly conditioned H-reflex decrease. The total H-reflex decrease produced by down-conditioning in people with iSCI is identical to that produced in people without iSCI. However, the groups differ substantially (and significantly; $P < 0.01$) in the relative sizes of the brain and spinal components of this identical decrease. In people without iSCI, the brain and spinal components are nearly equal. In people with iSCI, the brain component is small (23%) and the spinal component is large (77%). The small brain component is not surprising: iSCI presumably reduces the brain's influence. The large spinal component is surprising. Since the brain's influence gradually produces the spinal component, the spinal component should also be reduced, and the total decrease should be less than in people without iSCI. But this does not happen: the brain influence is weaker than in people without iSCI, but the spinal cord plasticity it produces is much greater. This puzzling result is explained by the heksor and negotiated equilibrium concepts. (See text and Thompson et al. (2009, 2013) for full discussion.) *D*, An appropriate new heksor improves

an impaired old heksor; an inappropriate new heksor does not further impair an impaired old heksor. Top: Impact of up- or down-conditioning of the right soleus H-reflex in a rat with a weak right soleus burst and a limp due to iSCI. Up-conditioning (HRup) increases the right soleus H-reflex, the right soleus locomotor burst and the right ankle angle during locomotion; it also increases step length and restores right–left step symmetry so that the limp is gone (e.g. *A*, bottom). In contrast, down-conditioning (HRdown) decreases the right soleus H-reflex, but it does not affect the right soleus locomotor burst, the right ankle angle during locomotion, step length, or step symmetry. The limp is unchanged (e.g. *A*, top); it does not get worse. The new H-reflex heksor achieves its key feature; at the same time, the old locomotion heksor changes so as to prevent further impairment of its key features. Bottom: Average right soleus locomotor bursts in a spinal cord-injured rat before (continuous line) and after (dashed line) up-conditioning (left) or down-conditioning (right) of the right soleus H-reflex. In the up-conditioned rat, the soleus burst increases. As illustrated in *A*, this restores right–left step symmetry. In contrast, in the down-conditioned rat, the soleus burst does not change. It appears that the impaired locomotion heksor responds to the inappropriate new H-reflex heksor by changing so as to prevent decrease in the soleus locomotor burst (e.g. perhaps by providing additional properly-timed excitatory input to the motoneuron); thus, the right–left step asymmetry does not get worse. (Modified from Chen et al. (2014a)). *E*, An appropriate new heksor leads to widespread plasticity that further improves an old heksor. Rectified locomotor EMG activity in soleus and tibialis anterior (TA) muscles of both legs before (dashed line) and after (continuous line) soleus H-reflex down-conditioning in a person with impaired locomotion due to iSCI. The step cycle is divided into 12 equal bins, starting from foot contact; bins 1—7 are the stance phase and bins 8—12 are the swing phase. Down-conditioning of the hyperactive soleus H-reflex in one leg improves locomotor EMG activity in soleus and TA muscles of both legs: appropriate stance-phase activity increases in both solei and inappropriate stance-phase activity decreases in both TA. Walking speed and right–left symmetry improve. (Modified from Thompson et al. (2013).) *F*, An appropriate new heksor created by H-reflex conditioning initiates beneficial plasticity that continues to increase after conditioning ends. Mean (±SEM) H-reflex (HR) size for up-conditioned and down-conditioned rats with iSCI (▲ and ▼, respectively) for each 5-day period for the final 10 days before conditioning, the 50 days of up- or down-conditioning, and 100 days after conditioning ends. Up-conditioning strengthens right stance and restores step symmetry (*A*). After up-conditioning ends, the H-reflex continues to increase; this is accompanied by further improvement in locomotion (e.g. in step length). In contrast, down-conditioning does not increase the impairment; and after down-conditioning ends, the H-reflex decrease gradually disappears and locomotion remains just as impaired as it was before down-conditioning. (Modified from Chen et al. (2014b).)

old heksors shapes the new heksor so as to maximize its spinal component, despite its iSCI-impaired brain component. In sum, the component differences between people with and without iSCI (Fig. 4*C*) are consistent with the heksor and negotiated equilibrium concepts; the concepts explain an otherwise inexplicable result.

In rats with asymmetrical locomotion, it is possible to examine the impact of inappropriate H-reflex conditioning. As noted, in rats with iSCI the deficit is weak stance; thus, down-conditioning is not appropriate therapeutically because it is expected to further weaken stance and make locomotion even more asymmetrical. In fact, down-conditioning has remarkable results in these rats. As shown in Fig. 4*D*, down-conditioning does successfully reduce the H-reflex in iSCI rats. However, it does not worsen the already existing locomotor asymmetry (Chen et al., 2014a). The network of neurons and synapses responsible for locomotion (the locomotion heksor) appears to undergo changes that prevent the overlapping new heksor that is responsible for the smaller H-reflex from further impairing locomotor symmetry. As a result, the weaker H-reflex pathway produced by down-conditioning, which reduces the soleus locomotor burst in an intact rat (Fig. 3*E*), does not do so in an iSCI rat (Fig. 4*D*). The H-reflex gets smaller, but the locomotor burst stays the same. How might the locomotion heksor do this? One possibility is that this heksor changes so as to increase other excitatory input to the soleus motoneurons

during the stance phase of locomotion. However it is done, the outcome is that the behavioural impact of the new heksor serves the key features of both the new and old heksors: the H-reflex is smaller as the conditioning requires, but locomotion is not further impaired. Once again, a surprising result is consistent with the heksor and negotiated equilibrium concepts.

In people with iSCI, appropriate H-reflex conditioning has remarkably widespread and lasting effects that are consistent with the heksor and negotiated equilibrium concepts. Several groups have shown that the down-conditioning protocol triggers bilateral plasticity that improves locomotor muscle activity in both legs (Manella et al., 2013; Thompson & Wolpaw, 2021; Thompson et al., 2013). It appears that, by weakening the hyperactive pathway responsible for the H-reflex and thereby eliminating locomotor impediments such as clonus or footdrop, the new H-reflex heksor enables more effective locomotor practice and thus more effective negotiation among the heksors. This leads to bilateral plasticity that improves other aspects of locomotion; it restores locomotor symmetry, eliminates limping (e.g. Fig. 4*B*), and increases walking speed. In these individuals with iSCI, the plasticity responsible for the new adaptive behaviour, a smaller H-reflex, is just a minor part of the total plasticity associated with acquisition of the new behaviour. Most of this plasticity comprises the widespread changes that the smaller H-reflex triggers

in the network of neurons and synapses responsible for the old behaviour of locomotion (i.e. the locomotion heksor). For example, the increase in the right soleus burst seen in Fig. 4*E* is clearly not a direct effect of down-conditioning the right soleus H-reflex: the direct effect of down-conditioning would be a decrease in the right soleus burst (Fig. 3*E*). Nevertheless, this increase in the right soleus burst and the corresponding increase in the left soleus burst are readily explicable as indirect effects of H-reflex down-conditioning: by eliminating the locomotor impediments due to the hyperactive reflex pathway, the new H-reflex heksor enables the old locomotion heksor to repair itself. The result is overall improvement in locomotion.

Figure 4*F* illustrates another remarkable aspect of the impact of appropriate H-reflex conditioning after an iSCI, an aspect that is consistent with the new concepts and provides insight into the properties of heksors. As noted above, in rats with iSCI the deficit is weak stance, and thus H-reflex up-conditioning is appropriate. The remarkable finding is that the H-reflex continues to increase after the up-conditioning protocol ends (Fig. 4*F*). This is not what occurs in intact animals. When up-conditioning (or down-conditioning) ends in intact animals, the H-reflex stops increasing (or decreasing) and gradually returns toward its original size (as it also does in the down-conditioned rats in Fig. 4*F*). Why does it keep increasing in iSCI rats after the end of the 50 days of up-conditioning, when a larger H-reflex is no longer providing a food reward? In the present context, the simplest explanation appears to be that the locomotion heksor drives the further increase. During H-reflex up-conditioning, the new H-reflex heksor drives the increase; this benefits the locomotion heksor by improving locomotor symmetry. After up-conditioning ends, it is the locomotion heksor that drives the increase, thus further restoring the key features of locomotion (Chen et al., 2014b).

If this explanation is correct, it raises a further question: if the locomotion heksor increases the H-reflex after the end of up-conditioning, why did it not increase the H-reflex before up-conditioning began? Given that the plasticity that increases the H-reflex improves locomotion in iSCI rats, why did the H-reflex not increase in iSCI rats that were never exposed to H-reflex up-conditioning (Chen et al., 2006)? In other words, why was the up-conditioning protocol necessary for the H-reflex to increase? The fact that it was necessary implies that the up-conditioning protocol modified the way in which the locomotion heksor changed itself in order to restore its key features: the protocol guided the locomotion heksor to make the changes that increase the H-reflex, and the locomotion heksor kept doing that after the end of the up-conditioning protocol. In short, it appears that the locomotion heksor learned from the beneficial

locomotor effects of the up-conditioning protocol. This implication recalls Bernstein's description of behaviours as 'biodynamical structures [that] live and develop' (Bernstein, 1967). It also supports the choice of the *or* suffix in the word heksor, which indicates that a heksor is an active agent.

**Beyond the H-reflex heksor.** The studies described so far evaluate the response of one old heksor (the locomotion heksor) to one new heksor (a soleus H-reflex heksor) in people and animals with or without an iSCI that impaired locomotion. More recently, studies in people with or without impaired locomotion have begun to explore the response of the locomotion heksor to a new corticospinal MEP heksor (Thompson & Sinkjær, 2020; Thompson et al., 2018a, 2018b, 2019). This new work begins to assess the wider applicability of the heksor and negotiated equilibrium concepts.

In these studies, the corticospinal MEP of the tibialis anterior (TA) muscle in one leg is up-conditioned by a protocol similar to that used to increase or decrease the soleus H-reflex (three 1–h sessions/week over 8–10 weeks) (Thompson & Sinkjær, 2020; Thompson et al., 2018a, 2018b, 2019). The MEP is produced by TMS. Control participants complete a protocol that is identical, except that the TA MEP is simply elicited without feedback as to whether its size has satisfied a criterion.

The results of these MEP conditioning studies parallel those of the H-reflex conditioning studies in supporting the heksor and negotiated equilibrium concepts. First, in people with or without impaired locomotion, exposure to the control protocol does not change MEP size or locomotion. Second, most people with or without impaired locomotion (due to iSCI or multiple sclerosis) can gradually increase MEP size in response to the up-conditioning protocol. Third, in people with intact locomotion, unilateral MEP up-conditioning does not impair locomotion. Fourth, in people with impaired locomotion, up-conditioning of the TA MEP in one leg improves locomotor EMG and kinematics in multiple muscles in both legs, and thereby improves walking speed. Fifth, these improvements persist after MEP conditioning ends. Figure 5 illustrates these findings. They are completely consistent with the H-reflex conditioning studies illustrated in Figs. 3 and 4.

In all these ways, the response of the locomotion heksor to a new unilateral MEP heksor is comparable to its response to a new unilateral H-reflex heksor. Both sets of studies confirm predictions based on the heksor and negotiated equilibrium concepts. If locomotion is normal, the new heksor does not impair it. If locomotion is abnormal, an appropriate new heksor improves it; this improvement reflects beneficial changes in the locomotor activity of muscles on both sides, and the improved

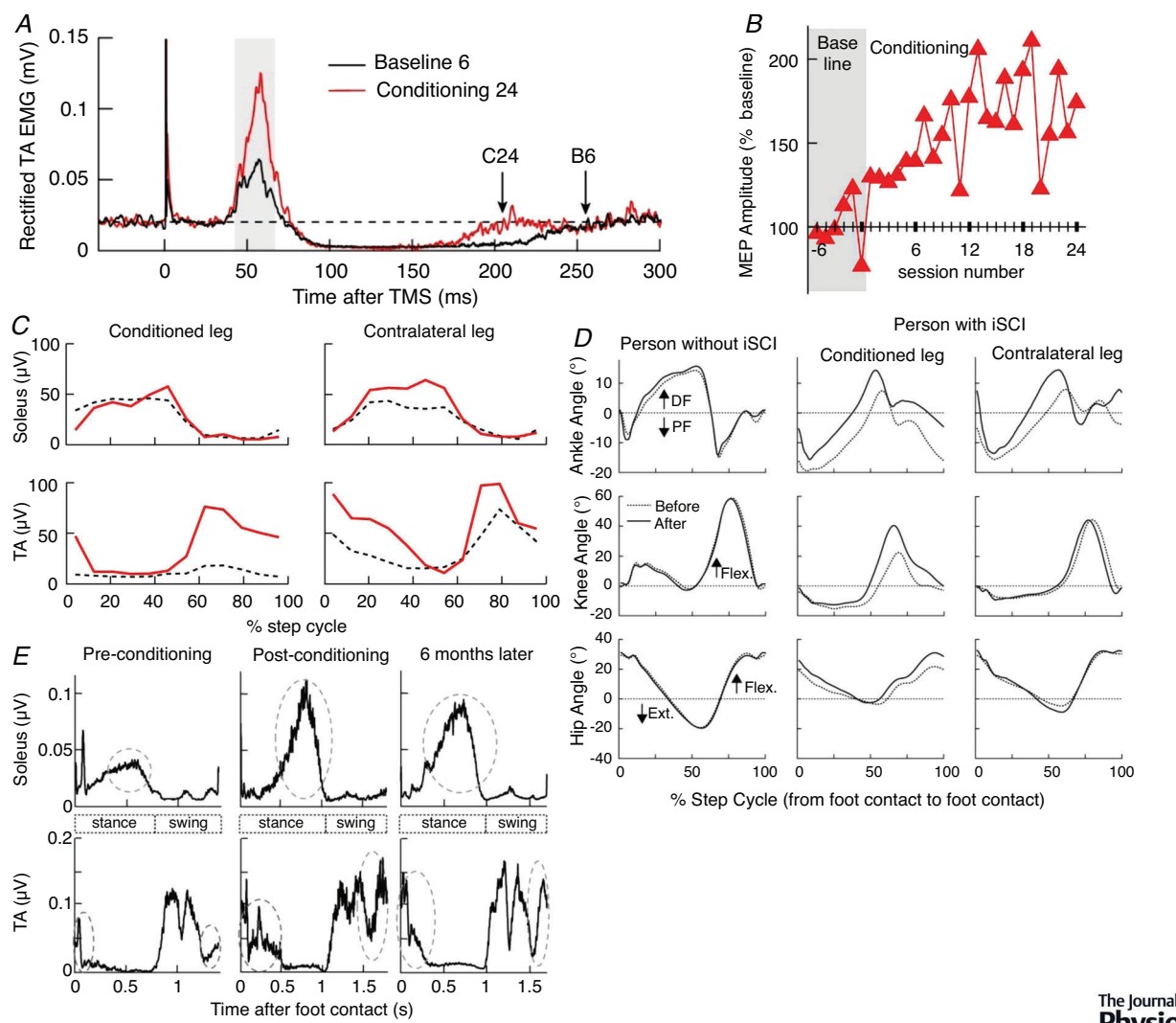

**Figure 5. Operant conditioning of the tibialis anterior (TA) motor evoked potential (MEP) evoked by transcortical magnetic stimulation (TMS) in people with or without impaired locomotion**

*A* and *B*, Up-conditioning of the right TA MEP. *A*, In a person with impaired locomotion due to chronic incomplete spinal cord injury (iSCI), the MEP (shaded area) is much larger after up-conditioning (conditioning session 24, red) than before up-conditioning (baseline session 6, black). In addition, the subsequent silent period is shorter. (From Thompson et al. (2018b).) *B*, In a person with locomotion impaired by multiple sclerosis (MS), up-conditioning gradually increases the TA MEP over three 1-h sessions/week for 8 weeks. (From Thompson and Sinkjær (2020).) *C*, Up-conditioning of the TA MEP in one leg improves TA and soleus locomotor EMG in both legs. Rectified locomotor EMG activity in soleus and TA bilaterally before (dashed black line) and after (continuous red line) up-conditioning of the TA MEP in one leg in a person with MS. The step cycle, from foot contact (onset of the stance phase) to foot contact (end of the swing phase), is divided into 12 equal bins. After TA MEP up-conditioning, the swing-phase TA burst is increased in the conditioned leg and in the contralateral leg as well. As a result, swing-phase ankle dorsiflexion returns in the conditioned leg and improves in the other leg, and foot-drop is ameliorated. Additional beneficial plasticity improves the stance-phase soleus burst in the contralateral leg. (From Thompson and Sinkjær (2020).) *D*, Up-conditioning of the TA MEP in one leg can improve locomotor kinematics in both legs. Ankle, knee, and hip joint angles over the step cycle before (dashed line) and after (continuous line) operantly conditioned increase in the TA MEP in representative individuals with or without iSCI. (ankle angle: + dorsiflexion (DF), − plantarflexion (PF); knee or hip angle: + flexion (Flex.), − extension (Ext.). In a person without iSCI (left), TA MEP up-conditioning does not change rotation at any of the three joints. In contrast, in a person with iSCI, TA MEP up-conditioning reduces footdrop by increasing ankle and knee dorsiflexion (DF) throughout the step-cycle in the conditioned leg (centre); it also increases ankle angle at foot contact and at peak dorsiflexion in the contralateral leg (right). (From Thompson et al. (2019).) *E*, Improved locomotion persists after TA MEP up-conditioning ends. Rectified soleus and TA locomotor EMG in the conditioned leg over the step-cycle (foot contact to foot contact) before and after TA MEP up-conditioning in a person with iSCI. TA MEP up-conditioning

greatly increases swing-phase TA EMG activity immediately before and during foot contact; this activity prevents footdrop (dashed ovals). TA MEP conditioning is also associated with marked increase in the soleus burst (dashed ovals) during the mid–late stance phase; this enhances push-off. These improvements in locomotor EMG activity are still present 6 months later. (From Thompson et al. (2019).)

locomotion persists after conditioning ends. The animal H-reflex studies also confirm the prediction that abnormal locomotion is not further impaired by an inappropriate new heksor (i.e. a heksor that could further impair locomotion). Animal studies that test this prediction for an inappropriate new MEP heksor have yet to be done.

The spinal cord's accessibility, simplicity and role as the final common pathway have facilitated these ongoing animal and human studies of the impact of a new behaviour on an old behaviour that is either intact or impaired. Their detailed results are wholly consistent with and thereby support the heksor and negotiated equilibrium concepts. Furthermore, the concepts explain surprising aspects of the results, aspects that would be inexplicable without the concepts. The applicability of these concepts to other adaptive behaviours is supported by imaging studies such as those illustrated in Figs. 1*B* and 2*B–D*. More extensive and detailed testing of their wider applicability is needed to determine the validity and significance of these new concepts and the new paradigm that they embody. This is the focus of the final section.

### Evaluating the wider applicability of the new concepts

The first piece of evidence leading to the heksor and negotiated equilibrium concepts was the surprising observation that some of the changes associated with acquisition of even the simplest adaptive behaviour – a larger or smaller soleus H-reflex in one leg – seemed to have nothing to do with the new behaviour (e.g. change on the other side of the spinal cord; Fig. 3*B*; Wolpaw & Lee, 1989). Current evidence for the validity of the concepts consists mainly of studies showing that they can predict such unexpected results.

The concepts predict that the plasticity associated with acquisition of a new behaviour will comprise the new plasticity constituting the new heksor that produces the new behaviour, *plus* the plasticity in pre-existing (i.e. old) heksors that maintains their key features despite the new plasticity in neurons and synapses that they share with the new heksor. From the perspective of an investigator studying acquisition of the new behaviour, the changes creating the new heksor can be called primary plasticity, and the changes in old heksors can be called secondary plasticity. Furthermore, in some situations the secondary plasticity is far more extensive than the primary plasticity. This can occur when an appropriate new heksor enables an old heksor to improve an old behaviour that has been impaired by injury or disease (e.g. Figs. 4 and 5).

The phenomenon of secondary plasticity associated with creation of a new heksor is both similar to and different from the phenomenon of diaschisis – the remote often harmful effects of a focal CNS lesion on distant CNS regions (e.g. the effects of a subcortical lesion on cortical areas) (Carrera & Tononi, 2014; Price et al., 2001). The two phenomena are similar in that they both reflect the wider impact of an event that directly affects specific neurons and synapses. For secondary plasticity, the specific neurons and synapses directly affected are those that are incorporated into the new heksor; for diaschisis, they are those located in the CNS area that is lesioned. The two phenomena differ in what their wider impact affects. Secondary plasticity affects old heksors that include neurons and synapses that are incorporated into the new heksor. Diaschisis affects CNS areas that are connected to the lesioned area. Both phenomena are likely to occur in the damaged CNS; thus, diaschisis will be an important consideration in developing new therapies based on the heksor and negotiated equilibrium concepts.

For the present purpose of evaluating the applicability of the new concepts to other adaptive behaviours, the principle is that the CNS plasticity associated with acquisition of a new behaviour should be explicable in terms of its role in producing the new behaviour (primary plasticity) and/or in maintaining old behaviours (secondary plasticity). Data that confirm this principle support the heksor and negotiated equilibrium concepts; data that do not do so call the concepts into question. This section considers how this principle could be tested with other kinds of adaptive behaviours; and it introduces several additional ways in which the wider applicability of the new concepts could be assessed. Furthermore, it considers the potential reach of the new concepts by discussing their ability to suggest new answers to important questions in neuroscience.

**Predicting primary and secondary plasticity.** The heksor and negotiated equilibrium concepts predict that the secondary plasticity associated with creation of a new H-reflex or MEP heksor will depend on how the new heksor (the primary plasticity) affects locomotion. As the studies described above show, this prediction proved correct. When locomotion was intact, and the new heksor would impair it, secondary plasticity in the old locomotion heksor prevented this impairment and thereby maintained the key features of locomotion (e.g. right–left symmetry in hip height; Figs. 3*F* and 5*D*). When locomotion was already impaired and the new

heksor improved it, the old heksor accepted the change and underwent secondary plasticity that further improved locomotor EMG and kinematics bilaterally, and thus further improved locomotion (Figs. 4*A*–*F* and 5*C*–*E*). Moreover, when locomotion was already abnormal and the new heksor would worsen it, the old heksor changed to prevent this further impairment (Fig. 4*D*). In all these situations, the secondary plasticity served the key features of the old heksor and accommodated the new heksor.

These clear results confirm the prediction that the plasticity associated with acquisition of a new behaviour will be explicable in terms of its role in producing the new behaviour (primary plasticity) and/or in maintaining old behaviours (secondary plasticity). Their clarity is due in large part to the spinal cord's accessibility, simplicity, and role as the final common pathway. A variety of other adaptive behaviours have these same advantages. Thus, they are logical initial targets for studies of the wider applicability of the heksor and negotiated equilibrium concepts. Promising candidates are sports in which the two legs and/or the two arms perform very different movements. These include high-jumping, pole-vaulting, kicking in American football, shot-putting, pitching in baseball, bowling in cricket, and throwing the discus (illustrated in Fig. 2*A*). These sports are essentially natural experiments that can reveal the impact of a lateralized new heksor on old bilaterally symmetrical heksors such as those responsible for locomotion and flexion-withdrawal.

The plasticity underlying mastery of these asymmetrical sports will include changes from cortex to spinal cord (e.g. Figs. 1*B* and *C*). Because right and left extremities have different tasks, the two sides of the CNS will change in different ways. Nevertheless, all these athletes continue to walk satisfactorily and retain satisfactory flexion-withdrawal reflexes. In terms of the heksor and negotiated equilibrium concepts, the highly lateralized new heksor and the old locomotion and flexion-withdrawal heksors negotiate an equilibrium satisfactory to them all (e.g. Fig. 2*A*). The new concepts predict that the responsible plasticity comprises the primary plasticity that constitutes the new heksor plus secondary plasticity in the old heksors that maintains their behaviours. This prediction could be tested by measuring EMG activity, kinematics, spinal reflexes, MEPs, somatosensory evoked potentials (SEPs), and electroencephalographic (EEG) sensorimotor rhythms during the old and new behaviours and at rest. The prediction is that these measures will display right–left differences that serve the new behaviour and/or maintain the old. For example, in an athlete learning to throw the discus (Fig. 2*A*), the heksor created by training is likely to include right–left asymmetries in the spinal reflex pathways and corticospinal connections of specific muscles – asymmetries that contribute to performance (i.e. to achievement of the key features of the skill,

most notably throwing direction and distance). This asymmetrical primary plasticity is likely to require the old flexion-withdrawal and locomotion heksors to respond with asymmetrical secondary plasticity in the same and/or other spinal and corticospinal pathways, plasticity that produces right–left asymmetries in the combinations and sequencing of the muscle activations that ensure the maintenance of their key features. As the sport is mastered, the EMG activity and kinematics of the movement that withdraws the foot from a nail, or the index finger from a hot stove, may come to differ for right and left extremities; mirror-image EMG and kinematics present before mastery of the asymmetric sport may well disappear. Nevertheless, the speed of withdrawal will remain satisfactory (or perhaps might even improve).

In these situations, the equilibrium negotiated between the new and old heksors is likely to differ across individuals: first, because their nervous systems and their bodies differ before the athletic training; second, because their training methods and its effects also differ; third, because CNS complexity and redundancy offer the heksors a variety of satisfactory negotiated equilibria. For example, people may differ in the distribution of plasticity between spinal and supraspinal components of the heksors. Spinal plasticity may be reflected in H-reflexes and/or other spinal reflexes, and supraspinal plasticity in MEPs, SEPs, and/or sensorimotor rhythms. Thus, while overall differences across different sports and between athletes and non-athletes may be apparent (e.g. Nielsen et al., 1993), each person will also be a case study. As a result, studies that focus only on group differences may miss important insights (for discussion, Makihara et al., 2014).

Behaviours not strongly tied to the spinal cord should also be amenable to studies that test predictions concerning primary and secondary plasticity. Figures 2*C* and *D* illustrate the impact of a second language on the heksor of a first language. Creation of the new heksor produces secondary plasticity in the old heksor; the old heksor adds new regions, increases activity in regions that were already part of it, and makes new and/or stronger inter-regional connections. The heksor and negotiated equilibrium concepts predict that these additions serve to maintain the key features of the old heksor. Thus, they should be most active when the new heksor interferes with the old. They should be particularly engaged by situations in which the two languages are very similar or very different, such as using or under-standing words in the two languages that sound alike but differ in meaning (or sound different but have the same meaning), parsing language-specific sentence structures, ensuring language-specific pronunciation and interpretation of pronunciation, or interpreting or using language-specific inflections. The prediction that the changes in the old heksor caused by creation of the new

one will be particularly engaged by such situations could be tested by assessing EEG, magnetoencephalographic, and fMRI ongoing activity and responses to specific words or images, and the effects of region-specific stimulation (e.g. TMS, transcranial direct current stimulation). The analyses could extend to subcortical as well as cortical components of heksors (e.g. Sathyamurthy et al., 2020). These methods could test the applicability of the new concepts to overtly cognitive behaviours.

**Detecting heksors in action.** While adaptive behaviours such as locomotion or language occur frequently, others such as flexion-withdrawal reflexes or riding a bicycle may occur infrequently or even rarely. Nevertheless, all these behaviours are maintained through life despite ongoing changes due to acquisition of new behaviours, as well as to growth, ageing, and other life events. This long-term maintenance of adaptive behaviours suggests that heksors are continually active and that their negotiations are ongoing, even in the absence of their behaviours. Ongoing heksor interaction is consistent with the long-recognized superiority of spaced practice over massed practice (e.g. Dempster, 1989; Sisti et al., 2007). This superiority suggests that negotiation between the newly developing heksor and old heksors occurs between as well as during practice periods. For H-reflex operant conditioning specifically, ongoing negotiation is consistent with the lack of correlation across animals between the number of reflex conditioning trials/day and the magnitude of reflex change. Ongoing negotiation is also consistent with the fact that humans, who perform only 3–5% as many conditioning trials as animals, but distributed over similar time periods, change the reflex almost as much (Thompson et al., 2009; Wolpaw et al., 1993).

These considerations support the hypothesis that heksor interactions underlie the growing evidence that ostensibly spontaneous CNS activity contributes to learning (e.g. Albert et al., 2009; Gulati et al., 2017; Litwin-Kumar and Doiron, 2014). This novel hypothesis about the nature of spontaneous CNS activity could be tested: ongoing heksor interactions should be detectable. The size, accessibility and extensive plasticity of sensori-motor cortex make it a particularly appealing region for study (Adkins et al., 2006; Francis & Song, 2011; Papale & Hooks, 2018; Peters et al., 2017; Sur et al., 2013). Recent success in using electrocorticography to track the step-by-step progress of a single trial of a simple reaction-time behaviour through cortex from primary sensory to primary motor areas (Paraskevopoulou et al., 2021) suggests that, with further development, similar methods might detect the ongoing activity of the heksor underlying such a behaviour. For example, the heksors that produce reaction-time behaviours that differ in

their ipsi *vs*. contra combinations of sensory input and motor output (e.g. which visual field receives the 'go' signal, and which hand responds) might be detected as ongoing synchronizations of activity between the specific ipsilateral *vs*. contralateral cortical areas responsible for the behaviours (e.g. Potes et al., 2014). By creating two such heksors that differ in their ipsi/contra combinations, it might also be possible to observe their negotiation and link it to the development of their behaviours. A video game format could facilitate the design and enhance the implementation of these studies, and could increase the intensity and duration of each participant's commitment to creating the new heksors.

Animal studies with methods such as two-photon calcium imaging could examine putative heksor interactions in cortex on neuronal and microcircuit levels (e.g. Komiyama et al., 2010). These studies might benefit from neural network-based models that suggest how multiple adaptive behaviours might be acquired and maintained in a continually plastic CNS (e.g. Ajemian et al., 2013). If these studies do succeed in detecting and charting an individual heksor, they may enable spatially and/or temporally targeted interventions (e.g. electrical or optogenetic) that manipulate specific aspects of the heksor to change the behaviour it produces. Such interventions could illuminate the mechanisms that define and maintain the key features of a behaviour.

**Testing the hypothesis that heksor negotiations create muscle synergies.** The aetiology of muscle synergies – the stereotyped combinations of muscle activations that are thought to be the building blocks of complex motor behaviours – is uncertain (Berger et al., 2013; Bizzi et al., 2008; Pierrot-Deseilligny & Burke, 2012). The new paradigm introduced in this paper suggests that they may result from long-term negotiation among heksors. This hypothesis could be tested by examining the effects of acquiring a new behaviour that conflicts with an established synergy. For example, the new behaviour could require reciprocal contractions of two muscles that normally act together in an existing synergy. In the process of mastering this new behaviour, existing synergies might change or disappear and new synergies might arise. If negotiation between the new heksor and old heksors is responsible, these changes should serve the new behaviour (primary plasticity) or maintain the key features of old behaviours (e.g. locomotion) (secondary plasticity). Furthermore, evidence that beta (13–30 Hz) activity in EEG over sensorimotor cortex underlies and reflects muscle synergies (Aumann & Prut, 2015) suggests that it might be possible to detect the initial and ongoing interactions among the new and old heksors. These studies might also incorporate parallel analyses of the spinal motoneuron synergies recently hypothesized by

Hug et al. (2021). Once again, a video game format in which the participants control EMG activity in specific muscles (or EMG produced by specific groups of motoneurons) could be useful.

Results consistent with the heksor hypothesis would support the recent proposal that muscle synergies are not fixed patterns, but are instead habits that can adjust to changing sensorimotor demands (Loeb, 2021) (e.g. Sawers et al., 2015). Further work that uses spatially or temporally targeted electrical or magnetic interventions to modify synergies might clarify how key features of a behaviour are maintained and how a key feature might be modified, replaced, or simply eliminated.

**Testing the hypothesis that heksors participate in homeostatic plasticity.** Homeostatic plasticity is a descriptive term for numerous subcellular processes that operate at neuronal and network levels to maintain neuronal excitability in an effective operating range (Wefelmeyer et al., 2016). Heksors have the same objective – they need to ensure that the excitabilities of their neurons support the satisfactory production of their behaviours. Thus, they may have a role in homeostatic plasticity.

It should be possible to test the hypothesis that heksors participate in homeostatic plasticity. For example, Jamann et al. (2021) found that exposing mice to an enriched environment that increased sensory input led to a reduction in the excitability of pyramidal neurons in primary sensory cortex that received input from whiskers; and they linked the reduction to specific changes in the axon initial segment. This study could be repeated with inclusion of an additional experimental group that had previously acquired a food-acquisition behaviour in which food was contingent on choosing correctly among several very weak whisker stimuli. The drop in neuronal excitability produced by the increased stimulation of the enriched environment would likely impair the ability to distinguish among these weak stimuli. Thus, the heksor that produces the food-acquisition behaviour would be expected to respond by reducing, focusing or otherwise modifying the drop in excitability so that it did not impair the acquisition of food. As a result, the homeostatic plasticity found in these mice would differ in neuronal distribution and/or in other respects from that found in mice that had not previously acquired the food-acquisition heksor; it might also differ mechanistically.

Confirmation of this expectation, and similar results from other studies of this kind, would imply that homeostatic plasticity is to some degree a by-product of the ongoing interactions through which heksors establish and maintain a negotiated equilibrium of neuronal and synaptic properties that enables each one to maintain the key features of its behaviour.

**Using brain–computer interface-based experiments to study heksor interactions.** Brain–computer interface (BCI) experiments can make neurons in sensorimotor cortex into the final common pathway for behaviours; they can give these cortical neurons the role normally filled by spinal motoneurons (e.g. Oby et al., 2019). Thus, BCI studies could characterize putative heksor interactions in cortex and compare them to the spinal-level interactions described with the H-reflex model. BCI-based models, while not perfect surrogates for natural behaviours, avoid the complications of the musculoskeletal apparatus and its complex sensory feedback. Insights from BCI-based models that incorporate cortical neurons may well be more relevant to the CNS than those gleaned from neural network models (Hennig et al., 2021). For example, BCI-based models could compare the impact on an old heksor of a new heksor that lies entirely within the current intrinsic manifold (i.e. already available combinations of active single neurons) to the impact of a new heksor that requires combinations not within the manifold (Oby et al., 2019). In the former case, the old heksor could move to a different place in the existing manifold; in the latter, the manifold itself would be modified and the old heksor might not remain within the confines of the original manifold.

This approach would parallel that suggested above for muscle synergies. In both cases, the focus would be on determining whether the plasticity that is not part of the new heksor is explicable as secondary plasticity in old heksors that maintains the key features of their behaviours despite the primary plasticity comprising the new heksor. Furthermore, BCI-based models might also identify in overtly spontaneous activity interactions among heksors that are currently not producing their behaviours.

**Therapeutic applications.** As Figs. 4 and 5 illustrate, much of the existing evidence that supports the new concepts comprises animal and human studies showing that an appropriate new heksor can enable an impaired old heksor to restore its key features. These exciting early results are driving further efforts to apply the new concepts to the development of new therapies. This is already a far too complex endeavour – both in theory and in practice – to be substantively addressed here. Nevertheless, it is clear that the ultimate fate of the heksor and negotiated equilibrium concepts rests in considerable part on their ability to guide design and implementation of new therapies that can enhance functional recovery for people with spinal cord or brain injury, stroke, cerebral palsy, and other chronic neuromuscular disorders.

## Conclusion

The heksor and negotiated equilibrium concepts respond to the question asked at the beginning of this paper: how

do numerous adaptive behaviours share a ubiquitously plastic CNS? The heksor concept describes an entity with the unique properties needed to maintain an adaptive behaviour despite ongoing plasticity. The negotiated equilibrium concept describes the CNS state that the concurrent actions of all the heksors produce, a state in which every heksor is able to maintain the key features of its adaptive behaviour. Together the two concepts comprise a new paradigm that can explain how adaptive behaviours are acquired and maintained through life in a continuously plastic CNS.

At present, the strongest support for the concepts and the paradigm comes from studies of interactions among several relatively simple behaviours in the healthy CNS and in the damaged CNS. Similarly detailed studies of their applicability to other adaptive behaviours are essential, and are possible with present methods. Many questions concerning the properties of heksors and heksor interactions need exploration. The promising therapeutic implications and applications of the concepts invite attention. Their ability to advance understanding of major issues in neuroscience also warrants evaluation. These studies will test the scientific validity and clinical usefulness of the new word *heksor* and, in doing so, they will assess the adequacy of the new paradigm based on heksors and the negotiated equilibrium that they create.

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

## Additional information

### Competing interests

None.

### Author contributions

Both authors have read and approved the final version of this manuscript and agree to be accountable for all aspects of the work in ensuring that questions related to the accuracy or integrity of any part of the work are appropriately investigated and resolved. All persons designated as authors qualify for authorship, and all those who qualify for authorship are listed.

### Funding

The work in J.R.W.'s laboratory described in this paper has been supported by: NIH grants NS22189, HD36020, NS061823, NS069551, P01HD32571, and P41EB018783; VA Merit Awards 1 I01 BX002550 and 5 I01 CX001812; the United Cerebral Palsy Research and Educational Foundation; the International Spinal Research Trust; the Paralyzed Veterans of America; the Christopher and Dana Reeve Paralysis Foundation; the New York State Spinal Cord Injury Research Board; and the Albany Stratton VA Medical Centre.

### Acknowledgements

The authors thank Drs Elizabeth Winter Wolpaw, Aaron P. Batista, and Gerald E. Loeb for their many important and detailed comments and suggestions as this paper evolved through numerous drafts. The authors are also grateful to Drs Björn Brembs, Jonathan S. Carp, Xiang Yang Chen, Russell L. Hardesty, John Krakauer, Dennis J. McFarland, Emily Oby, Rajiv R. Ratan, Aiko K. Thompson, and Yu Wang and to Ms Theresa M. Vaughan, all of whom made substantive comments that led to improvements. Mr Kevin Long provided excellent assistance in preparation of the figures and references.

### Keywords

behaviour, learning, memory, neuroplasticity, rehabilitation, skill

## Supporting information

Additional supporting information can be found online in the Supporting Information section at the end of the HTML view of the article. Supporting information files available:

**Peer Review History**

