## [Peer Review History · The Journal of Physiology]

Heksor: The CNS Substrate of an Adaptive Behavior

Jonathan R. Wolpaw and Adam Kamesar

DOI: 10.1113/JP283291

Corresponding author(s): Jonathan Wolpaw (wolpaw@neurotechcenter.org)

The following individual(s) involved in review of this submission have agreed to reveal their identity: Karl Friston (Referee #3)

Review Timeline:

Submission Date:	22-Jan-2022
Editorial Decision:	10-Mar-2022
Resubmission Received:	10-May-2022
Editorial Decision:	20-May-2022
Revision Received:	03-Jun-2022
Accepted:	20-Jun-2022

Senior Editor: Ian Forsythe

Reviewing Editor: Richard Carson

Transaction Report:

Dear Dr Wolpaw,

Re: JP-TR-2022-282854X "Heksor: The CNS Substrate of an Adaptive Behavior" by Jonathan R. Wolpaw and Adam Kamesar

Thank you for submitting your manuscript to The Journal of Physiology. It has been assessed by a Reviewing Editor and by 4 Referees and the reports are copied below.

Please let your co-author know of the following editorial decision as quickly as possible.

As you will see, in its current form, the manuscript is not acceptable for publication in The Journal of Physiology. In comments to me, the Reviewing Editor expressed interest in the potential of this study, but much work still needs to be done in order to satisfactorily address the concerns raised in the reports.

In view of this interest, I would like to offer you the opportunity to carry out all of the changes requested in full, and to resubmit a new manuscript using the "Submit Special Case Resubmission for JP-TR-2022-282854X..." on your homepage.

We cannot, of course, guarantee ultimate acceptance at this stage as the revisions required are substantial. However, we encourage you to consider the requested changes and resubmit your work to us if you are able to complete or address all changes.

A new manuscript would be renumbered and redated, but the original referees would be consulted wherever possible. An additional referee's opinion could be sought, if the Reviewing Editor felt it necessary. A full response to each of the reports should be uploaded with a new version.

I hope that the points raised in the reports will be helpful to you.

Yours sincerely,

Ian D. Forsythe
Deputy Editor-in-Chief
The Journal of Physiology
<https://jp.msubmit.net>
<http://jp.physoc.org>
The Physiological Society
Hodgkin Huxley House
30 Farringdon Lane
London, EC1R 3AW
UK
<http://www.physoc.org>
<http://journals.physoc.org>

EDITOR COMMENTS

Reviewing Editor:

This is an ambitious paper. The authors seek to provide "a new paradigm for understanding how adaptive behaviors are acquired and maintained". As acknowledged by all referees, the topic is important and of great intrinsic interest. The piece has several positive attributes, and in many respects it is scholarly and engaging. The potential offered to the scientific community by this new formulation does however depend on the extent to which it can offer new insights, provide a conceptual framework with which to generate novel experimental hypotheses, and advance knowledge in a manner that would not otherwise be possible. As you will discern, the referees differ in the degree to which they believe that the project satisfies these requirements. I also share the concern that the Heksor concept failed to advance my own understanding beyond that accrued from classical experimental approaches and various types of formal modelling. Perhaps I simply lack the capacity to derive the necessary understanding. This does however point to a problem that has been emphasised by the referees. The way in which the project offers something that is new and worthy of the attention of those working in the relevant fields is not yet obvious. This is not to say that the Heksor concept does not possess this capacity. It is rather that its presentation seems to require further development, to convince the target audience that any shift in thinking prompted by this approach will prove to be worthwhile.

Senior Editor:

As you can see your article has attracted a lot of interest - both positive and negative. The RE and I are keen to see a response and revision of your article. The presentation requires a "root and branch" revision with a view to enhancing the clarity of exposition. It remains to be seen whether this would reduce or consolidate the dissent. When you are re-writing, please make the article short, with clear arguments that will appeal to a wide audience. I hope you are able to undertake this revision, as I think there is an opportunity, which at present is not realised. I look forward to your revised MS.

REFEREE COMMENTS

Referee #2:

This is a brilliant review on the highly important and hotly debated issue of the neurophysiological mechanisms of everyday behaviors. The authors present an in-depth historical analysis, explain the origins and meaning of the new term (heksor), and then review more recent studies demonstrating and exploring interactions among neural circuits underlying components of natural behaviors (heksors) leading to the notion of negotiated equilibrium. Overall, the authors make a very convincing case for the new term, its meaning, and its role in behaviors and their changes with development and practice. I have only a few small comments that the authors are free to use or to ignore. The paper is very good as it stands.

On a number of occasions, the authors refer to Bernstein (1967). The more recently (2020) published translation of Bernstein's classic (1947) offers a more in-depth presentation of his ideas that seem rather directly related to some of the central notions in the review. In particular, Bernstein describes his "Level of Synergies" as composed on neural circuits with the purpose to ensure dynamical stability of salient performance variables (while allowing other variables to vary). This notion seems rather closely related to the concept of heksor as a network maintaining "the key features of the behavior, the features that make the behavior satisfactory". Further in the 1947/2020 book, Bernstein describes interactions among synergies that evolve with development and practice and help the person to avoid negative interference between them, which seem to be precursors of the concept of "negotiated equilibrium".

The definition of synergies accepted by the authors reflects only half of the definition introduced by Bernstein (1947/2020). In that book, synergies have two functions: (1) grouping variables (frequently studied in our times as reflected in the references cited in the paper) and (2) ensuring dynamical stability of action (less commonly studied). Whether the former function reflects a negotiated equilibrium among heksors is a very interesting question. Whether there are more important (basic) and less important (more volatile) heksors is another interesting question. Maybe the former define the grouping of elements, whereas the latter have to deal with already established groups.

The concept of "satisfactory behavior" seems to be rather closely related to the concept of satisficing introduced by Herbert Simon and the idea of "good enough" (in contrast to optimal) behaviors advanced by Jerry Loeb and others.

Heksors are sometimes described in the paper as entities that "produce behaviors". There is a nontrivial issue of producing "same behaviors" in different external force fields (e.g., pointing or throwing a tennis ball to a partner during rotation in the centrifuge, cf. Lackner). Some variations in the force field probably do not require new heksors, whereas others do. The key word seems to be "destabilization" (cf. Bernstein). If salient performance variables are destabilized by the new force field, the behavior becomes qualitatively different and requiring a new heksor. This point may deserve an in-depth discussion.

Another issue that seems to be relevant is the fact that the central nervous system, by its very design, cannot prescribe external mechanics and muscle activations independently of the (everchanging and unpredictable) external force field. This issue has been discussed in detail starting from Bernstein 1947/2020. So, neural networks have to operate with indirect variables (e.g., Feldman's lambdas), which then indirectly produce behaviors. What variables do heksors operate with within the suggested scheme and how do these variables translate into important behavioral variables? Maybe an explicit set of answers to these questions would be useful.

Referee #3:

I enjoyed reading this scholarly and engaging paper. It is clearly written by authors who have a deep intuition about neural plasticity and distributed processing in the motor system. I also enjoyed the interesting and eclectic mixture of etymology and neurophysiology.

There are a few points that I felt you could unpack for the general reader. Perhaps you could consider the following:

There is an extensive and compelling case made for the word *heskor*, which is carefully defined. It would be nice to balance this with an equivalent motivation of 'negotiated equilibrium'. In other words, spend a paragraph or two describing the etymology of negotiated equilibrium - providing a concise definition. The choice of the word equilibrium is interesting here. It reminds one of things like the Nash equilibrium in game theory. Alternatively, from the point of view of statistical physics, it speaks to the notion of a nonequilibrium steady-state. I mention this because there is an interesting connection between the exchange and synchronisation of multiple *heskors* and the self organisation implicit in nonequilibrium steady-state dynamics. The key thing here is that a nonequilibrium steady-state deals with systems that are open. In the present setting, this means the notion of a nonequilibrium steady-state accommodates the exchange of energy and information between *heskors*. Finally, the existence of a nonequilibrium steady-state implies an attracting set of ('virtuous') states to which the system will self organise. I'm not suggesting that you cover this in your paper; however, it would be nice to substantiate the concept of a negotiated equilibrium by providing some synonyms or homologues in other fields (e.g., generalised synchrony, self organisation, dynamic pattern coordination, nonequilibrium steady-state, et cetera).

A lot of the neurophysiological arguments rests upon the notion of down-conditioning. I think it would be useful for the general reader to include a brief description of the operant conditioning paradigm that produces down-conditioning. For example, a brief vignette or description of how you would implement the paradigm in the clinic.

Furthermore, what determines the difference between down and up-conditioning? Is it the scheduling of the conditioning or the nature of the conditioned and unconditioned stimuli?

On line 279, you introduce the notion of a "continually plastic CNS". It might be useful to emphasise this point with a few quantitative examples. For example, cite a couple of studies of synaptic turnover, showing that synapses only persist for minutes or hours in (certain parts of) the brain and can be regarded as a dynamic flux or fluid of connectivity, especially in systems with high synaptic turnover.

On line 289, you talk about the *heskor* "changing the set of neurons and synapses that comprise it". When exemplifying this using the H-reflex, it might be useful to introduce the notion of diaschisis (Greek for shocked throughout). In particular, the changing connectivity in one region - induced by changes in another - has been studied under the notion of a dynamic diaschisis [1, 2]. This might be closely related to the notions of primary and secondary plasticity introduced on line 646.

On line at 710, you talk about the new *heskor* interfering with the old. It might be nice to reference a couple of papers that consider 'catastrophic interference' and 'catastrophic forgetting', which are cardinal concepts in computational neuroscience - and represent an outstanding challenge that speaks to the ability of *heskors* to maintain their negotiated equilibrium.

I hope that these comments help should any revision be required.

1. Price, C.J., et al., Dynamic diaschisis: anatomically remote and context-sensitive human brain lesions. *J Cogn Neurosci*, 2001. 13(4): p. 419-29.

2. Carrera, E. and G. Tononi, Diaschisis: past, present, future. *Brain*, 2014. 137(Pt 9): p. 2408-22.

Referee #4:

Everything that I have to say about this article is summarized in my 11 page referee report. [ATTACHED]

Referee #5:

This manuscript presents an outline of what the authors claim is a "new paradigm" for understanding adaptive skeletomotor behaviour (at least that's how I read it, I saw no evidence for inclusion of other kinds of behaviour). It identifies and defines, in general terms, a new conceptual CNS structure (if 'structure' is the right word), which is given the name 'heksor'. An historical review is provided that provides a background for the development of the new paradigm and the basis for the new heksor concept. It then reviews some further recent evidence for the existence, explanatory validity and wide applicability of this kind of entity, drawing on the first author's previous notion of 'negotiated equilibrium' described in an earlier review for *The Journal of Physiology*.

There is no doubt that this article deals with several important and interesting ideas that are fundamental to the understanding of motor behaviour. In addition, the background review provides an intriguing discussion of a number of empirical findings. However, I found it hard to see exactly what was new in the authors' paradigm and what novel insights it offers. Perhaps this is because I did not properly understand the conceptual framework being developed. If this is the case, the comments that follow should be taken as pointers to how the authors' message might better be communicated. If the authors have indeed some profound new concepts to offer, then clarity of exposition is vital if they are to have the impact they deserve.

The comments below concern conceptual matters rather than specifics of exposition, data interpretation or typographical errors. I see the conceptual issues as things that will require major revisions to sort out and I would not expect a revised manuscript to retain much of the text in its current form. For this reason, I offer no comments on non-conceptual matters.

Specific comments

(1) 'Adaptive' behaviour

The notion of 'adaptive behaviour' is fundamental to the discussion and is introduced on page 3, but it is not defined. Perhaps the authors think that this is too obvious to bother with. In lines 75-76, it is stated that an adaptive behaviour has a 'standard' by which it can be evaluated as 'good' or 'bad'. On the face of it, this seems to be fairly straightforward and appears to mean that adaptive behaviour is goal-directed behaviour. After all, a behavioural goal is a 'standard' by which success (good) and failure (bad) can be evaluated (of course, a given behaviour may have several 'goals' since criteria like 'minimize energy expenditure' are also a goals).

The problem with 'goal-directedness' in biological systems is determining what the goals are: how can it be done? This is a question that is fundamental in behavioural ecology and ethology, but the authors do not address it - perhaps they are of the opinion that this is too straightforward to be worthy of comment. But it isn't and the authors' treatment of H-reflexes would

seem to support this.

The authors claim (e.g., lines 294-5) that the H-reflex is not an adaptive behaviour. This is (probably) true when the H-reflex is viewed as something that doesn't exist in nature but is instead a consequence or artifact of the spinal circuitry that connects muscle spindles to motoneurons. From this point of view, the H-reflex is without biological purpose. Is this what the authors are saying? I'm not sure since they aren't explicit, instead they emphasise that the H-reflex is the electrical analogue of the spinal muscle stretch reflex. The stretch reflex is a goal-directed/adaptive behaviour of a sort. What is the goal of the reflex action? This is not a question that is easily answered, but the effect of the reflex is to increase the stiffness and damping properties of skeletal muscles, which has led to the hypothesis that the stretch reflex functions to give muscles the stiffness and damping properties necessary to support effective behaviours of other kinds: the stretch reflex is useful, it is an adaptive behaviour. What are the authors saying, therefore? I'm not sure, but it would help to be clear about it. If the H-reflex is to be viewed as an electrical artifact of underlying neuromuscular circuitry and not an adaptive behaviour, then it's not a very good example to be using in a discussion of adaptive behaviour. I found the discussion of how the H-reflex can be given a goal using instrumental conditioning to be rather unconvincing and limited in scope.

It is asserted that there are three insights concerning adaptive behaviours obtained over the last few decades: (i) even the simplest adaptive behaviours are learned (line 79). (ii) "... the CNS substrate of an adaptive behavior comprises a network of neurons and synapses that is widely distributed through the CNS" (lines 90-91). (iii) "... the network of neurons and synapses underlying an adaptive behavior changes through life" (lines 99-100). I will discuss these three insights in turn.

(2) Insight (i) - all adaptive behaviors are learned.

The problem with the authors' treatment of this matter is again partly definitional: what do they mean by 'learning'? They don't tell us, so perhaps they think this is a matter too obvious or trivial to be worthy of comment. If so, I beg to differ.

The authors emphasize 'simple' behaviours, by which they mean skeletomotor reflexes like the withdrawal reflex. They seem to be saying that such simple behaviours had been viewed as 'hardwired' into the nervous system, requiring no 'learning' to establish the underlying circuitry. Since the 1970s, this view has been shown to be wrong and that such reflexes (and possibly other kinds of simple behavior as well) are learned.

There are two points that should be made here. The first is that the view that some behaviours are hardwired was never widespread and constantly came in for criticism whenever it was suggested, such critiques can be found in literature dating from at least the beginning of the 20th century.

In the 1930s, Konrad Lorenz and Niko Tinbergen famously argued for a version of the hardwired theory usually called 'instinct theory'. Instinctual behaviour is, "hereditarily determined ... part of the original constitution of the animal ... arises quite independently of the animal's experience and environment ... is distinct from acquired or learned behavior ... [and] develops fully-formed in animals which have been prevented from practicing it". This quote comes from Lehrman's (1953) critique of Lorenz and Tinbergen's 'theory'.

The essence of Lehrman's critique is that there is no such thing as an instinctual behaviour as defined by Lorenz and Tinbergen. He cites a number of findings from literature published between about 1920 and 1950 to support this contention. Many of the examples demonstrate not that the behaviours need to be learned (at least not according to any definition of learning I am aware of), but that activity of the muscles and neurons is essential for development of the circuitry underlying the behaviour. Surely this is true of any phenotype - for anything to develop, some kind of environment is required. In the simplest case, the environment provides the materials from which tissues are constructed, but other factors such as the force of gravity, interuterine space, presence of other developing foetuses, freedom to contract muscles and move limbs (and resistance/constraint on such motion) may be equally significant. Only to the extent to which such factors are the same across all individuals, would we be justified in speaking of genetically determined phenotypes. If they vary, then the phenotype is not genetically determined. In short, developing and maturing in interaction with the environment is not the same as learning, in my view anyway. What is the authors' view? If it's different, can they justify it?

There is clearly a difference between reflex behaviour that is learned through specific kinds of experience - such as, for example, conditional reflexes acquired as the result of Pavlovian conditioning procedure - and reflexes that develop through interaction with uterine or other pre-natal environments without such specific experiences, in some cases through simple exercise.

In my opinion, the hard-wired behaviour notion the authors claim as having been believed in 1970 and only gradually abandoned under the weight of more recent empirical evidence, is a caricature, a straw-man erected only for the purpose of being dismantled. A suitably restricted historical review can be used to erect such a straw man, but it's unlikely to convince those of us who cast our nets into the wider scientific literature pre-1970.

(3) Insight (ii), the substrate of adaptive behavior

The claim that the substrate is "a network of neurons and synapses that is widely distributed through the CNS" seems to me to be something that could hardly be less controversial as a general statement. Perhaps it might seem controversial from the point of view of the Sherringtonian concept of reflex. Sherrington introduced the notion of the 'simple reflex' at the beginning of the 20th century, but it is now often forgotten that this notion was not that of the reflex as observed behaviourally - it was, according to Sherrington, a 'fiction', a hypothetical entity introduced as a kind of basic component of reflex behaviour. Sherrington's simple reflex was an effort to identify the atoms from which observed natural reflex behaviour is constructed. Thus, the stretch reflex or the withdrawal reflex are not (in Sherrington's terms) simple reflexes - they are natural reflexes built of many reflex atoms, many 'simple reflexes'.

The circuit for a simple reflex is, of course, the reflex arc. The circuitry of a natural reflex must, by definition, be far more complex. Sherrington himself didn't get very far with describing what this circuitry might involve, but he did discuss the need for things like reciprocal inhibition arcs and the circuitry mediating interactions between competing and cooperating reflexes. These circuits were subsequently elucidated in greater detail and we now know quite a lot about the circuitry underlying the various components of, for example, the vestibular-ocular reflex (VOR) and its interactions with cervico-ocular reflexes, with cervico-collic and vestibulo-collic reflexes, as well as with reflex saccades and fusional vergence. The circuitry of some ocular reflex interactions, such as those between fusional vergence reflexes, pupillary and accommodation reflexes were hypothesized by Helmholtz in the 19th century.

A nicely instructive example of the substrate of a natural, functional reflex is the wiping reflex of the frog. This is a marvellously complex behaviour that is clearly goal-directed. So marvellous in fact, that the reflex as manifested by the headless (spinal) frog suggested to scientists in the 18th and 19th Centuries that the frog's spinal cord must be in possession of a 'soul'. Investigation of the circuitry mediating this 'soul' was conducted from the 1950s to the 1970s by scientists in the Soviet Union and was revealed as involving complex circuitry throughout the spinal cord that enabled the reflex to be instantly adapted to a wide variety of novel circumstances without any prior experience.

It seems to me to be inconceivable that anyone would mistake the reflex arc of introductory textbooks with the circuitry underlying natural reflexes like the withdrawal reflex, the VOR or the frog's wiping reflex - behaviours that flexibly adapt to changes in the conditions in which the responses are executed and that change with experience in numerous ways from habituation and sensitization to improvements in skill.

Needless to say, more 'sophisticated' voluntary behaviour has been viewed as involving complex distributed neural mechanisms for as long as people have speculated about how such behaviour is produced. Since most, if not all, such behaviour is viewed as being wholly or partly learned, the neural substrate has never been considered to be anything other than "a distributed network of neurons and synapses that ... changes as needed to maintain the key features of the behavior." (lines 274-275). Thus, I cannot agree with the authors' claims that such an 'entity' is a "previously unrecognized" one. I for one have recognized it (or something closely equivalent) for a very long time (I don't care to admit just how long) and so have many of my colleagues. Curiously, unlike the authors, I have never thought it needed a special name and remain unconvinced of the necessity.

(4) Insight (iii), the substrate changes through life

The authors assert that this insight is the "most consequential" (line 99), yet from my perspective it seems the least insightful

and most obvious. In some ways, the notion that the substrate changes through life is trivial since it simply means that behaviour can change as a result of experience, which has been known to be true for voluntary behaviours for a lot longer than 50 years. Reflex behaviours have been known to change since at least Pavlov's day and it's been known since at least the mid 1970s that changes in 'simple' reflexes like the stretch reflex can involve regions of the nervous system outside of the circuits that mediate the reflex response itself. For example, the suppression of the autogenic stretch reflex in the gastrocnemius that occurs in response to repeated tilts of the surface on which a person stands was shown to involve circuitry beyond the autogenic reflex arcs by Nashner (1976). Curiously, in the 1960s and early 1970s it was quite widely believed that the changes in simple spinal reflex mechanisms was driven exclusively by circuits in the brain - the basic spinal mechanisms themselves (the reflex arcs) were not intrinsically capable of learning. An odd point of view when you consider that it was well known at the time that learning could take place in the simple nervous systems of invertebrates in circuitry far simpler than that of any vertebrate spinal cord (see Grau et al, 2006; Oakley, 1979) and that habituation and sensitization of withdrawal reflexes involved changes in the spinal circuits themselves (Groves & Thompson, 1970).

(5) Additional comments

It seems to me that the insights covered above have been around a lot longer than a mere 50 years. Granted, they have become more mainstream over the last few decades and the evidence base is now overwhelming, but the alternative hard-wired view was never as mainstream as the authors seem to believe - at least not in the behavioral sciences. Admittedly, the instinct theory of Lorenz and Tinbergen was influential during the 1940s and 50s, but it was already under pressure in the late 1940s and was being revised to incorporate the data on behavioural adaptability and learning in the 50s and 60s. The title of J. P. Hailman's Scientific American article of 1969 describing his body of work on feeding behaviour in gull chicks was provocatively entitled 'How an instinct is learned'.

It would be very surprising, therefore, given the antiquity of these insights and their widespread acceptance from the 1980s onwards, if there hadn't been efforts to understand in greater detail the neural mechanisms of adaptive behaviour along the lines the authors deem necessary. To my mind, there has been a lot of work on exactly this. The work on the mechanisms of the frog's wiping reflex provides one example, but there are many others. The work on the neural basis of rhythmic behaviour, particularly locomotion, is also very pertinent.

The circuitry underlying locomotor behaviour in species such as the lamprey is now well understood thanks, in large part, to the work of Sten Grillner and his colleagues. The circuitry involved seems to have exactly the character required by the insights (ii) and (iii) listed earlier, so I was surprised to find that none of the work of Grillner's group was described or cited, despite locomotor behaviour being discussed at several points in the manuscript (first on page 10). The cursory discussion of locomotor mechanisms on page 10 is framed in 'heksor' terms and seems not only extremely vague and generic, but also appears to have overlooked the possibility of hopping. Human beings are capable of dealing with the loss of one leg by hopping on the other one and other animals can also deal effectively with injury to or loss of one or more legs. Changes in behaviour that preserve locomotor functionality occur almost immediately (they do not need to be learned) and such adaptation can be readily explained by existing models of the underlying mechanisms.

Much of the discussion of the H-reflex and its 'natural' counterpart, the stretch reflex, also suffers from a failure to discuss alternatives. Many researchers have sought to understand the stretch reflex in terms of its role in other behaviours, both reflex and voluntary. Merton's servo hypothesis of 1953 is perhaps the first of these efforts, Anatol Feldman's lambda-model is a more recent development, though it appeared in an early form as early as 1966. In these cases, the reflex is a manifestation of a negative feedback type control mechanism - a more useful conceptualization than the rather difficult term 'reflex' (which is not an aid to understanding).

I would like to make one last comment. This concerns the treatment of the concept of memory that begins at the bottom of page 10. The authors contrast a 'standard' view of memory with that associated with the heksor concept. This seems to me simply to be the long recognised distinction between declarative and non-declarative memory. A distinction known to be associated with different CNS networks as demonstrated in humans in the 1950s based on studies of the famous anterograde amnesic Henry Molaison (HM). Declarative memory (the non-heksor kind of memory described on page 11) is exactly the sort of memory that is not associated with motor behaviour and skill. The kind of memory involved in motor skill acquisition and underlying learned behaviours is a kind of non-declarative memory. The kind of non-declarative memory involved is one that is not separable from the mechanisms/circuitry underlying production: there is not memory plus the circuitry, the memory is part of the circuitry - it is realized in the strength and nature of the connections that exist in the circuit. This connectivity is changed by experience and so the behaviour changes. This, after all, is the kind of memory described by Kandel and colleagues in their studies of reflex habituation and sensitization and it applies to other forms of behavioural change and acquisition.

I was reminded of Karl Lashley's famous 'search for the engram' dating from the 1920s and summarized in his famous paper of 1950 (Lashley, 1950). His work was guided by the view prevalent at the beginning of the 20th Century that sensorimotor learning involves connections between posterior, sensory cortical areas and frontal motor areas. He found that excising cortical tissue so as to separate sensory and motor areas did not impair the ability of experimental animals (rats) to learn sensorimotor tasks. From these studies, he concluded that sensorimotor memory was very widely distributed throughout cortex and that any neuron within the tissue could be involved (principle of 'equipotentiality'). He also found that memory impairment was correlated not with the region of cortical damage, but with the amount of damage and concluded that all the areas acted together to form the engram (principle of 'mass action'). Although the evidence base for Lashley's conclusions came mainly from lesion studies of the cerebral cortex, his conclusions represent ideas clearly at odds with any simple notion of hard-wired sensorimotor circuitry and separable sensorimotor memory. They were, to some extent at least, validated by later studies of formal neural networks and distributed sensorimotor learning and memory.

Much of the later sections of the manuscript (from page 18 on), I found to be rather vague. It would help to make these section more specific, perhaps more model based. At present they seem to have been tacked on at the end in order to 'prove' the usefulness of the heksor concept. For this reason, I will not comment on them further.

(6) Concluding remarks

Overall, despite the fact that the authors touch on a number of important, fundamental issues in skeletomotor control, I did not feel that I had learned anything new from reading the manuscript. The treatment of these issues was rather vague and couched in verbal, conceptual terms; formal models and detailed circuits were not presented or described. I was left asking how the authors conceptual treatment relates to more formal, model-based approaches to the issues. It is not that such approaches are rare or of poor quality - quite the opposite.

The manuscript devotes itself to a restricted range of basic and reflex behaviours, but even here formal, model-based approaches are not referred to. I found this to be a glaring omission. In particular, model-based approaches to the role of spinal stretch reflexes are completely absent from the manuscript, despite a large amount of it being devoted to behavioural phenomena associated with these reflexes and their electrical analogue (the H-reflex).

If the authors are to convince the reader that they have something genuinely new to say, then they are going to have to explain how and why previous work has failed to address the issues or has failed to address them 'properly'. My impression is that previous work has addressed them, perhaps not successfully, but they have been addressed. In addition, I do not see that the authors are offering a genuinely new approach - the current text is too vague to be certain, however. Unfortunately, much of the manuscript is devoted to erecting a straw man and identifying a better evidenced alternative - this effort fails to reference important literature and seems to cherry pick research in order to establish the straw man's reality. I was not convinced, though I confess that I may have misunderstood. If so, others are likely to have similar problems.

References

- Grau, J.W. et al. (2006) Behavioral & Cognitive Neuroscience Reviews, 5, 1-48.
- Groves, P. & Thompson, R.F. (1970) Psychological Review, 77, 419-450
- Lashley, K.S. (1950) Symposiums of the Society of Experimental Biology, 4, 454-482
- Lehrman, D.S. (1953) Quarterly Review of Biology, 28, 337-363
- Nashner, L. (1976) Experimental Brain Research, 26, 59-72
- Oakley, D. (1979) in Oakley, D. & Plotkin, H. (Ed) Brain, Behaviour & Evolution. London: Methuen.

ADDITIONAL FORMATTING REQUIREMENTS:

-Your MS must include a complete "Additional information section" with the following 4 headings and content:

Competing Interests: A statement regarding competing interests. If there are no competing interests, a statement to this effect must be included. All authors should disclose any conflict of interest in accordance with journal policy.

Author contributions: Each author should take responsibility for a particular section of the study and have contributed to writing the paper. Acquisition of funding, administrative support or the collection of data alone does not justify authorship; these contributions to the study should be listed in the Acknowledgements. Additional information such as 'X and Y have contributed equally to this work' may be added as a footnote on the title page.

It must be stated that all authors approved the final version of the manuscript and that all persons designated as authors qualify for authorship, and all those who qualify for authorship are listed.

Funding: Authors must indicate all sources of funding, including grant numbers. If authors have not received funding, this must be stated.

It is the responsibility of authors funded by RCUK to adhere to their policy regarding funding sources and underlying research material. The policy requires funding information to be included within the acknowledgement section of a paper. Guidance on how to acknowledge funding information is provided by the Research Information Network. The policy also requires all research papers, if applicable, to include a statement on how any underlying research materials, such as data, samples or models, can be accessed. However, the policy does not require that the data must be made open. If there are considered to be good or compelling reasons to protect access to the data, for example commercial confidentiality or legitimate sensitivities around data derived from potentially identifiable human participants, these should be included in the statement.

Acknowledgements: Acknowledgements should be the minimum consistent with courtesy. The wording of acknowledgements of scientific assistance or advice must have been seen and approved by the persons concerned. This section should not include details of funding.

Review of
Heksor: The CNS Substrate of an Adaptive Behavior
by J. Wolpaw and A. Kamesar
submitted to *The Journal of Physiology*
February 7, 2022

GENERAL COMMENTS AND RECOMMENDATION

I have read the Wolpaw and Kamesar article carefully and am forced to recommend that it be rejected without possibility of resubmission.

Partly this recommendation is based on the fact that the authors seem to have no knowledge of advances in biological neural models during the past half century. These models have provided unified and principled explanations of large amounts of psychological and neurobiological data, including data of the kind described by the authors, in contradiction to the authors' apparent claim that such explanations are not available. The models have also proposed numerous predictions, many of which have been supported by subsequent neurobiological experiments.

The authors seem to add nothing to available explanations of such data. As an active neural modeler for many years, I found nothing new in the manuscript.

Partly this recommendation is due to the fact that the introduction of the term "heksor" that is a focus of the current article has no explanatory power. I believe that one could remove the word throughout the manuscript without changing the text. Modest changes in sentences that include the word would be sufficient.

The fact that the authors believe that a new word was needed highlights their unawareness of the current state of knowledge. There are already many names in the biological neural modeling literature for concepts, mechanisms, and models that explain neurobiological data, as I will illustrate below.

Finally, what the authors write about the data that they discuss does not explain these data. Adding the word "heksor" adds no explanatory power to their discussions. And these data already have descriptive names that are widely accepted in the experimental and modeling communities, as I will illustrate below.

SPECIFIC COMMENTS

I comment below on sentences in the manuscript that illustrate the concerns summarized in the **GENERAL COMMENTS**. My comments are made in *boldface italics* to distinguish them from the authors' text. I print some of the authors' phases in red to highlight them.

Lines 46-71: “Introduction: In the past half-century, the hardwired CNS of 1970, which could change in only a few places and in only a few ways, has gradually become the ubiquitously plastic CNS of today, in which change is the rule rather than the exception...This transformation requires and enables a new paradigm for addressing a central question in neuroscience: **how are adaptive behaviors (behaviors that serve the needs of the individual) acquired and maintained throughout life?** Studies that support various aspects of this new paradigm are appearing with increasing frequency in the literature. **This article describes the new paradigm.** It addresses the advances leading to the paradigm, the evidence that now supports it, and experimental strategies that can test its wider validity.

“**The core of the new paradigm is a previously undefined CNS entity** that underlies an adaptive behavior. This entity is here given the name *heksor*. The first section reviews the advances that **transformed the prevailing view of the CNS** and, in doing so, **revealed the existence of the entity to which we are applying the name *heksor*.** The next section describes the process that arrived at *heksor* as an appropriate name, defines and explicates the new term, and states its fundamental differences from existing terms.

“The central message of the presentation is that the progress of the past 50 years compels neuroscience to **develop a new paradigm. The old paradigm – with its assumptions and its terminology – was designed for a largely hardwired CNS; it is not suitable for what we now understand to be a ubiquitously plastic CNS.** By introducing the concept of the *heksor*, linking it to the complementary concept of the *negotiated equilibrium* that *heksors* create, and discussing the implications, this article introduces, formalizes, and begins to explore **a new conceptual paradigm** that is driven by the recent advances and encompasses them.”

The authors seem to be unaware of a torrent of biological neural modeling articles that have provided principled and unifying explanations of hundreds of psychological and neurobiological experiments the past 50 years. Many of these models individually unify the explanations of psychological, neuroanatomical, neurophysiological, biophysical, and biochemical data. They have also made many predictions that have been supported by subsequent experiments.

Making such a linkage between brain and mind is essential for the understanding of either of them. This is true because BRAIN evolution needs to achieve BEHAVIORAL success. The models do more than claim that brains achieve behavioral success. They explain HOW this is accomplished.

In particular, biological neural models explain HOW interacting brain mechanisms generate EMERGENT PROPERTIES that match the psychological experiences that they generate. Rigorous models are needed to explain how such emergent properties arise. Informal descriptions cannot.

The present article adds nothing to this burgeoning interdisciplinary literature.

I collaborate with a leader in this field, Stephen Grossberg. His web page sites.bu.edu/steveg includes over 500 articles about essentially all the major brain processes that give rise to perception, cognition, emotion, and action in healthy individuals and clinical patients.

These models include qualitative explanations and quantitative computer simulations that explicate the three “New Insights about Adaptive Behaviors” that Wolpaw and Kamesar describe. These explanations and simulations show that the insights are not “new”.

The insights that are proposed to be new are the following ones:

1. even the simplest adaptive behaviors are learned.
e.g., flexion-withdrawal reflexes

Page 4:

Lines 90-91:

2. “the CNS substrate of an adaptive behavior comprises a network of neurons and synapses that is widely distributed through the CNS.”

Lines 99-110:

3. “The third and most consequential insight is that the widely distributed network of neurons and synapses underlying an adaptive behavior changes through life.”

“Changes are guided by feedback during the behavior and by its outcome; they reduce errors and improve performance”.

In contrast to the above purely verbal statements, Adaptive Resonance Theory, or ART, has been incrementally developed since it was first published in 1976 into the most advanced cognitive and neural theory of how our brains learn to attend, recognize, and predict objects and events in a changing world.

The neural networks that are described in ART are defined by systems of nonlinear feedback networks that have been supported by multiple kinds of experimental data. For example:

All the foundational hypotheses of ART have been supported by subsequent psychological and neurobiological experiments.

ART has also provided principled and unifying explanations of hundreds of additional experiments.

ART was derived in 1980 in an article in the journal Psychological Review from a thought experiment about how ANY learning system can autonomously correct predictive errors.

The hypotheses of this thought experiment are facts that are familiar to everyone because they represent ubiquitous environmental pressures on the evolution of our brains.

Nowhere during the thought experiment are the words mind or brain mentioned.

ART design principles and mechanisms are thus a UNIVERSAL solution to this learning problem, which is called the STABILITY-PLASTICITY DILEMMA because it asks how any system can learn quickly without experiencing catastrophic forgetting.

ART has also led to rigorous models of how humans consciously see, hear, feel, and know things about the world, and use these conscious states to effectively plan and act to realize valued goals.

This proposed solution of the mind-body problem thus arises from a computational analysis of how humans and other animals autonomously learn in a changing world.

Two mechanistically distinct kinds of “feedback” play a role in ART models:

(1) Recurrent competitive networks help to normalize distributed activity patterns while preventing them from getting overwhelmed by cellular noise or saturating, thereby solving the NOISE-SATURATION DILEMMA throughout the brain. These circuits also help to make competitive decisions.

(2) Top-down cortico-cortical and thalamo-cortical circuits learn to select the critical feature patterns to which ATTENTION is paid, and that control what is learned as well as the dynamical stabilization of learned memories, hereby solving the STABILITY-PLASTICITY DILEMMA.

Across brain regions and species, this attentional circuit obeys the ART Matching Rule, which is realized by specializations of a top-down, modulatory on-center, off-surround network.

ART-like networks govern processes of perception and cognition.

Spatial representation and action are governed by models that obey computationally complementary laws. This difference between the laws for perception/cognition and space/action embody the paradigm of COMPLEMENTARY COMPUTING.

For example, variations of Vector Associative Map, or VAM, model circuits obey computationally complementary laws to those of ART.

VAM circuits interact with spinal circuits that are modeled by Factorization of Length and Tension, or FLETE, circuits. Both VAM and FLETE interact with adaptively-timed and adaptively-gain-controlled circuits in the cerebellum to control properly calibrated actions.

Flexion-withdrawal reflexes, and how they adapt to an actor’s learned behaviors, form part of VAM-FLETE-cerebellar circuits.

Page 5:

Lines 125-131:

“In 2010, the *negotiated equilibrium* concept was introduced to acknowledge and begin to understand the process that enables the many widely distributed, overlapping, and continually changing networks of neurons and synapses underlying adaptive behaviors to maintain their behaviors throughout life (Wolpaw, 2010, 2018). According to the negotiated equilibrium concept, the process is organized and supervised by the networks themselves. Their concurrent efforts to maintain their own behaviors constitute a negotiation among them; they negotiate the properties of the CNS neurons and synapses that they all use. Through this process, they establish and maintain an equilibrium satisfactory to all of them.”

As noted earlier, this paradigm and its ability to maintain and adapt behaviors during life was introduced far before 2010 and rapidly developed from 1976 to the present day.

Moreover, using the phrase “negotiated equilibrium” is just a shorthand for saying that such learned adaptations occur. It adds no explanatory power. A “negotiated equilibrium” is just a name for the existence of feedback in the networks, but does not explain how the feedback works or what psychological functions it supports in different parts of a brain.

Only rigorous neural models can mechanistically explain how such feedback-mediated adaptations occur. The models in question, among other things, explicate the Piagetian concept of CIRCULAR REACTION by modeling how CYCLES of a perception-cognition-emotion-action loop with the environment discover, learn, and maintain behaviorally successful brain dynamics

Lines 140-141:

The text mentions the example of finger withdrawal from a hot stove.

Before a finger can be withdrawn from a hot stove, it must be placed on it. This problem requires a neural model of how humans and other animals control the DESIRED TARGET POSITION during reaching and multiple other behaviors with the fingers, hands, and arms. One of the most successful such models was introduced in 1988. It is the Vector Integration to Endpoint, or VITE, model for pointing and reaching behaviors by a finger, hand, arm system. VITE was progressively developed over the years into the VAM model, which can autonomously calibrate its VITE parameters, and the DIRECT model of MOTOR EQUIVALENT movement control.

The DIRECT model explains how, using a redundant arm, multiple trajectories of elbow, wrist, and finger configurations can all realize a desired goal position in space. A model that offers a computationally rigorous solution of this problem was published in 1993 in the Journal of Cognitive Neuroscience. It builds upon earlier modeling of reaching behaviors.

Lines 147-148:

“This widely distributed and continually changing network of neurons and synapses that underlies an adaptive behavior is a previously unrecognized entity. As such, it has no name.”

This claim is not correct and shows ignorance of a large modeling literature, including the VITE, VAM, FLETE, VITEWRITE, LIST PARSE, TELOS, and lisTELOS models, each of which is rigorously defined and used to explain and simulate large databases about adaptive sensory-motor control.

Page 6, line 158:

“find an appropriate name for the CNS substrate of an adaptive behavior.”

As noted above, many such names exist in the neural network literature both for fundamental concepts and for the models that explain different aspects of biological intelligence.

Another classical concept is due to Piaget, namely a CIRCULAR REACTION. This process helps to explain how an infant or young child learns behaviors such as goal-oriented reaching and language imitation by emitting simple movements or sounds during a developmental critical period and using sensory feedback from these behaviors to learn a perception-action loop, hence a CIRCULAR reaction, whereby to volitionally generate movements or language sounds later in life.

Grossberg has called the circular reactions of Piaget INTRAPERSONAL circular reactions to differentiate them from the INTERPERSONAL circular reactions that he and his colleagues have modeled to explain facts about social cognition, including how children learn to share JOINT ATTENTION with teachers whose actions are experienced at different positions in space.

Another, now classical, concept is that of an ADAPTIVE RESONANCE. Grossberg has shown how a FEATURE-CATEGORY RESONANCE links together processes of Consciousness, Learning, Expectation, Attention, Resonance, and Synchrony; what he calls the CLEARS processes.

Grossberg has shown how six different adaptive resonances, each with their own names and specialized properties, occur during different experiences of conscious perception and knowing. He has also explained how these resonances work, where they occur in the human brain, and even WHY evolution may have been driven to discover conscious states of mind whereby to control effective actions in the world.

Grossberg has also introduced the concept of COMPLEMENTARY COMPUTING to explain the computationally complementary properties of perception/cognition vs. spatial/action processes and uses these concepts to explain large interdisciplinary databases.

Some examples of higher-order processes that work with sensory-motor processes to carry out newly learned behaviors without necessarily changing their target positions in space are:

Item-Order-Rank prefrontal cortical working memories explain how sequences of items can be temporarily stored before being categorized through incremental learning, stably remembered, and used to control actions to realize desired goals in space. The same canonical laminar cortical circuit design, suitably specialized, can be used to model a linguistic, spatial, or motor working memory.

The ARTSCAN Search model proposes a rigorous solution of the Where's Waldo Problem, or how perceptual cognitive, and emotional brain processes cooperate during learning to categorize and find desired objects in a cluttered scene.

The GridPlaceMap model explains how entorhinal grid cells and hippocampal place cells are learned during spatial navigation and used to determine the positions where humans and animals are in a scene and how they navigate effectively within it.

Page 9, lines 274-275:

“A heksor is a distributed network of neurons and synapses that produces an adaptive behavior and changes as needed to maintain the key feature of the behavior”.

All of the above model examples accomplish the same thing, and characterize how to do it for different kinds of behaviors. The word “heksor” adds nothing to this understanding.

Page 10, lines 309-312:

“The locomotion heksor produces and maintains concurrent movements of the left and right sides of the body. Why then is locomotion not the product of two heksors, one controlling the right side and one controlling the left? Certainly, the movement of either side has key features that need to be satisfied for locomotion to be satisfactory.”

Instead of worrying about how to define a heksor, Grossberg and his colleagues have introduced and developed rigorously defined neural circuits for the control of movement gaits.

All of these gaits have been shown to emerge from interactions of specialized RECURRENT SHUNTING ON-CENTER OFF-SURROUND NETWORKS, a circuit design that is used in all parts of the brain to support multiple different functions.

This type of model simulates MOVEMENT GAITS such as walk and run in bipeds; and walk, trot, pace, and gallop in quadrupeds. In particular, the same Central Pattern Generator, or CPG, model simulates gaits and their transitions in cats (walk-trot-pace-gallop), humans (walk-run), and elephants (amble-walk). This CPG also clarifies how a volitional GO signal from the basal ganglia can cause these movements and their gait transitions to occur at variable speeds.

The heksor concept does not shed any light on why shunting on-center off-surround networks are ubiquitous in the brain. Grossberg has shown that they solve a fundamental problem, the NOISE-SATURATION DILEMMA, that is needed to effectively compute ANY distributed activity pattern across CELLS that have finite upper and lower bounds of activity. Their use in locomotion specializes this ubiquitous circuit design.

Page 11, lines 328-332:

“Like memories, heksors are created by experience; but there the similarity ends. The difference is simple: memories are passive; heksors are active. Memories are thought to be consolidated and stored, and then retrieved, accessed, or uploaded when they are needed. In contrast, **heksors change continually**. Memories are used; heksors are users. The CNS uses memories to guide behavior; heksors use the CNS to produce their behaviors.”

The above claim about memories is contradicted by neural models of how humans and other animals remember what they have learned. The STABILITY-PLASTICITY DILEMMA is solved by models such as Adaptive Resonance Theory by showing how learning can occur quickly without causing catastrophic forgetting. Stabilization of learned memories does not preclude continual refinement of them during subsequent experiences.

Memories hereby regulate behaviors as they are changed by them.

ART explains how memories are kept in a dynamical equilibrium that can be maintained for a long time, but then rapidly refined or corrected by either new information or disconfirmed expectations, respectively.

This is not speculation. Large amounts of data support the above conclusions, and their properties have been mathematically proved to occur in ART models.

Pages 11-12, 358-367:

“Sacco et al. (2009): (1) taught people a simple foot plantarflexion/dorsiflexion behavior; (2) recorded fMRI while they performed it; (3) trained them for a week 360 on a new dance style; and (4) again recorded fMRI while they again performed the simple plantarflexion/dorsiflexion foot behavior...These results indicate that acquisition of the new heksor has changed the old heksor; the behavior is maintained, but the CNS activity that produces it has changed.”

Multiple adaptations occur throughout the brain to accommodate the new plans and gains that learning the new dance style required. A general theme in known models is how environmental challenges engage the multiple brain regions that are needed to solve them, yet can still realize the same end behaviors, albeit in the service of different sensory-motor skills; e.g., walking vs. dancing.

Page 12, 368-378:

“Parker-Jones et al. (2012) compared in monolingual and bilingual people the brain activity associated with naming pictures or reading aloud in either language. As Figure 2c shows, a second language changes the activity that produces the first language. In bilingual people who are speaking either one of their two languages, several cortical areas participate much more than they do in people who can speak only one language. The need for two languages to share brain areas poses difficulties for each of them (e.g., using or understanding words in the two languages that sound similar but have different meanings, or sound different but have the same meaning; ensuring language-specific pronunciation and interpretation of pronunciation; parsing language-specific sentence structures). Such requirements may explain why the heksor responsible for a second language is more extensive than that responsible for a single language, and why the creation of the new language heksor is associated with similar expansion of the old (i.e., native) language heksor.”

Adaptive Resonance Theory has been applied to explain many facts about language development and performance throughout the lifespan, and the differences in learning difficulty if two or more languages are learned simultaneously in childhood vs. if a second language is learned later on.

ART explains via its stability-plasticity property how our brains dynamically stabilize previous learning against the challenges of new learning with similar stimuli, whether languages or other aspects of biological intelligence. Learning a second language sequentially is harder because many of the neural circuits that it engages are already dynamically buffering themselves against catastrophic forgetting of the first language.

A series of articles were published by Grossberg and his colleagues to explain different aspects of speech and language learning and performance. Model names include PHONET, ARTPHONE, ARTWORD, and conscious ARTWORD. The most advanced models explain how the laminar circuits of cerebral cortex, with identified neurons, are specialized to support speech and language.

My impression from reading various uses of the word “heksor” throughout the article is that it is just another way to talk about a brain “representation” without explaining how any particular representation works.

Page 12, 379-386:

“Zou et al. (2012) examined the change in the brain activity associated with a language when people learned to sign in the same language. This is a particularly interesting situation. In the new behavior, the same language is now linked to and dependent on a different sensory modality (vision vs. audition) and a different motor output (hand control vs. speech). Thus, the two heksors overlap considerably in the CNS areas underlying language comprehension and generation, while they overlap minimally in the CNS areas directly connected to language-related sensory inputs or motor outputs. When speaking the language, people who could also

sign in the language showed greater activation in several cortical areas than people who could merely speak the language.”

The fact that all Item-Order-Rank linguistic, spatial, and motor working memories share a similar circuit design helps to explain how speaking and signing can be coordinated. A word like HEKSOR is redundant in such an explanation, and just saying the word does not constitute an explanation.

425-463: “The H-reflex is the electrical analog of the spinal stretch reflex (e.g., the knee-jerk reflex) (Fig. 3a). It is elicited by weak electrical stimulation of the peripheral nerve and is produced mainly by a two-neuron monosynaptic pathway comprised of the Ia sensory afferent neuron, its synapse on the spinal motoneuron, and the motoneuron itself. This pathway is influenced by descending activity from the brain; **it is this descending activity that is shaped by the H-reflex operant conditioning protocol...**Creation of the new H-reflex heksor on the trained side of the spinal cord affects behaviors that use both sides. **Thus, the new heksor compels existing heksors that produce bilateral behaviors (i.e., behaviors, such as locomotion, that use both sides) to change in order to maintain the key features of their behaviors.** In the behaving animal, the brain and spinal changes resulting from the negotiation combine to produce a smaller H-reflex on the conditioned side and an unchanged H-reflex on the other side.”

Pages 13-14, 466-467: “Subsequent studies in cats by Rossignol and colleagues give further examples of how brain and spinal changes combine to keep a behavior unchanged.”

Again, the word heksor is redundant. Consider the above comments in the light of the entire Piagetian perception-cognition-emotion-action CYCLE that maintains a desired behavior aimed at achieving valued goals IN SPACE. How this happens has been explained and simulated by biological neural models.

Page 24, 807-824:

“Conclusion

The heksor and negotiated equilibrium concepts acknowledge and engage the reality revealed by the advances of the past 50 years: that the CNS is continually and ubiquitously plastic; that adaptive behaviors are produced by distributed networks of neurons and synapses that overlap each other; that these networks change continually to maintain their behaviors. The term heksor names the networks;

The term heksor does nothing to “name” the networks. It is redundant to their scientific description.

the term negotiated equilibrium names the process and outcome of their concurrent adaptive changes.

The term “negotiated equilibrium” is nothing more than a different term to describe an adaptive behavior.

Together these complementary concepts

What is “complementary” about them. Complementary Computing is so called because it describes computationally complementary perception/cognition and spatial/action mechanisms of learning and matching.

may explain how the ubiquitously plastic CNS acquires and maintains many different adaptive behaviors throughout life.

These concepts “explain” nothing. They are just empty labels that ignore 50 years of progress in modeling how brains make minds.

Together the concepts recognize and embody a new paradigm

There is no paradigm if there are no rigorous principles, concepts, or mechanisms that can successfully be used to actually explain lots of data, not just to talk about them in a superficial way.

with which to understand and explore the acquisition and maintenance of adaptive behaviors. At present, the strongest support for the concepts and the paradigm comes from studies of interactions among several relatively simple behaviors in the healthy CNS and in the damaged CNS. Similarly detailed evaluations of their wider applicability to other kinds of adaptive behaviors are essential and are possible with currently available methods. Many questions concerning the characteristics and functions of heksors and heksor interactions need exploration. The promising therapeutic implications and applications of the concepts also invite attention. All these studies will test the scientific and clinical usefulness of the new word heksor and, in doing so, they will assess the adequacy of the new paradigm based on heksors and the negotiated equilibrium that they create.

Dear Journal of Physiology Editors,

As you suggested, we have now revised the topical review manuscript JP-TR-2022-282854X ("Heksor: the CNS substrate of an adaptive behavior," by Wolpaw and Kamesar) and are resubmitting it to the Journal of Physiology. Your comments and those of the referees have led to extremely extensive revisions that we believe have greatly improved the manuscript.

Here we include all the comments with our responses interdigitated in *Italics*. We hope you agree that the manuscript has been much improved, and we look forward to hearing from you.

Thank you and best regards,

Jon Wolpaw

EDITOR COMMENTS

Reviewing Editor:

This is an ambitious paper. The authors seek to provide "a new paradigm for understanding how adaptive behaviors are acquired and maintained". As acknowledged by all referees, the topic is important and of great intrinsic interest. The piece has several positive attributes, and in many respects it is scholarly and engaging. The potential offered to the scientific community by this new formulation does however depend on the extent to which it can offer new insights, provide a conceptual framework with which to generate novel experimental hypotheses, and advance knowledge in a manner that would not otherwise be possible. As you will discern, the referees differ in the degree to which they believe that the project satisfies these requirements. I also share the concern that the Heksor concept failed to advance my own understanding beyond that accrued from classical experimental approaches and various types of formal modelling. Perhaps I simply lack the capacity to derive the necessary understanding. This does however point to a problem that has been emphasised by the referees. The way in which the project offers something that is new and worthy of the attention of those working in the relevant fields is not yet obvious. This is not to say that the Heksor concept does not possess this capacity. It is rather that its presentation seems to require further development, to convince the target audience that any shift in thinking prompted by this approach will prove to be worthwhile.

We appreciate this circumspect assessment and its request for revisions that clarify the import of the manuscript. The reviews are most helpful in indicating the respects in which the manuscript is not clear and in guiding appropriate revisions. The many comments and questions have led to major changes in how we define the purpose of the paper, how we delineate exactly its new contribution, how we enunciate the new concepts, and how we illustrate their applicability and value. Most importantly, the reviews made it clear to us that the presentation in the manuscript obscured its purpose and raised unnecessary issues unrelated to the central message. Accordingly, we have endeavored in this revised manuscript to focus clearly, simply, and only on enunciating and addressing the critical question that is the entire reason for the paper: how do numerous adaptive behaviors share a ubiquitously plastic CNS? The extensive changes in the revised manuscript are described in our responses below. In accord with your summary assessment, we have taken care to highlight findings that illustrate the ability of the heksor and negotiated equilibrium concepts to explain experimental results that are surprising or inexplicable (lines 573-588, 589-609, 658-667, 685-717, 718-733, 734-751, 752-775). We have also indicated the potential reach of these new concepts by illustrating their ability to generate novel and testable hypotheses about several major topics in neuroscience (i.e., spontaneous neuronal activity (lines 932-948); muscle synergies (lines 957-977); homeostatic plasticity (lines 978-998)).

Senior Editor:

As you can see your article has attracted a lot of interest - both positive and negative. The RE and I are keen to see a response and revision of your article. The presentation requires a "root and branch" revision with a view to enhancing the clarity of exposition. It remains to be seen whether this would reduce or consolidate the dissent. When you are re-writing, please make the article short, with clear arguments that will appeal to a wide audience. I hope you are able to undertake this revision, as I think there is an opportunity, which at present is not realised. I look forward to your revised MS.

As indicated above, and in accord with the requests here, the primary goal of our revisions is simplicity and clarity of exposure, including emphasis on the ability of the new concepts to explain otherwise inexplicable results and to generate novel experimental hypotheses about important issues. We much appreciate the impetus and the specific guidance the referees have given us for doing this. As we think you will see, the revisions do encompass a bottom-up reformatting and streamlining of the presentation.

REFEREE COMMENTS

Referee #2:

This is a brilliant review on the highly important and hotly debated issue of the neurophysiological mechanisms of everyday behaviors. The authors present an in-depth historical analysis, explain the origins and meaning of the new term (heksor), and then review more recent studies demonstrating and exploring interactions among neural circuits underlying components of natural behaviors (heksors) leading to the notion of negotiated equilibrium. Overall, the authors make a very convincing case for the new term, its meaning, and its role in behaviors and their changes with development and practice. I have only a few small comments that the authors are free to use or to ignore. The paper is very good as it stands.

We appreciate this positive assessment, and the valuable comments and suggestions that follow.

On a number of occasions, the authors refer to Bernstein (1967). The more recently (2020) published translation of Bernstein's classic (1947) offers a more in-depth presentation of his ideas that seem rather directly related to some of the central notions in the review. In particular, Bernstein describes his "Level of Synergies" as composed on neural circuits with the purpose to ensure dynamical stability of salient performance variables (while allowing other variables to vary). This notion seems rather closely related to the concept of heksor as a network maintaining "the key features of the behavior, the features that make the behavior satisfactory". Further in the 1947/2020 book, Bernstein describes interactions among synergies that evolve with development and practice and help the person to avoid negative interference between them, which seem to be precursors of the concept of "negotiated equilibrium".

The definition of synergies accepted by the authors reflects only half of the definition introduced by Bernstein (1947/2020). In that book, synergies have two functions: (1) grouping variables (frequently studied in our times as reflected in the references cited in the paper) and (2) ensuring dynamical stability of action (less commonly studied). Whether the former function reflects a negotiated equilibrium among heksors is a very interesting question. Whether there are more important (basic) and less important (more volatile) heksors is another interesting question. Maybe the former define the grouping of elements, whereas the latter have to deal with already established groups.

As the Referee notes, Bernstein figures prominently in the original manuscript; and that prominence is increased in the revision (lines 773-775). As the manuscript makes clear, Bernstein's ideas presaged the entity we are now naming a heksor. In fact, the similarity is a major reason for our confidence in the new term; Bernstein and we started from very different places with very different perspectives and came by very different routes to essentially the same conclusion: "biodynamical structures that live and develop," that is, "heksors."

We are less familiar with his further thinking related to the problem of behaviors sharing the nervous system or his multi-level ideas about synergies. We have recently acquired the excellent new Latash translation of Bernstein's 1947 book "On the Construction of Movements" and look forward to incorporating its ideas into subsequent papers addressing specific aspects of the new concepts in more detail than possible or appropriate in this first paper.

The concept of "satisfactory behavior" seems to be rather closely related to the concept of satisficing introduced by Herbert Simon and the idea of "good enough" (in contrast to optimal) behaviors advanced by Jerry Loeb and others.

Yes, we agree. Loeb's idea is now mentioned in lines 195-196.

Heksors are sometimes described in the paper as entities that "produce behaviors". There is a nontrivial issue

of producing "same behaviors" in different external force fields (e.g., pointing or throwing a tennis ball to a partner during rotation in the centrifuge, cf. Lackner). Some variations in the force field probably do not require new heksors, whereas others do. The key word seems to be "destabilization" (cf. Bernstein). If salient performance variables are destabilized by the new force field, the behavior becomes qualitatively different and requiring a new heksor. This point may deserve an in-depth discussion.

This issue is now mentioned in lines 409-416; and it is illustrated with a related example in lines 489-502. We fear that an in-depth discussion would necessarily become very lengthy; and would be premature in this introductory presentation of the concepts. It is clearly a significant issue that subsequent papers need to address.

Another issue that seems to be relevant is the fact that the central nervous system, by its very design, cannot prescribe external mechanics and muscle activations independently of the (everchanging and unpredictable) external force field. This issue has been discussed in detail starting from Bernstein 1947/2020. So, neural networks have to operate with indirect variables (e.g., Feldman's lambdas), which then indirectly produce behaviors. What variables do heksors operate with within the suggested scheme and how do these variables translate into important behavioral variables? Maybe an explicit set of answers to these questions would be useful.

As now emphasized throughout, this manuscript focuses on a question that applies to all adaptive behaviors: how do they share a ubiquitously plastic CNS? As lines 527-538 indicate, the manuscript does not attempt nor presume to provide new understanding of how specific adaptive behaviors, such as locomotion, are produced. The question the Referee raises about indirect variables (e.g., Feldman's lambdas) applies to certain kinds of adaptive behaviors (e.g., reach-and-grasp). Its relevance to other adaptive behaviors (e.g., languages) is less clear, though these behaviors may well have their own equivalents of these indirect variables. This issue, while highly significant, is not directly relevant to the very focused purpose of this manuscript. Moreover, adequate discussion of it would necessarily be quite lengthy.

At the same time, as lines 432-443 indicate, it is important to note that the heksor and negotiated equilibrium concepts are fully compatible with existing models for how the substrate of an adaptive behavior actually goes about maintaining its behavior (e.g., how reach-and-grasp is actually accomplished). They are, for example, compatible with the proposed roles of indirect variables such as Feldman's lambdas. As the revised manuscript now emphasizes, the contribution of these two complementary new concepts is that they directly address what has now become a critically important problem: how do multiple behaviors share the neurons and synapses of a ubiquitously plastic CNS?

Referee #3:

I enjoyed reading this scholarly and engaging paper. It is clearly written by authors who have a deep intuition about neural plasticity and distributed processing in the motor system. I also enjoyed the interesting and eclectic mixture of etymology and neurophysiology.

We are glad that the Referee liked the manuscript.

There are a few points that I felt you could unpack for the general reader. Perhaps you could consider the following:

There is an extensive and compelling case made for the word heksor, which is carefully defined. It would be nice to balance this with an equivalent motivation of 'negotiated equilibrium'. In other words, spend a paragraph or two describing the etymology of negotiated equilibrium - providing a concise definition. The choice of the word equilibrium is interesting here. It reminds one of things like the Nash equilibrium in game theory. Alternatively, from the point of view of statistical physics, it speaks to the notion of a nonequilibrium steady-state. I mention this because there is an interesting connection between the exchange and synchronisation of multiple heksors and the self organisation implicit in nonequilibrium steady-state dynamics. The key thing here is that a nonequilibrium steady-state deals with systems that are open. In the present setting, this means the notion of a nonequilibrium steady-state accommodates the exchange of energy and information between heksors. Finally, the existence of a nonequilibrium steady-state implies an attracting set of ('virtuous') states to which the system will self organise. I'm not suggesting that you cover this in your paper; however, it would be nice to substantiate the concept of a negotiated equilibrium by providing some synonyms or homologues in

other fields (e.g., generalised synchrony, self organisation, dynamic pattern coordination, nonequilibrium steady-state, et cetera).

We appreciate these observations. They are largely new to us and go well beyond what we are at all qualified to address. Thus, we are relieved that the Referee is not asking us to do so. At the same time, the relevance of game theory has engaged us, and was in fact included in an earlier version of the manuscript. We agree that a fuller explication of what "equilibrium" means here is in order. Our intent agrees with the Reviewer's inference: we mean "equilibrium" in the sense of a Nash equilibrium, a state in which (in the healthy CNS) every heksor is satisfied and has no incentive to change (i.e., cannot gain by changing). This is now explained in lines 191-197. There we also indicate that the "negotiated equilibrium" we are defining is similar to what is termed in thermodynamics an "open-system nonequilibrium steady state."

A lot of the neurophysiological arguments rests upon the notion of down-conditioning. I think it would be useful for the general reader to include a brief description of the operant conditioning paradigm that produces down-conditioning. For example, a brief vignette or description of how you would implement the paradigm in the clinic.

Furthermore, what determines the difference between down and up-conditioning? Is it the scheduling of the conditioning or the nature of the conditioned and unconditioned stimuli?

The operant conditioning protocol, and the difference between the up-conditioning protocol and the down-conditioning protocol, are now described in lines 373-394. The description is in terms of the original implementation of the protocol: operant conditioning of the biceps spinal stretch reflex (SSR) in monkeys. As to the difference between up- and down-conditioning, it is simply the reward criterion: up-conditioning rewards SSRs that are above a criterion size (as measured by EMG) and down-conditioning rewards SSRs that are below a criterion size. The identical protocol was subsequently applied to the H-reflex, the electrical analog of the SSR. The only difference was that reward was based on H-reflex size rather than SSR size. The switch to the H-reflex eliminated the muscle spindle from the reflex arc and allowed the conditioning protocol to be implemented 24/7 in freely moving monkeys and rats. The greater focus on down-conditioning than up-conditioning developed when our first mechanistic studies showed that down- and up-conditioning are not mirror images of each other (they have different mechanisms) and gave simpler and clearer answers regarding the mechanisms of down-conditioning. (Actually, we now have a clearer answer for up-conditioning as well, but that is not yet published.) The clinical protocol is essentially identical to the protocol developed in animals, except that it is entirely non-invasive and it is not available 24/7. People compete three 1-hour sessions/week over 8-10 weeks of conditioning. SSR conditioning and H-reflex conditioning are similar to each other in course, magnitude, and other characteristics. Furthermore, as indicated in lines 386-394, the course, magnitude and other characteristics of SSR conditioning and H-reflex conditioning are similar in monkeys, rats, mice, and humans; and physiological and anatomical studies indicate that the responsible CNS substrate (i.e., the heksor) is also similar across species (for review: Thompson & Wolpaw (2014) and Wolpaw (2018) references in manuscript).

On line 279, you introduce the notion of a "continually plastic CNS". It might be useful to emphasise this point with a few quantitative examples. For example, cite a couple of studies of synaptic turnover, showing that synapses only persist for minutes or hours in (certain parts of) the brain and can be regarded as a dynamic flux or fluid of connectivity, especially in systems with high synaptic turnover.

Lines 116-123 now reference examples illustrating the wide range of time constants across the different mechanisms of plasticity, from minutes to weeks to decades.

On line 289, you talk about the heksor "changing the set of neurons and synapses that comprise it". When exemplifying this using the H-reflex, it might be useful to introduce the notion of diaschisis (Greek for shocked throughout). In particular, the changing connectivity in one region - induced by changes in another - has been studied under the notion of a dynamic diaschisis [1, 2]. This might be closely related to the notions of primary and secondary plasticity introduced on line 646.

This useful suggestion has been adopted in lines 831-841 of the revised manuscript.

On line at 710, you talk about the new heksor interfering with the old. It might be nice to reference a couple of

papers that consider 'catastrophic interference' and 'catastrophic forgetting', which are cardinal concepts in computational neuroscience - and represent an outstanding challenge that speaks to the ability of heksors to maintain their negotiated equilibrium.

This very relevant suggestion is now implemented on lines 139-147. It helps to make our case stronger by illustrating the difficulty of the sharing problem that is the focus of the manuscript.

I hope that these comments help should any revision be required.

1. Price, C.J., et al., Dynamic diaschisis: anatomically remote and context-sensitive human brain lesions. *J Cogn Neurosci*, 2001. 13(4): p. 419-29.
2. Carrera, E. and G. Tononi, Diaschisis: past, present, future. *Brain*, 2014. 137(Pt 9): p. 2408-22.

We much appreciate the Reviewer's valuable suggestions. They have substantially strengthened the manuscript.

Referee #4:

Everything that I have to say about this article is summarized in my 11 page referee report. [ATTACHED]

The Referee's 11 pages of criticisms have essentially one message: models have already explained all the aspects of CNS function addressed in this manuscript and have done so much better than this manuscript. This message rests on the assumption that the ability to imitate CNS function indicates understanding of how the CNS actually functions. As explicated in the following paragraphs, this assumption is simply not correct.

*We are very much aware of the extensive recent efforts to model brain functions as the products of adaptive neural networks (ANNs). ANNs continue a long tradition of brain models that go back at least as far as Parmenides in the 6th century BC. Aspects of their development over the past 2600 years are reviewed in Wolpaw, *Behav. & Cogn. Neurosci. Revs.* 1:130-163, 2002. In this long progression, each historical period has endeavored to account for brain function in terms of its current most advanced technology. For example, in the 17th century, when hydraulics were high technology, Descartes was impressed by the life-like movements of human figures activated by hydraulics, and went on to develop an hydraulic model of nervous system function (Fancher & Rutherford, *Pioneers of Psychology, Fourth Edition*, Norton & Company, New York & London, 2012)). The ANN models that the Referee describes are certainly considerably advanced from those of past centuries; they can reproduce in detail many CNS behaviors. Nevertheless, they are still simply models, and they have the limitations of all such models.*

First, these models imitate CNS function, they do not explain how it actually occurs. The hydraulic models of the 17th century imitated normal movement, but they did not explain how the CNS produces movement. At present, computer programs can play chess as well and indeed better than any human, but they do not explain how the brain of a grandmaster actually functions. And, most notably, while the Referee stresses the many ways in which ART can imitate actual CNS function, this impressive ability to imitate has not translated into novel and effective new methods for addressing the deficits in CNS function produced by injury or disease. Doing this requires more than the capacity to imitate CNS function, it requires understanding of how the CNS actually functions. If the many model-based processes described by the reviewer do in fact occur in the CNS, the models should certainly have generated transformative therapeutic approaches. This has not occurred. In contrast, the heksor-based paradigm introduced in this manuscript has already led to novel therapeutic protocols that can enhance restoration of functions impaired by spinal cord injury or other neuromuscular disorders (i.e., lines 668-806, Figs. 4&5). Furthermore, these protocols produce surprising results that are readily explained by the new paradigm based on the heksor and negotiated equilibrium concepts (e.g., lines 573-588, 589-609, 658-667, 685-717, 718-733, 734-751, 752-775).

*Second, models are based only on the knowledge that the modeler has. For models of CNS function, this has always been a major problem, and it remains so today. Each year continues to bring remarkable new observations that reveal major previously unknown CNS phenomena. The recent realization of the importance of the colonic microbiome in determining the impact of brain injury on brain function is one example (For review: Cryan et al., *Lancet Neurology*. 19(2):179-194, 2020; Tremlett et al., *Annals of Neurology*. 81(3):369-382, 2017). Perhaps even more striking, for its overturning of a centuries-long major assumption about brain function, is the recent article of Lukoyanov, Watanabe, Carvalho, et al. *eLife* 2021;10:e65247. DOI: <https://doi.org/10.7554/eLife.65247> (See also the accompanying commentary of*

Wolpaw and Carp. eLife 2021;10:e72048. DOI: <https://doi.org/10.7554/eLife.72048>). The frequency of such remarkable new discoveries shows no signs of diminishing and may actually be increasing. Those in the field, that is, those who study CNS function rather than simply model it, recognize that they remain very far from understanding it. And, while the models Referee #4 puts forward are certainly more advanced than the hydraulic models of the 17th century, the advance is probably modest compared to what remains unknown. The recent article of Hennig et al. (Neuron 109:3720-3735, 2021) provides a further view of the limitations of models for understanding CNS function.

In sum, we think that the Referee's dissatisfaction with this manuscript reflects a difference in goal. The Referee's goal is to develop a model that imitates behavior. The model is completely defined by the modeler, all its capabilities are known, and thus its outputs are wholly predictable. This is therefore an entirely tractable endeavor. It may have important practical results; for example, it may guide advances in numerous fields from climate prediction to systems engineering. At the same time, it may or may not provide insight into the operation of the CNS.

In contrast, the goal of this paper, and of much of neuroscience research in general, is very different; it is to explain how the CNS actually produces behavior. This is a more difficult and often frustrating endeavor. CNS anatomy and physiology are imperfectly understood, CNS capabilities and limitations are not fully known much less understood, experiments are laborious and time-consuming, and their results often messy. Concepts such as "heksor" and "negotiated equilibrium" emerge gradually and require extensive testing that they may or may not survive. On the other hand, this endeavor may yield new understanding of actual CNS function and new more effective methods for restoring impaired functions. Indeed, even at this early date, the heksor and negotiated equilibrium concepts appear able to illuminate mechanisms underlying normal and abnormal CNS function, and to guide new approaches to restoring function.

Referee #5:

This manuscript presents an outline of what the authors claim is a "new paradigm" for understanding adaptive skeletomotor behaviour (at least that's how I read it, I saw no evidence for inclusion of other kinds of behaviour).

The manuscript addresses in several places the relevance of the new paradigm to overtly cognitive behaviors, such as language. This is most evident in lines 466-488 and in Figures 2c and 2d. Moreover, experiments that could test in more detail this relevance to cognitive behaviors are discussed in lines 902-917.

It identifies and defines, in general terms, a new conceptual CNS structure (if 'structure' is the right word), which is given the name 'heksor'. An historical review is provided that provides a background for the development of the new paradigm and the basis for the new heksor concept. It then reviews some further recent evidence for the existence, explanatory validity and wide applicability of this kind of entity, drawing on the first author's previous notion of 'negotiated equilibrium' described in an earlier review for The Journal of Physiology.

There is no doubt that this article deals with several important and interesting ideas that are fundamental to the understanding of motor behaviour. In addition, the background review provides an intriguing discussion of a number of empirical findings. However, I found it hard to see exactly what was new in the authors' paradigm and what novel insights it offers. Perhaps this is because I did not properly understand the conceptual framework being developed. If this is the case, the comments that follow should be taken as pointers to how the authors' message might better be communicated. If the authors have indeed some profound new concepts to offer, then clarity of exposition is vital if they are to have the impact they deserve.

The comments below concern conceptual matters rather than specifics of exposition, data interpretation or typographical errors. I see the conceptual issues as things that will require major revisions to sort out and I would not expect a revised manuscript to retain much of the text in its current form. For this reason, I offer no comments on non-conceptual matters.

We understand the critical importance of clarity. Thus, we very much appreciate the many ways in which the Referee's comments have shown us where we have failed to be clear. As will be evident in our responses to the specific comments below, we have endeavored throughout our revisions to simplify and focus the presentation. In the service of this endeavor, we now emphasize throughout the principal purpose of the manuscript: to address the question of how numerous adaptive behaviors share the ubiquitously

plastic CNS. As now emphasized in lines 527-538, the manuscript's purpose is not to advance understanding of how specific adaptive behaviors, such as locomotion, are produced. As the Referee indicates (and as lines 527-538 note), this is, for locomotion alone, an extremely complex issue that has motivated decades of valuable research and generated important knowledge. Rather, the purpose of this paper is to advance understanding of how adaptive behaviors are maintained in the ubiquitously plastic CNS, that is, to begin to explain how numerous behaviors share the CNS. This issue has received relatively little attention in the literature to date; its enormous scientific and clinical importance has been barely acknowledged, much less effectively addressed. This paper seeks to change that.

We think that our revisions correct misunderstandings that the previous version created about the paper's purpose and about other issues as well, and that they thereby make the import of the manuscript clear. We hope that the Referee agrees. As noted above, a major aspect of the revised manuscript is that it highlights the ability of the new concepts to explain surprising and often inexplicable results (e.g., lines 573-588, 589-609, 658-667, 685-717, 718-733, 734-751, 752-775). It also illustrates their ability to generate novel and testable hypotheses about several topics that are now of great interest (i.e., spontaneous neuronal activity (lines 932-948), muscle synergies (lines 957-977), homeostatic plasticity (978-998)).

Specific comments

(1) 'Adaptive' behaviour

The notion of 'adaptive behaviour' is fundamental to the discussion and is introduced on page 3, but it is not defined. Perhaps the authors think that this is too obvious to bother with. In lines 75-76, it is stated that an adaptive behaviour has a 'standard' by which it can be evaluated as 'good' or 'bad'. On the face of it, this seems to be fairly straightforward and appears to mean that adaptive behaviour is goal-directed behaviour. After all, a behavioural goal is a 'standard' by which success (good) and failure (bad) can be evaluated (of course, a given behaviour may have several 'goals' since criteria like 'minimize energy expenditure' are also a goals).

The problem with 'goal-directedness' in biological systems is determining what the goals are: how can it be done? This is a question that is fundamental in behavioural ecology and ethology, but the authors do not address it - perhaps they are of the opinion that this is too straightforward to be worthy of comment. But it isn't and the authors' treatment of H-reflexes would seem to support this.

As defined in lines 76-83, an adaptive behavior is defined as a behavior that serves a goal, or need, of the individual; thus, for flexion-withdrawal the goal is to remove the finger from the hot stove, and for locomotion the goal is to move across the room. And, as described in those lines, an adaptive behavior has attributes, or key features, that determine how well it serves that goal. For the adaptive behavior of locomotion, minimal energy expenditure is a key feature: locomotion that is energy efficient is better than locomotion that is energy inefficient. Energy efficiency is not a goal that is served by locomotion. If it were, the best solution would be to not move at all. The goal of locomotion, the need it serves, is to move across the room. Minimal energy expenditure is a desirable attribute, a key feature, of locomotion.

At the same time, we acknowledge the fundamental difficulty of the goal question, for behavioural ecology and ethology specifically, and for many other disciplines. It devolves ultimately to defining the nature of the good, which has engaged philosophers, theologians, and others since well before Aristotle. For our present purposes, we believe it is possible for most standard adaptive behaviors to define a reasonably clear goal that is acceptable to all (e.g., to move the finger off the hot stove, to move across the room). Ultimately, the goal might be defined by the capabilities (i.e., the affordances) provided by the set of key features that the heksor maintains: that is, what does a behavior with that set of key features enable the individual to do? This consideration is of practical importance for neurorehabilitation. Different individuals, faced with the same disability, are likely to differ in which key features they most want to regain. Among people with impaired locomotion due to spinal cord injury, one might prioritize increased walking speed, another more symmetrical stepping, another better balance, another longer distance, etc.

The authors claim (e.g., lines 294-5) that the H-reflex is not an adaptive behaviour. This is (probably) true when the H-reflex is viewed as something that doesn't exist in nature but is instead a consequence or artifact of the spinal circuitry that connects muscle spindles to motoneurons. From this point of view, the H-reflex is without biological purpose. Is this what the authors are saying? I'm not sure since they aren't explicit, instead they emphasise that the H-reflex is the electrical analogue of the spinal muscle stretch reflex.

We originally used the H-reflex in this example because it is used in most current operant conditioning studies. We now realize that this choice produced confusion. In the revision, we use as an example the spinal stretch reflex (SSR) itself. Historically, the SSR was operantly conditioned before the H-reflex was. (We switched to the H-reflex because it eliminated the muscle spindle from the reflex arc and it enabled 24/7 conditioning in freely moving monkeys and rats.) The SSR does exist in nature, it is not merely an experimental creation. Extensive studies over the past 40 years show that SSR operant conditioning and H-reflex operant conditioning are very similar. Furthermore, as lines 386-394 now indicate, the course, magnitude and other characteristics of the conditioning phenomenon are similar in monkeys, rats, mice, and humans; and physiological and anatomical studies indicate that the responsible CNS substrate (i.e., the heksor) is also similar across species (for review: Thompson & Wolpaw (2014) and Wolpaw (2018) references in manuscript).

The stretch reflex is a goal-directed/adaptive behaviour of a sort. What is the goal of the reflex action? This is not a question that is easily answered, but the effect of the reflex is to increase the stiffness and damping properties of skeletal muscles, which has led to the hypothesis that the stretch reflex functions to give muscles the stiffness and damping properties necessary to support effective behaviours of other kinds: the stretch reflex is useful, it is an adaptive behaviour.

In terms of the adaptive behavior definition given here – a behavior that serves the needs of the individual – the spinal stretch reflex (SSR) is not an adaptive behavior. The SSR does contribute to a variety of adaptive behaviors; and it may do so by regulating muscle stiffness (as now mentioned in lines 367-368 and lines 531-534). But the SSR does not by itself serve a specific need of the individual. Nor does the SSR itself have an evaluative standard by which it can be rated good or bad. SSR size varies widely across people with perfectly normal CNS function; a large (or small) SSR is neither good nor bad. This revised manuscript tries to make this important point more clearly (lines 360-372).

What are the authors saying, therefore? I'm not sure, but it would help to be clear about it. If the H-reflex is to be viewed as an electrical artifact of underlying neuromuscular circuitry and not an adaptive behaviour, then it's not a very good example to be using in a discussion of adaptive behaviour. I found the discussion of how the H-reflex can be given a goal using instrumental conditioning to be rather unconvincing and limited in scope.

As noted above, the revised manuscript eliminates the confusion produced by using as an example the H-reflex, which is only a laboratory phenomenon. It uses instead the SSR, which is an actual behavior. As also noted above (and in lines 360-372), in normal life the SSR is not itself an adaptive behavior, it contributes to adaptive behaviors (e.g., by regulating muscle stiffness). The SSR operant conditioning protocol changes this by giving a reward that is entirely contingent on SSR size. If the SSR satisfies the criterion, it benefits the individual. The result is that the SSR becomes an adaptive behavior that is good when its size satisfies the size criterion and bad when it does not. The operant up-conditioning or down-conditioning protocol creates a new heksor with one key feature: an SSR larger (for up-conditioning) or smaller (for down-conditioning) than a criterion.

It is asserted that there are three insights concerning adaptive behaviours obtained over the last few decades: (i) even the simplest adaptive behaviours are learned (line 79). (ii) "... the CNS substrate of an adaptive behavior comprises a network of neurons and synapses that is widely distributed through the CNS" (lines 90-91). (iii) "... the network of neurons and synapses underlying an adaptive behavior changes through life" (lines 99-100).

In response to the Referee's concerns, and in order to simplify the presentation, avoid unnecessary controversy, and focus on the central purpose and message of the manuscript, these three insights are no longer described as such. They are described simply as background (lines 84-103 & 148-157), without any presumptions of their recency or overwhelming importance. We hope that this overall change will reduce the Referee's problems with the presentation.

I will discuss these three insights in turn.

Insight (i) - all adaptive behaviors are learned.

The problem with the authors' treatment of this matter is again partly definitional: what do they mean by

'learning'? They don't tell us, so perhaps they think this is a matter too obvious or trivial to be worthy of comment. If so, I beg to differ.

That most adaptive behaviors, including simple ones, are acquired through experience is now presented in lines 84-94 without any mention of learning. For the purposes of this manuscript, there is no need to address or resolve the controversial issue of what constitutes learning.

The authors emphasize 'simple' behaviours, by which they mean skeletomotor reflexes like the withdrawal reflex. They seem to be saying that such simple behaviours had been viewed as 'hardwired' into the nervous system, requiring no 'learning' to establish the underlying circuitry. Since the 1970s, this view has been shown to be wrong and that such reflexes (and possibly other kinds of simple behavior as well) are learned. There are two points that should be made here. The first is that the view that some behaviours are hardwired was never widespread and constantly came in for criticism whenever it was suggested, such critiques can be found in literature dating from at least the beginning of the 20th century.

In the 1930s, Konrad Lorenz and Niko Tinbergen famously argued for a version of the hardwired theory usually called 'instinct theory'. Instinctual behaviour is, "hereditarily determined ... part of the original constitution of the animal ... arises quite independently of the animal's experience and environment ... is distinct from acquired or learned behavior ... [and] develops fully-formed in animals which have been prevented from practicing it". This quote comes from Lehrman's (1953) critique of Lorenz and Tinbergen's 'theory'.

The essence of Lehrman's critique is that there is no such thing as an instinctual behaviour as defined by Lorenz and Tinbergen. He cites a number of findings from literature published between about 1920 and 1950 to support this contention. Many of the examples demonstrate not that the behaviours need to be learned (at least not according to any definition of learning I am aware of), but that activity of the muscles and neurons is essential for development of the circuitry underlying the behaviour. Surely this is true of any phenotype - for anything to develop, some kind of environment is required. In the simplest case, the environment provides the materials from which tissues are constructed, but other factors such as the force of gravity, interuterine space, presence of other developing foetuses, freedom to contract muscles and move limbs (and resistance/constraint on such motion) may be equally significant. Only to the extent to which such factors are the same across all individuals, would we be justified in speaking of genetically determined phenotypes. If they vary, then the phenotype is not genetically determined.

We acknowledge the relevance of these criticisms, and we have revised the manuscript accordingly. As noted above, we no longer present the importance of experience in shaping simple reflexes as a new insight. We present it simply as relevant background (lines 84-94).

In short, developing and maturing in interaction with the environment is not the same as learning, in my view anyway. What is the authors' view? If it's different, can they justify it?

There is clearly a difference between reflex behaviour that is learned through specific kinds of experience - such as, for example, conditional reflexes acquired as the result of Pavlovian conditioning procedure - and reflexes that develop through interaction with uterine or other pre-natal environments without such specific experiences, in some cases through simple exercise.

Actually, based mainly on the Schoenberg studies referenced in the manuscript (lines 84-94 & Fig. 1a), we do not believe that the pre- and early post-natal development of flexion withdrawal reflexes is fundamentally different from the learning of other reflex behaviors; that is, we do not believe that effective flexion withdrawal reflexes are simply the result of exercise. At the same time, we also believe that this is an issue that the Referee and we can safely agree to disagree on; its resolution is not directly relevant to this paper. Accordingly, and as noted above, we do not mention "learning" in the presentation of this background information; we mention only "experience."

In my opinion, the hard-wired behaviour notion the authors claim as having been believed in 1970 and only gradually abandoned under the weight of more recent empirical evidence, is a caricature, a straw-man erected only for the purpose of being dismantled. A suitably restricted historical review can be used to erect such a straw man, but it's unlikely to convince those of us who cast our nets into the wider scientific literature pre-1970.

As noted above, we now present the role of experience in shaping simple reflexes during early development simply as background information (lines 84-94).

In addition, we have moderated our presentation of the mid-20th century view that the adult CNS was largely hardwired. There was clinical and experimental evidence going back to the early 20th century and before that the hardwired CNS was a fiction. In fact, it was Anna DiGiorgio's 1920's findings of spinal cord plasticity (Arch Fisiol 27:518–580, 1929) that inspired our demonstration of SSR operant conditioning 50 years later. The revised manuscript now acknowledges such evidence (in lines 105-114). Nevertheless, as now indicated in lines 109-114, in the mid-20th century many prominent neuroscientists were in pursuit of the supposedly rare "modifiable synapse." In the same period, the Sherringer and Dykeman demonstration (J Comp Physiol Psychol 44, 52-62, 1951) of restoration of locomotion after spinal transection through training was essentially ignored (to be rediscovered 30 years later by the Rossignol and Edgerton groups). The assumption of the limited mechanisms and locations of CNS plasticity survives in the common assumption that memory is due entirely to synaptic plasticity; and in articles assuming that neuroplasticity is basically a cerebral phenomenon (e.g., Brain 134:1591–609 (2011), which drew a response (Brain (2012) doi: 10.1093/brain/aws017)). The hardwired belief was also responsible for the fact that 50 years ago neurorehabilitation research was widely considered to be a backwater – CNS injury was permanent, there was little to do but settle for what was left and build better wheelchairs. Now rehab research is among the most exciting and promising areas for both scientific and clinical research. This transformation is due to the recent recognition that the CNS is ubiquitously plastic through life and to the new availability of computer-based technology that can support complex real-time interactions with the damaged CNS so as to initiate and guide beneficial plasticity that restores useful function.

(3) Insight (ii), the substrate of adaptive behavior

The claim that the substrate is "a network of neurons and synapses that is widely distributed through the CNS" seems to me to be something that could hardly be less controversial as a general statement. Perhaps it might seem controversial from the point of view of the Sherringtonian concept of reflex. Sherrington introduced the notion of the 'simple reflex' at the beginning of the 20th century, but it is now often forgotten that this notion was not that of the reflex as observed behaviourally - it was, according to Sherrington, a 'fiction', a hypothetical entity introduced as a kind of basic component of reflex behaviour. Sherrington's simple reflex was an effort to identify the atoms from which observed natural reflex behaviour is constructed. Thus, the stretch reflex or the withdrawal reflex are not (in Sherrington's terms) simple reflexes - they are natural reflexes built of many reflex atoms, many 'simple reflexes'.

The circuit for a simple reflex is, of course, the reflex arc. The circuitry of a natural reflex must, by definition, be far more complex. Sherrington himself didn't get very far with describing what this circuitry might involve, but he did discuss the need for things like reciprocal inhibition arcs and the circuitry mediating interactions between competing and cooperating reflexes. These circuits were subsequently elucidated in greater detail and we now know quite a lot about the circuitry underlying the various components of, for example, the vestibular-ocular reflex (VOR) and its interactions with cervico-ocular reflexes, with cervico-collic and vestibulo-collic reflexes, as well as with reflex saccades and fusional vergence. The circuitry of some ocular reflex interactions, such as those between fusional vergence reflexes, pupillary and accommodation reflexes were hypothesized by Helmholtz in the 19th century.

A nicely instructive example of the substrate of a natural, functional reflex is the wiping reflex of the frog. This is a marvellously complex behaviour that is clearly goal-directed. So marvellous in fact, that the reflex as manifested by the headless (spinal) frog suggested to scientists in the 18th and 19th Centuries that the frog's spinal cord must be in possession of a 'soul'. Investigation of the circuitry mediating this 'soul' was conducted from the 1950s to the 1970s by scientists in the Soviet Union and was revealed as involving complex circuitry throughout the spinal cord that enabled the reflex to be instantly adapted to a wide variety of novel circumstances without any prior experience.

It seems to me to be inconceivable that anyone would mistake the reflex arc of introductory textbooks with the circuitry underlying natural reflexes like the withdrawal reflex, the VOR or the frog's wiping reflex - behaviours that flexibly adapt to changes in the conditions in which the responses are executed and that change with experience in numerous ways from habituation and sensitization to improvements in skill.

Needless to say, more 'sophisticated' voluntary behaviour has been viewed as involving complex distributed neural mechanisms for as long as people have speculated about how such behaviour is produced. Since most, if not all, such behaviour is viewed as being wholly or partly learned, the neural substrate has never been

considered to be anything other than "a distributed network of neurons and synapses that ... changes as needed to maintain the key features of the behavior." (lines 274-275). Thus, I cannot agree with the authors' claims that such an 'entity' is a "previously unrecognized" one. I for one have recognized it (or something closely equivalent) for a very long time (I don't care to admit just how long) and so have many of my colleagues. Curiously, unlike the authors, I have never thought it needed a special name and remain unconvinced of the necessity.

We are familiar with essentially all the material the Referee summarizes here (including the interesting post-Marshall Hall speculations of Pflüger and others that the spinal cord has a soul); and we do not disagree with the points made in these comments. Moreover, as noted above, we now describe the recognition that a plastic network underlies an adaptive behavior as background rather than as a new insight (lines 148-157).

These comments made us realize that we needed to explain more clearly exactly what we are trying to say. That is, we needed to focus on what is new in this manuscript. Thus, we have essentially replaced the paragraphs that included the old lines 274-275. We hope the Referee finds that the new lines 198-232 are clear, and that they are effective in their justification for a new name. What is new is not that there is a distributed and plastic network underlying an adaptive behavior. What is new here is that the recognition of ubiquitous CNS plasticity means that that network has a unique set of properties that enable it to fulfill its unique role in CNS function (lines 211-222). These properties support its internal operation and its interactions with other networks. Specifically, it has properties that enable it to retain the key features of its behavior and to produce that behavior through life; and it has properties that enable it to join with other networks in establishing what is essentially a Nash equilibrium in which all the adaptive behaviors that share the ubiquitously plastic CNS can maintain their behaviors and have no incentive to change (lines 191-197). As now indicated in lines 211-232, it is these properties that motivate and justify a name for the network.

(4) Insight (iii), the substrate changes through life

The authors assert that this insight is the "most consequential" (line 99), yet from my perspective it seems the least insightful and most obvious. In some ways, the notion that the substrate changes through life is trivial since it simply means that behaviour can change as a result of experience, which has been known to be true for voluntary behaviours for a lot longer than 50 years. Reflex behaviours have been known to change since at least Pavlov's day and it's been known since at least the mid 1970s that changes in 'simple' reflexes like the stretch reflex can involve regions of the nervous system outside of the circuits that mediate the reflex response itself. For example, the suppression of the autogenic stretch reflex in the gastrocnemius that occurs in response to repeated tilts of the surface on which a person stands was shown to involve circuitry beyond the autogenic reflex arcs by Nashner (1976). Curiously, in the 1960s and early 1970s it was quite widely believed that the changes in simple spinal reflex mechanisms was driven exclusively by circuits in the brain - the basic spinal mechanisms themselves (the reflex arcs) were not intrinsically capable of learning. An odd point of view when you consider that it was well known at the time that learning could take place in the simple nervous systems of invertebrates in circuitry far simpler than that of any vertebrate spinal cord (see Grau et al, 2006; Oakley, 1979) and that habituation and sensitization of withdrawal reflexes involved changes in the spinal circuits themselves (Groves & Thompson, 1970).

Again, the Referee's comments compel us to explain exactly what we are trying to say. We are familiar with the studies mentioned here and we agree that they show that the networks can change. As noted above, the manuscript no longer presents the ongoing plasticity in these networks as a new insight. It now focuses on exactly what is new (lines 198-222). Specifically, the recognition of ubiquitous CNS plasticity forces attention to the problem of how behaviors share this ubiquitously plastic CNS, and it thereby indicates that these plastic networks have unique properties. These properties enable the network to retain a set of key features and to maintain a behavior that has these key features. They also enable the network to interact with the many other networks that share the same neurons and synapses so as to maintain what is essentially a Nash equilibrium of CNS neuronal and synaptic properties in which all the networks are satisfied (i.e., are able to maintain their key features) and have no incentive to change (lines 191-197). The recognition of these unique properties motivates and justifies a name for the networks (lines 211-232).

Parenthetically, we particularly appreciate the Referee's emphasis on spinal plasticity. Unfortunately, the belief that the spinal cord cannot learn remains alive today, even among prominent neuroscientists who know better (e.g., Brain 134:1591-609 (2011), which drew a response (Brain (2012) doi:

10.1093/brain/aws017)). *This anachronism may reflect the continuing influence of Marshall Hall's originally innovative and transitionally important, but now pernicious distinction between the "True-spinal" (or "Excito-Motory") nervous system and the "Cerebral" (or "Sentient and Voluntary") nervous system.*

(5) Additional comments

It seems to me that the insights covered above have been around a lot longer than a mere 50 years. Granted, they have become more mainstream over the last few decades and the evidence base is now overwhelming, but the alternative hard-wired view was never as mainstream as the authors seem to believe - at least not in the behavioral sciences. Admittedly, the instinct theory of Lorenz and Tinbergen was influential during the 1940s and 50s, but it was already under pressure in the late 1940s and was being revised to incorporate the data on behavioural adaptability and learning in the 50s and 60s. The title of J. P. Hailman's Scientific American article of 1969 describing his body of work on feeding behaviour in gull chicks was provocatively entitled 'How an instinct is learned'.

It would be very surprising, therefore, given the antiquity of these insights and their widespread acceptance from the 1980s onwards, if there hadn't been efforts to understand in greater detail the neural mechanisms of adaptive behaviour along the lines the authors deem necessary. To my mind, there has been a lot of work on exactly this. The work on the mechanisms of the frog's wiping reflex provides one example, but there are many others. The work on the neural basis of rhythmic behaviour, particularly locomotion, is also very pertinent.

As noted above, the insights described in the original manuscript are no longer described as such. They are presented simply as background, without any presumptions of their recency or overwhelming importance. And, as also stressed above and in lines 527-538, the goal of this paper is not new understanding of the neural mechanisms of individual adaptive behaviors, such as locomotion. The goal is new understanding of how all these behaviors are maintained in a ubiquitously plastic CNS, that is, how they share the CNS. As noted above, this is a problem that has received little attention to date, and that the recent appreciation of the ubiquitous plasticity of the CNS through life has now made critically important. At the same time, it has also turned therapeutic goals previously thought impossible into realistic possibilities.

The circuitry underlying locomotor behaviour in species such as the lamprey is now well understood thanks, in large part, to the work of Sten Grillner and his colleagues. The circuitry involved seems to have exactly the character required by the insights (ii) and (iii) listed earlier, so I was surprised to find that none of the work of Grillner's group was described or cited, despite locomotor behaviour being discussed at several points in the manuscript (first on page 10). The cursory discussion of locomotor mechanisms on page 10 is framed in 'heksor' terms and seems not only extremely vague and generic, but also appears to have overlooked the possibility of hopping. Human beings are capable of dealing with the loss of one leg by hopping on the other one and other animals can also deal effectively with injury to or loss of one or more legs. Changes in behaviour that preserve locomotor functionality occur almost immediately (they do not need to be learned) and such adaptation can be readily explained by existing models of the underlying mechanisms.

Much of the discussion of the H-reflex and its 'natural' counterpart, the stretch reflex, also suffers from a failure to discuss alternatives. Many researchers have sought to understand the stretch reflex in terms of its role in other behaviours, both reflex and voluntary. Merton's servo hypothesis of 1953 is perhaps the first of these efforts, Anatol Feldman's lambda-model is a more recent development, though it appeared in an early form as early as 1966. In these cases, the reflex is a manifestation of a negative feedback type control mechanism - a more useful conceptualization than the rather difficult term 'reflex' (which is not an aid to understanding).

As described above, and now stated explicitly in lines 527-538 of the revised manuscript, this paper does not attempt nor presume to provide new understanding of the mechanisms of locomotion (or the mechanisms of hopping). Nor does it attempt to provide new understanding of how the stretch reflex contributes to one or another adaptive behavior. Rather, it seeks to advance understanding of how numerous adaptive behaviors share a ubiquitously plastic CNS. It is not trying to provide alternatives to Merton's servo hypothesis or Feldman's lambda model. As emphasized in lines 432-443, the heksor/negotiated equilibrium paradigm presented here is in principle compatible with these models because it is addressing a different question: how do adaptive behaviors share a ubiquitously plastic CNS? Locomotion figures prominently in the paper simply because it is very well suited for addressing this question experimentally. The reasons why it is well suited are described in lines 539-546.

I would like to make one last comment. This concerns the treatment of the concept of memory that begins at the bottom of page 10. The authors contrast a 'standard' view of memory with that associated with the heksor concept. This seems to me simply to be the long recognised distinction between declarative and non-declarative memory. A distinction known to be associated with different CNS networks as demonstrated in humans in the 1950s based on studies of the famous anterograde amnesic Henry Molaison (HM). Declarative memory (the non-heksor kind of memory described on page 11) is exactly the sort of memory that is not associated with motor behaviour and skill. The kind of memory involved in motor skill acquisition and underlying learned behaviours is a kind of non-declarative memory. The kind of non-declarative memory involved is one that is not separable from the mechanisms/circuitry underlying production: there is not memory plus the circuitry, the memory is part of the circuitry - it is realized in the strength and nature of the connections that exist in the circuit. This connectivity is changed by experience and so the behaviour changes. This, after all, is the kind of memory described by Kandel and colleagues in their studies of reflex habituation and sensitization and it applies to other forms of behavioural change and acquisition.

We now address this comment in lines 428-431 of the revised manuscript. As indicated there, the comparison we make in lines 417-427 is for procedural, or skill memory. We have also modified these lines to avoid the unintended implication that we are discussing declarative memory. Thus, in the procedural memory of the Kandel et al. example mentioned by the Referee, the memory is accessed when the stimulus activates the modified (e.g., habituated or sensitized) synapses. All the adaptive behaviors discussed in the paper, from spinal stretch reflexes and flexion withdrawal reflexes to complex athletics to languages, are skills, they are not declarative or episodic memories.

I was reminded of Karl Lashley's famous 'search for the engram' dating from the 1920s and summarized in his famous paper of 1950 (Lashley, 1950). His work was guided by the view prevalent at the beginning of the 20th Century that sensorimotor learning involves connections between posterior, sensory cortical areas and frontal motor areas. He found that excising cortical tissue so as to separate sensory and motor areas did not impair the ability of experimental animals (rats) to learn sensorimotor tasks. From these studies, he concluded that sensorimotor memory was very widely distributed throughout cortex and that any neuron within the tissue could be involved (principle of 'equipotentiality'). He also found that memory impairment was correlated not with the region of cortical damage, but with the amount of damage and concluded that all the areas acted together to form the engram (principle of 'mass action'). Although the evidence base for Lashley's conclusions came mainly from lesion studies of the cerebral cortex, his conclusions represent ideas clearly at odds with any simple notion of hard-wired sensorimotor circuitry and separable sensorimotor memory. They were, to some extent at least, validated by later studies of formal neural networks and distributed sensorimotor learning and memory.

We are familiar with Lashley's work and agree with the Referee's observations of their broad consistency with more recent findings. We also note that his work is also broadly consistent with the heksor concept. Stated most generally, the progressive removal of cortical tissue progressively reduces the options that individual heksors have for restoring their key features after each lesion, and it thereby reduces the options that heksors collectively have for restoring a satisfactory negotiated equilibrium.

Much of the later sections of the manuscript (from page 18 on), I found to be rather vague. It would help to make these section more specific, perhaps more model based. At present they seem to have been tacked on at the end in order to 'prove' the usefulness of the heksor concept. For this reason, I will not comment on them further.

The sections from pages 16-24 (lines 510-806) describe the evidence that led to the heksor and negotiated equilibrium concepts; thus, they are an essential component of the manuscript. In accord with the comments of the Editors (see above), these sections have been revised at multiple points to indicate the explanatory power of the heksor and negotiated equilibrium concepts. That is, the concepts can explain findings that are would otherwise be puzzling or inexplicable (e.g., lines 573-588, 589-609, 658-667, 685-717, 718-733, 734-751, 752-775). Furthermore, as noted above and in lines 527-546, these sections focus on the question addressed by the manuscript: how do multiple adaptive behaviors share the ubiquitously plastic CNS? They do not attempt to provide new understanding of how a specific adaptive behavior, such as locomotion, is produced.

The sections from pages 24-30 (lines 816-1016) were added after the Editors provided brief comments from their preliminary evaluation of an earlier version of the manuscript. They requested discussion of how the applicability of the heksor and negotiated equilibrium concepts to complex motor skills and overtly cognitive skills such as languages might be evaluated. And, as the Reviewing Editor requests above, these pages also illustrate the concepts' ability to generate novel and testable hypotheses about several topics that are currently of major interest (i.e., spontaneous neuronal activity (lines 932-948); muscle synergies (lines 957-977); homeostatic plasticity (lines 978-998)).

(6) Concluding remarks

Overall, despite the fact that the authors touch on a number of important, fundamental issues in skeletomotor control, I did not feel that I had learned anything new from reading the manuscript. The treatment of these issues was rather vague and couched in verbal, conceptual terms; formal models and detailed circuits were not presented or described. I was left asking how the authors' conceptual treatment relates to more formal, model-based approaches to the issues. It is not that such approaches are rare or of poor quality - quite the opposite.

The manuscript devotes itself to a restricted range of basic and reflex behaviours, but even here formal, model-based approaches are not referred to. I found this to be a glaring omission. In particular, model-based approaches to the role of spinal stretch reflexes are completely absent from the manuscript, despite a large amount of it being devoted to behavioural phenomena associated with these reflexes and their electrical analogue (the H-reflex).

As described above, the revised manuscript focuses explicitly throughout on a single and now critically important question: how do many adaptive behaviors share the ubiquitously plastic CNS? And the manuscript stresses (lines 527-546) that the paper's goal is not new understanding of the neural mechanisms of individual adaptive behaviors, or a new model of how specific behaviors are produced. As lines 432-443 indicate, the manuscript does not conflict with or attempt to replace current models. It is trying to begin to answer a different question. Its goal is new understanding of how all these behaviors are maintained in a ubiquitously plastic CNS, that is, how they share the CNS.

The specific adaptive behaviors that receive the most attention, mainly the stretch reflex (and its electrical analog the H-reflex), the TMS-evoked MEP, and locomotion receive that attention because they are very well suited for analyzing how adaptive behaviors share the CNS. The reasons why they are well suited are described on lines 517-526 & 539-546). At the same time, other more complex behaviors do receive attention, including overtly cognitive behaviors (lines 446-488). Moreover, strategies for determining whether the new concepts apply to how a wider variety of more complex behaviors share the CNS are described (lines 816-1016). This description was requested by the Editors in their original invitation to submit the paper as a topical review. They wanted us to make it possible for the reader to appreciate how the applicability of the concepts to complex motor skills and overtly cognitive skills such as languages might be evaluated.

If the authors are to convince the reader that they have something genuinely new to say, then they are going to have to explain how and why previous work has failed to address the issues or has failed to address them 'properly'. My impression is that previous work has addressed them, perhaps not successfully, but they have been addressed. In addition, I do not see that the authors are offering a genuinely new approach - the current text is too vague to be certain, however. Unfortunately, much of the manuscript is devoted to erecting a straw man and identifying a better evidenced alternative - this effort fails to reference important literature and seems to cherry pick research in order to establish the straw man's reality. I was not convinced, though I confess that I may have misunderstood. If so, others are likely to have similar problems.

References

- Grau, J.W. et al. (2006) Behavioral & Cognitive Neuroscience Reviews, 5, 1-48.
- Groves, P. & Thompson, R.F. (1970) Psychological Review, 77, 419-450
- Lashley, K.S. (1950) Symposiums of the Society of Experimental Biology, 4, 454-482
- Lehrman, D.S. (1953) Quarterly Review of Biology, 28, 337-363
- Nashner, L. (1976) Experimental Brain Research, 26, 59-72
- Oakley, D. (1979) in Oakley, D. & Plotkin, H. (Ed) Brain, Behaviour & Evolution. London: Methuen.

We appreciate the Referee's point that the original manuscript would raise similar problems for other readers. We believe that the extensive revisions, many of them guided by the Referee's comments, greatly

diminish this danger. Most importantly, it is evident that the original manuscript failed to state clearly and emphasize sufficiently the problem that it was trying to address. We believe the revision fixes this problem. The manuscript is not trying to provide new or better answers to issues concerning how specific adaptive behaviors are produced. Rather, it is attempting to address an issue that has received relatively little attention in the past and has now become of critical importance: how do many behaviors share a ubiquitously plastic CNS? As described in multiple places above, we have endeavored to make this focus clear throughout the revised manuscript. In the process, we have also eliminated the straw man criticized by the referee. We see now that the straw man was actually not needed in the presentation, and in fact obscured the critical question that the manuscript is intended to address.

We thank the Referee for the many extremely substantive comments; we believe that they have enabled us to make the paper substantially better. We hope that the Referee agrees.

Dear Dr Wolpaw,

Re: JP-TR-2022-283291X "Heksor: The CNS Substrate of an Adaptive Behavior" by Jonathan R. Wolpaw and Adam Kamesar

Thank you for submitting your Topical Review to The Journal of Physiology. It has been assessed by a Reviewing Editor and by 3 expert referees and I am pleased to tell you that it is considered to be acceptable for publication following satisfactory revision.

The reports are copied at the end of this email. Please address all of the points and incorporate all requested revisions, or explain in your Response to Referees why a change has not been made.

NEW POLICY: In order to improve the transparency of its peer review process The Journal of Physiology publishes online as supporting information the peer review history of all articles accepted for publication. Readers will have access to decision letters, including all Editors' comments and referee reports, for each version of the manuscript and any author responses to peer review comments. Referees can decide whether or not they wish to be named on the peer review history document.

I hope you will find the comments helpful and have no difficulty in revising your manuscript within 4 weeks.

Your revised manuscript should be submitted online using the links in Author Tasks Link Not Available. This link is to the Corresponding Author's own account, if this will cause any problems when submitting the revised version please contact us.

You should upload:

- A Word file of the complete text (including any Tables);
- An Abstract Figure, (with accompanying Legend in the article file)
- Each figure as a separate, high quality, file;
- A full Response to Referees;
- A copy of the manuscript with the changes highlighted.
- Author profile. A short biography (no more than 100 words for one author or 150 words in total for two authors) and a portrait photograph of the two leading authors on the paper. These should be uploaded, clearly labelled, with the manuscript submission. Any standard image format for the photograph is acceptable, but the resolution should be at least 300 dpi and preferably more.

- A 'Cover Art' file for consideration as the Issue's cover image;
- Appropriate Supporting Information (Video, audio or data set https://jp.msubmit.net/cgi-bin/main.plex?form_type=display_requirements#supp).

To create your 'Response to Referees' copy all the reports, including any comments from the Senior and Reviewing Editors into a Word, or similar, file and respond to each point in colour or CAPITALS. Upload this when you submit your revision.

I look forward to receiving your revised submission.

Yours sincerely,

Ian D. Forsythe
Deputy Editor-in-Chief
The Journal of Physiology
<https://jp.msubmit.net>
<http://jp.physoc.org>
The Physiological Society
Hodgkin Huxley House
30 Farringdon Lane
London, EC1R 3AW
UK
<http://www.physoc.org>
<http://journals.physoc.org>

EDITOR COMMENTS

Reviewing Editor:

The authors are to be commended on the positive manner in which they responded to the initial set of reviews. The referees share the view that the manuscript has been improved significantly as a consequence. In particular, the manner in which the undertaking may offer something worthy of attention has been made more transparent. These positive observations notwithstanding, there remain matters of concern. As emphasised by Referee #5 in a detailed commentary, the current treatment of model-based approaches to problems that are the focus of the current manuscript, appears partisan and perhaps even unduly dismissive. While it can be appreciated that the authors are keen to stress the perceived advantages of their analysis, it is by no means apparent that other (e.g., modelling) approaches are thereby rendered redundant, or even that they are less useful as means of advancing fundamental knowledge.

I would therefore ask that you consider carefully the reservations expressed by Referee #5 and revise the manuscript in a manner such that the contributions of this complementary literature, and its continuing practical utility, are acknowledged. Ideally, the benefits that might accrue from an inclusive route to knowledge formation, that draws upon several paradigms, would be made apparent.

Senior Editor:

It is good to see that your extensive and thoughtful revision has produced an interesting article. Although there are some minor issues outstanding, these can be straightforwardly resolved and will broaden the appeal of your article.

In this regard could you re-consider the last two sentences of your abstract? It is always true to say that more work needs to be done, but that is not really the conclusion of your article. Please adapt the abstract to come to a clear conclusion about your review topic.

I look forward to reading the final revision.

REFeree COMMENTS

Referee #2:

I was very positive about the original version of this review. The authors improved the manuscript, and it is now even better than the original version.

I have only a couple of small comments that the authors are free to address or ignore.

First, on a number of occasions, the authors emphasize that heksors change themselves continually. If the animal is not learning a new skill, what could be the reason for these changes? Do heksors change spontaneously? If so, are heksors highly unstable (as suggested by their spontaneous changes)?

Second, there are tasks of different degrees of compatibility. For example, learning how to play badminton makes it much harder to master tennis because of the "bad habit" of using one's wrist, much harder as compared to a seemingly irrelevant original skill (e.g., how to ride monocycle or how to play guitar). Does this mean that, to become easily compatible, heksors should be sufficiently different? Why?

Referee #3:

Many thanks for attending to my previous suggestions. And congratulations on a thoughtful contribution.

Referee #5:

The manuscript has been improved in a number of respects and it is now much clearer what its domain and goals are. The authors adequately resolve the historical matters that I took issue with. It is now clear that the fundamental question that the ms seeks to address concerns how a multitude of different adaptive behaviours is generated by a neural substrate that is (in part at least) shared between them and that possesses plasticity throughout its entire extent and lifespan (or very nearly so).

There are two significant issues that this question draws our attention to. First, the neural mechanisms that generate individual behaviours overlap with one another extensively. Thus, it is not that they merely share a common final pathway in the form of motoneurons (different behaviours share the same motoneurons and muscles), but that they share neural components throughout their entirety. Second, the neural substrate is plastic and changes in response to its activity, not merely during certain critical periods, but throughout the lifespan of the organism. This means that there is the potential for activity associated with one behaviour to alter the substrate and so alter subsequent performance of the many other behaviours that share that substrate, potentially in a negative way.

Three questions follow directly: (1) how does neural circuitry simultaneously support the generation of numerous different behaviours? (2) How are the changes in the neural substrate produced by experience related to one particular behaviour (and that improve or support that behaviour) prevented from having negative effects on other behaviours that share the substrate? In other words, how is the capacity for producing these other behaviours maintained? (3) How are the underlying mechanisms recruited and changed in the acquisition of new behaviours?

The notion of 'negotiated equilibrium' (previously discussed by the first author in JPhysiol) and 'heksors' are presented as conceptual tools that can help provide answers to these questions. Both these notions are conceptual (rather than formal like the concepts of equilibrium used in mathematics or physics). 'Negotiated equilibrium', for instance, doesn't seem to amount to more than the idea that through some kind of process taking place within the shared neural substrate, the essential aspects of different behaviours are maintained when changes due to experience associated with a particular occur. Thus, when the circuitry changes in response to experience with one behaviour, the circuitry changes in such a way that the other behaviors it is responsible for producing are maintained.

There doesn't seem to me to be much here except a restatement of the fact that the other behaviours are indeed maintained and that the mechanisms underlying production are distributed, overlapping and plastic - in effect, the authors state the problem and show that if the premises are true, then some of the ideas and concepts of the past are inadequate and that new ones will be needed. What is offered is conceptual and metaphorical rather than precisely formulated and mechanistic.

Even in the absence of mechanistic precision, the ms could be a valuable contribution. Just providing names for particular concepts and metaphors (i.e., 'heksor' and 'negotiated equilibrium') can be an important advance. However, it seems to me that the contribution will only be really valuable if the following things are true:

- (1) The old ideas described by the authors should represent widespread current thinking on how adaptive behaviour is generated.
- (2) There is a lack of recognition of the problems identified by the authors and/or an absence of any recognition that the older ideas are inadequate for addressing them.
- (3) The kinds of concepts introduced by the authors to address the problems are to some extent novel. In particular, these concepts have not yet been developed by others to the extent that more precise, mechanistic formulations exist in the literature.

Are these true? My view is as follows: regarding item (1), it may well be that in certain research domains, the older ideas are still mainstream. Whilst I don't think they are mainstream in all research domains, even in all those that fit under the general heading of 'adaptive sensorimotor behaviour', the mere fact that the first author - who has decades of research experience within relevant domains - has written this manuscript suggests that the old ideas must still be widely accepted by many. Regarding item (2), given that the old ideas are still prevalent, it must be true that their inadequacy is insufficiently well and

widely recognized, and that despite the evidence described in the ms, the problems being addressed are not as widely acknowledged as one might hope.

The above being the case, the manuscript is a potentially valuable contribution for those researchers who have overlooked or not adequately considered the issues and ideas discussed. This brings me to what I think is the only significant outstanding issue, which is connected to item (3) on my list.

Are the authors' concepts novel? I do not believe that they are, as I think I made clear in my first review. Similar kinds of ideas have been around for a while, some for many decades. I don't, however, see this a big issue given that the authors are offering an integrated treatment targeted at an audience who may be unaware of earlier work or at least unaware of its possible application to their research domain. Of more concern is the fact that there are precise, mechanistic formulations of the ideas in the extant literature, contrary to what the authors claim. These may not be entirely to the authors' taste, but that hardly invalidates them or justifies their omission. In fact, merely by existing they render the authors' purely conceptual/metaphorical treatment inadequate.

The authors' attitude to formal/mechanistic treatments is summarized not in the article itself, but in their telling response to the 4th referee. As someone originally trained in the physical sciences, I found much of this response to be bizarre since you could advance a similar set of criticisms against models and theories in physics, in which context they would (I think we can all agree) be absurd. A main point of the authors is that models of brain and behaviour seek merely to imitate - they are elaborate exercises in data fitting. The goal of the modeller, according to the authors, is to imitate behaviour. The authors' goal, in contrast, is to explain how the CNS actually generates behaviour. One could level the same complaint against Newton, Einstein and the architects of quantum theory. Their goal, you could say, is to imitate nature with their darned equations, whereas the goal of the true scientist is to explain how nature actually does the things it does. I am unmoved by such objections and I was equally unmoved by the authors' claims about the lowly imitative aims of the neural modeller and the loftier explanatory aims of empirical neuroscientists.

The authors other 'argument' concerning the continuous accrual of new facts rendering old models irrelevant is equally dubious. You could level the same criticism at Newton or Maxwell, for their equations may have imitated data available when they formulated their theories, but new findings have rendered them either worthless or only useful for certain technological applications. This is not how they are viewed in the physical sciences.

In the present context, models do not merely imitate behaviour, they also make predictions of new phenomena and of quantitative details. But not only that, they provide examples of the kind of mechanisms that solve particular kinds of problem. Take, for example, the problem of instantiating multiple mechanisms or entities within the same distributed substrate, a substrate that is continually changing as a result of experience. This is a well-known problem sometimes referred to as the stability-plasticity dilemma (a term due to Stephen Grossberg, see his scholarpedia entry: http://www.scholarpedia.org/article/Adaptive_resonance_theory). It has been addressed in detail in the literature on artificial neural networks and researchers have demonstrated what is needed for a solution in networks of interacting elements. These formal/mechanistic solutions are not merely 'imitating' behaviour - they are giving us important insights into the properties that are needed in order for multiple memories or other entities to be realized within the same shared network of elements.

I am sure that the 4th referee will provide a more comprehensive rebuttal of what the authors say in their response letter, so I won't go on. I will, however, briefly note that the authors are guilty of some double-standards when they appeal to formal, model-based concepts like Nash Equilibria and stable states in systems far from thermodynamic equilibrium. It seems to be a bit disingenuous to appeal to concepts that have strict formal treatments to justify loosely specified, verbally stated ideas, whilst at the same time denying that formal approaches have anything useful to offer.

Overall, I am ready to accept that the revised manuscript has something to offer certain groups (the positive comments of two of the referees can be taken as additional evidence of that) and so it has the potential to be a useful contribution. However, I think the authors' refusal to discuss model-based approaches or even to acknowledge that such approaches to the problems exist is unhelpful and misleading. My view is that this should somehow be sorted out - if that means that the authors must ultimately take less credit for the ideas they describe, then so be it.

REQUIRED ITEMS:

-Your MS must include a complete "Additional information section" with the following 4 headings and content:

Competing Interests: A statement regarding competing interests. If there are no competing interests, a statement to this effect must be included. All authors should disclose any conflict of interest in accordance with journal policy.

Author contributions: Each author should take responsibility for a particular section of the study and have contributed to writing the paper. Acquisition of funding, administrative support or the collection of data alone does not justify authorship; these contributions to the study should be listed in the Acknowledgements. Additional information such as 'X and Y have contributed equally to this work' may be added as a footnote on the title page.

It must be stated that all authors approved the final version of the manuscript and that all persons designated as authors qualify for authorship, and all those who qualify for authorship are listed.

Funding: Authors must indicate all sources of funding, including grant numbers. If authors have not received funding, this must be stated.

It is the responsibility of authors funded by RCUK to adhere to their policy regarding funding sources and underlying research material. The policy requires funding information to be included within the acknowledgement section of a paper. Guidance on how to acknowledge funding information is provided by the Research Information Network. The policy also requires all research papers, if applicable, to include a statement on how any underlying research materials, such as data, samples or models, can be accessed. However, the policy does not require that the data must be made open. If there are considered to be good or compelling reasons to protect access to the data, for example commercial confidentiality or legitimate sensitivities around data derived from potentially identifiable human participants, these should be included in the statement.

Acknowledgements: Acknowledgements should be the minimum consistent with courtesy. The wording of acknowledgements of scientific assistance or advice must have been seen and approved by the persons concerned. This section should not include details of funding.

-It is the authors' responsibility to obtain any necessary permissions to reproduce previously published material
https://jp.msubmit.net/cgi-bin/main.plex?form_type=display_requirements#use

-The Journal of Physiology is pleased to fund authors of provisionally accepted papers to use the Journal's premium BioRender site to create high resolution schematic figures. Follow this link <https://app.biorender.com/portal/jphysiol> and enter your details and ensure you enter the manuscript number JP-TR-2022-283291X to be directed to enter our premium site. Select a figure type when creating the figure so The Journal of Physiology logo appears. When you have completed your figure(s) download and then upload as the figure file(s) for your revised submission. If you choose not to take up this offer we require figures to be of similar quality and resolution. If you are opting out of this service to authors, state this in the Comments section on the Detailed Information page of the submission form.

END OF COMMENTS

1st Confidential Review

10-May-2022

Dear Journal of Physiology Editors,

As you requested, we have now further revised the topical review manuscript JP-TR-2022-282854X ("Heksor: the CNS substrate of an adaptive behavior," by Wolpaw and Kamesar) and are resubmitting it to the Journal of Physiology. Your comments and those of the referees have led to further revisions that we believe have further improved the manuscript.

Here we include all the comments with our responses interdigitated in *green Italics*. We hope you agree that the manuscript has been further improved, and we look forward to hearing from you.

Thank you and best regards,

Jon Wolpaw

EDITOR COMMENTS

Reviewing Editor:

The authors are to be commended on the positive manner in which they responded to the initial set of reviews. The referees share the view that the manuscript has been improved significantly as a consequence. In particular, the manner in which the undertaking may offer something worthy of attention has been made more transparent. These positive observations notwithstanding, there remain matters of concern. As emphasised by Referee #5 in a detailed commentary, the current treatment of model-based approaches to problems that are the focus of the current manuscript, appears partisan and perhaps even unduly dismissive. While it can be appreciated that the authors are keen to stress the perceived advantages of their analysis, it is by no means apparent that other (e.g., modelling) approaches are thereby rendered redundant, or even that they are less useful as means of advancing fundamental knowledge.

I would therefore ask that you consider carefully the reservations expressed by Referee #5 and revise the manuscript in a manner such that the contributions of this complementary literature, and its continuing practical utility, are acknowledged. Ideally, the benefits that might accrue from an inclusive route to knowledge formation, that draws upon several paradigms, would be made apparent.

We acknowledge that our response to Referee #4's comments was a less than ideally politic and thoughtful response to the Referee's categorical dismissal of our manuscript.

That said, we would like to emphasize that the manuscript itself repeatedly recognizes and discusses the valuable contributions of models and modeling studies (lines 140-149; 211-217; 452-463; 975-982; 1025-1042). Our problem with Referee #4's position was its assumption that successful modeling alone is all that is necessary to understand CNS function; that once a model can accurately imitate the CNS, CNS function is adequately understood and no further efforts are needed. We cannot agree with that position. At the same time, we certainly agree that modeling of various kinds is an important aspect of studies aimed at defining how the CNS functions, and this agreement is reflected in the manuscript.

Senior Editor:

It is good to see that your extensive and thoughtful revision has produced an interesting article. Although there are some minor issues outstanding, these can be straightforwardly resolved and will broaden the appeal of your article.

In this regard could you re-consider the last two sentences of your abstract? It is always true to say that more work needs to be done, but that is not really the conclusion of your article. Please adapt the abstract to come to a clear conclusion about your review topic.

I look forward to reading the final revision.

We appreciate the Editor's positive comments; we have revised the last sentences of the abstract as requested (lines 39-44).

REFEREE COMMENTS

Referee #2:

I was very positive about the original version of this review. The authors improved the manuscript, and it is now even better than the original version.

I have only a couple of small comments that the authors are free to address or ignore.

First, on a number of occasions, the authors emphasize that heksors change themselves continually. If the animal is not learning a new skill, what could be the reason for these changes? Do heksors change spontaneously? If so, are heksors highly unstable (as suggested by their spontaneous changes)?

We believe that heksors change continually (or at least very frequently) in response to changes in other heksors, in response to growth, aging, and other life events, and in response to injury or disease. This is indicated in lines 146-149 and elsewhere.

Second, there are tasks of different degrees of compatibility. For example, learning how to play badminton makes it much harder to master tennis because of the "bad habit" of using one's wrist, much harder as compared to a seemingly irrelevant original skill (e.g., how to ride monocycle or how to play guitar). Does this mean that, to become easily compatible, heksors should be sufficiently different? Why?

Actually, this badminton/tennis example was included in an earlier version of the manuscript. Their overall similarity could certainly make their dissimilarities a problem for the creation of the second heksor. At the same time, their overall similarity might ultimately become an advantage; it might lead to a composite heksor that could switch between the two skills and would constitute an alliance that might also give them an advantage in negotiations with other heksors. This was discussed in an earlier version of the manuscript that considered in more detail the possible wider relevance of game theory to the heksor and negotiated equilibrium concepts.

Referee #3:

Many thanks for attending to my previous suggestions. And congratulations on a thoughtful contribution.

We thank the Referee for the useful suggestions and much appreciate this approval.

Referee #5:

The manuscript has been improved in a number of respects and it is now much clearer what its domain and goals are. The authors adequately resolve the historical matters that I took issue with. It is now clear that the fundamental question that the ms seeks to address concerns how a multitude of different adaptive behaviours is generated by a neural substrate that is (in part at least) shared between them and that possesses plasticity throughout its entire extent and lifespan (or very nearly so).

There are two significant issues that this question draws our attention to. First, the neural mechanisms that generate individual behaviours overlap with one another extensively. Thus, it is not that they merely share a common final pathway in the form of motoneurons (different behaviours share the same motoneurons and muscles), but that they share neural components throughout their entirety. Second, the neural substrate is plastic and changes in response to its activity, not merely during certain critical periods, but throughout the lifespan of the organism. This means that there is the potential for activity associated with one behaviour to alter the substrate and so alter subsequent performance of the many other behaviours that share that substrate, potentially in a negative way.

Three questions follow directly: (1) how does neural circuitry simultaneously support the generation of numerous different behaviours? (2) How are the changes in the neural substrate produced by experience related to one particular behaviour (and that improve or support that behaviour) prevented from having negative effects on other behaviours that share the substrate? In other words, how is the capacity for producing these other behaviours maintained? (3) How are the underlying mechanisms recruited and changed in the acquisition of new behaviours?

The notion of 'negotiated equilibrium' (previously discussed by the first author in JPhysiol) and 'heksors' are presented as conceptual tools that can help provide answers to these questions. Both these notions are conceptual (rather than formal like the concepts of equilibrium used in mathematics or physics). 'Negotiated equilibrium', for instance, doesn't seem to amount to more than the idea that through some kind of process taking place within the shared neural substrate, the essential aspects of different behaviours are maintained when changes due to experience associated with a particular occur. Thus, when the circuitry changes in

response to experience with one behaviour, the circuitry changes in such a way that the other behaviors it is responsible for producing are maintained.

There doesn't seem to me to be much here except a restatement of the fact that the other behaviours are indeed maintained and that the mechanisms underlying production are distributed, overlapping and plastic - in effect, the authors state the problem and show that if the premises are true, then some of the ideas and concepts of the past are inadequate and that new ones will be needed. What is offered is conceptual and metaphorical rather than precisely formulated and mechanistic.

Even in the absence of mechanistic precision, the ms could be a valuable contribution. Just providing names for particular concepts and metaphors (i.e., 'heksor' and 'negotiated equilibrium') can be an important advance.

The Referee's summary of the problem addressed by the manuscript is very accurate. We thank the Referee for earlier comments that caused us to simplify and clarify the manuscript.

At the same time, we would submit that a "heksor" as defined in our paper is not merely a concept and is certainly not a metaphor, any more than a "synapse" is merely a concept or a metaphor. A synapse is a connection between two neurons that has specific unique properties; in the same way, a heksor is a network of neurons and synapses that has specific unique properties. A synapse is an entity that can be pointed at; and so is a heksor (see Figs. 1b and 1c, which summarize what is currently known about the structure of two heksors). Each term also denotes a concept that is defined by the entity's unique properties. As understanding of these properties grows, the concept evolves. At present the concept of a synapse is far more advanced than the concept of a heksor, but this should change as the heksor entity is explored. The latter parts of the manuscript describe some promising ways to begin that exploration.

"Negotiated equilibrium" is not a metaphor and is not merely a concept. It is a name for a physical process; and a concept of what that process is. The etiology and meaning of this term is now explicated much more fully in the revised manuscript (lines 160-252). We thank the Referee for giving us the impetus for this important revision. As the Referee says, the term acknowledges the necessary existence of a process that supports the maintenance of all adaptive behaviors in a ubiquitously plastic CNS. But the term does more than that. It presents a concept of what that process is. To the best of our knowledge, this concept first appeared in Wolpaw (2010, 2018). We believe that that was the first coherent and comprehensive statement of such a concept in a form that can be easily understood and used. The individual action of each heksor is reasonably well understood from the studies of recent decades (lines 150-159). As to their collective action, the concept hypothesizes that it comprises interactions among the heksors that produce a Nash equilibrium, hence the term negotiated equilibrium. As indicated in lines 218-230, this heksor-based concept of the process through which adaptive behaviors are maintained in the ubiquitously plastic CNS starts from what is already known about how an individual heksor maintains its behavior (i.e., lines 150-159); thus, it starts from experimental evidence, unlike possible alternatives (e.g., a central executive).

However, it seems to me that the contribution will only be really valuable if the following things are true:

- (1) The old ideas described by the authors should represent widespread current thinking on how adaptive behaviour is generated.
- (2) There is a lack of recognition of the problems identified by the authors and/or an absence of any recognition that the older ideas are inadequate for addressing them.
- (3) The kinds of concepts introduced by the authors to address the problems are to some extent novel. In particular, these concepts have not yet been developed by others to the extent that more precise, mechanistic formulations exist in the literature.

Are these true? My view is as follows: regarding item (1), it may well be that in certain research domains, the older ideas are still mainstream. Whilst I don't think they are mainstream in all research domains, even in all those that fit under the general heading of 'adaptive sensorimotor behaviour', the mere fact that the first author - who has decades of research experience within relevant domains - has written this manuscript suggests that the old ideas must still be widely accepted by many. Regarding item (2), given that the old ideas are still prevalent, it must be true that their inadequacy is insufficiently well and widely recognized, and that despite the evidence described in the ms, the problems being addressed are not as widely acknowledged as one might hope.

The above being the case, the manuscript is a potentially valuable contribution for those researchers who have overlooked or not adequately considered the issues and ideas discussed. This brings me to what I think is the only significant outstanding issue, which is connected to item (3) on my list.

Are the authors' concepts novel? I do not believe that they are, as I think I made clear in my first review. Similar kinds of ideas have been around for a while, some for many decades. I don't, however, see this a big issue given that the authors are offering an integrated treatment targeted at an audience who may be unaware of earlier work or at least unaware of its possible application to their research domain. Of more concern is the fact that there are precise, mechanistic formulations of the ideas in the extant literature, contrary to what the authors claim. These may not be entirely to the authors' taste, but that hardly invalidates them or justifies their omission. In fact, merely by existing they render the authors' purely conceptual/metaphorical treatment inadequate.

Certainly, Bernstein's description of behaviors as "biodynamical structures that live and develop" presages the heksor concept; we stress that at several points in the manuscript (i.e., lines 152-154; 201-203; 795-797). Beyond that, we are uncertain as to what "precise, mechanistic formulations of the ideas...in the extant literature" the Referee is referring. We are not aware of any other previous ideas comparable to the heksor and negotiated equilibrium concepts. Referee #5's comments do not provide references, nor are similar ideas enunciated in Referee #4's very lengthy comments.

As discussed above, heksor and negotiated equilibrium are more than a "purely conceptual/metaphorical treatment." Heksor is a name for a structure that in a few simple examples has been described (albeit in a doubtless incomplete fashion (Figs. 1b & 1c)) and it is also a concept of the unique properties of that structure. Negotiated equilibrium is a name for a process that certainly exists but is yet to be understood, and it is also a concept of what that process is, a concept based on the hypothesized unique properties of the entity that we now call a heksor. It is in no way a metaphor.

The neuroscientific problem that is the focus of the manuscript – how multiple adaptive behaviors share a ubiquitously plastic CNS – has been recognized in the psychological literature for a long time, and more recently in the neural network literature (e.g., Grossberg's stability-plasticity dilemma). Despite Referee #4's assertion to the contrary, the stability-plasticity dilemma, also known as catastrophic interference, still remains an unresolved problem in general and for adaptive neural network (ANN) models specifically (as described and referenced in lines 140-149 of the manuscript). To the best of our knowledge, the heksor and negotiated equilibrium concepts constitute a new approach to this problem. They present and pursue this problem as a problem that is addressed by the numerous heksors involved, that is, by many active and independent agents each of which is seeking to maintain the key features of its own adaptive behavior. The distinctive feature of our new formulation of the problem is that it is a problem for something specific (i.e., for the heksor that produces an adaptive behavior), rather than simply a general problem. This leads to the hypothesis that the solution is equivalent to a Nash equilibrium. How this happens depends on the properties of heksors, which are yet to be substantively explored.

Nevertheless, even at this early date, the heksor and negotiated equilibrium concepts suggest new approaches to important issues in neuroscience (as described below). Furthermore, they provide obvious explanations for otherwise inexplicable experimental results ((e.g., lines 593-609; 610-630; 679-688; 706-738; 739-75; 755-772; 773-797). These ready explanations of puzzling results, did not follow from ideas previously available in the extant literature. In contrast, the heksor and negotiated equilibrium concepts do offer ready and obvious explanations. This suggests that these concepts are indeed new. How significant these concepts will turn out to be is as yet uncertain and will depend on their value for understanding and explaining experimental results.

Referee #5's review of the previous version of this manuscript dismissed its latter half as "rather vague" and devoted little attention to it; we are concerned that Referee #5 may have treated the latter half of this revision similarly. That half is now simplified, better focused, and extended in this revised version. We believe that it substantively addresses the questions now raised by the Referee in regard to the novelty and significance of the heksor and negotiated equilibrium concepts. Its overall message is that these concepts have not previously been expressed in a form sufficiently coherent and complete to have significant implications for understanding of CNS function, or to have important applications to practical issues, including treatment of CNS disorders. As the latter half of the revised manuscript illustrates, the heksor and negotiated equilibrium concepts render obvious some novel scientific implications and therapeutic applications that have not been generally recognized previously and have received little or no attention.

Thus, we believe that these concepts are in fact new. How important and useful they are remains to be seen.

Strong indication that the concepts are new is provided by the subsections devoted to three currently major topics in neuroscience: spontaneous neuronal activity; muscle synergies; and homeostatic plasticity. Recent studies have focused on the role of spontaneous activity (particularly during sleep) in the acquisition of new skills (lines 958-960). As discussed in lines 958-982, the heksor and negotiated equilibrium concepts suggest that the spontaneous activity associated with the acquisition of a new skill, and spontaneous activity in general, reflect at least in part the ongoing efforts of heksors to maintain the key features of their behaviors. Thus, the concepts indicate a potentially important new direction for research. The manuscript suggests how this direction might be pursued experimentally. The fact that these research directions have not been pursued to date implies that coherent ideas comparable to the heksor and negotiated equilibrium concepts are not extant in the existing literature.

Muscle synergies are receiving much attention at present. As discussed in lines 983-1003, the heksor and negotiated equilibrium concepts make obvious the possibility that these synergies result from interactions among heksors, that they are an aspect of the negotiated equilibrium that heksors create. This invites a new approach to their study; the section suggests how this could be pursued experimentally. That this new approach has not yet been undertaken implies that ideas similar to the heksor and negotiated equilibrium concepts are not extant in the existing literature.

Perhaps most obvious are the implications of the heksor and negotiated equilibrium concepts for the topic of homeostatic plasticity (lines 1004-1024). Each of the heksors that underlie existing behaviors has a clear interest in maintaining the status quo, the negotiated equilibrium that they have all achieved together. Thus, they can be expected to oppose major changes. Again, the new concepts make obvious a possible etiology for an important phenomenon that to our knowledge has not been considered. The discussion also suggests how the role of heksors in homeostatic plasticity might be studied.

We believe that these implications are new and potentially important; they suggest new and practical research directions that have not yet been pursued. That these implications follow from the heksor and negotiated equilibrium concepts suggests that the concepts themselves are new and potentially important. How productive and impactful these new research directions prove to be should play a major role in determining the ultimate significance of these new concepts. At present, their generation of these new directions implies that the concepts are new.

The authors' attitude to formal/mechanistic treatments is summarized not in the article itself, but in their telling response to the 4th referee. As someone originally trained in the physical sciences, I found much of this response to be bizarre since you could advance a similar set of criticisms against models and theories in physics, in which context they would (I think we can all agree) be absurd. A main point of the authors is that models of brain and behaviour seek merely to imitate - they are elaborate exercises in data fitting. The goal of the modeller, according to the authors, is to imitate behaviour. The authors' goal, in contrast, is to explain how the CNS actually generates behaviour. One could level the same complaint against Newton, Einstein and the architects of quantum theory. Their goal, you could say, is to imitate nature with their darned equations, whereas the goal of the true scientist is to explain how nature actually does the things it does. I am unmoved by such objections and I was equally unmoved by the authors' claims about the lowly imitative aims of the neural modeller and the loftier explanatory aims of empirical neuroscientists.

The authors other 'argument' concerning the continuous accrual of new facts rendering old models irrelevant is equally dubious. You could level the same criticism at Newton or Maxwell, for their equations may have imitated data available when they formulated their theories, but new findings have rendered them either worthless or only useful for certain technological applications. This is not how they are viewed in the physical sciences.

In the present context, models do not merely imitate behaviour, they also make predictions of new phenomena and of quantitative details. But not only that, they provide examples of the kind of mechanisms that solve particular kinds of problem. Take, for example, the problem of instantiating multiple mechanisms or entities within the same distributed substrate, a substrate that is continually changing as a result of experience. This is a well-known problem sometimes referred to as the stability-plasticity dilemma (a term due to Stephen Grossberg, see his scholarpedia entry: http://www.scholarpedia.org/article/Adaptive_resonance_theory). It has been addressed in detail in the literature on artificial neural networks and researchers have demonstrated what is needed for a solution in networks of interacting elements. These formal/mechanistic solutions are not merely

'imitating' behaviour - they are giving us important insights into the properties that are needed in order for multiple memories or other entities to be realized within the same shared network of elements.

I am sure that the 4th referee will provide a more comprehensive rebuttal of what the authors say in their response letter, so I won't go on. I will, however, briefly note that the authors are guilty of some double-standards when they appeal to formal, model-based concepts like Nash Equilibria and stable states in systems far from thermodynamic equilibrium. It seems to be a bit disingenuous to appeal to concepts that have strict formal treatments to justify loosely specified, verbally stated ideas, whilst at the same time denying that formal approaches have anything useful to offer.

Overall, I am ready to accept that the revised manuscript has something to offer certain groups (the positive comments of two of the referees can be taken as additional evidence of that) and so it has the potential to be a useful contribution. However, I think the authors' refusal to discuss model-based approaches or even to acknowledge that such approaches to the problems exist is unhelpful and misleading. My view is that this should somehow be sorted out - if that means that the authors must ultimately take less credit for the ideas they describe, then so be it.

Our response to Referee #4's comments was definitely not intended to be a rejection of the value of models in general. We agree that models have always been and will presumably continue to be a major component of effective scientific inquiry, including neuroscientific inquiry. Rather, our admittedly somewhat impolitic response was simply meant to be a rejection of Referee #4's position that modeling alone is all that is needed, that a model that yields the same results as the CNS constitutes a fully adequate understanding of CNS function; that consequently there is no need for messy often confusing studies of the actual biology. As Referee #5 points out to us above, our regard for models as an important aspect of scientific research in general and neuroscientific research in particular is indicated at several points in the manuscript (lines 140-149; 211-217; 452-463; 975-982; 1025-1042).

Finally, we would once again like to express our extreme gratitude for Referee #5's extensive, very substantive, and enormously valuable comments on the original and revised manuscripts. We appreciate the considerable time and careful thought that went into these comments, and we think that they have made our manuscript far better than it was originally. For this, we are very grateful. We hope sometime in the future to have the opportunity to continue discussions with Referee #5 in less formal circumstances.

Dear Dr Wolpaw,

Re: JP-TR-2022-283291XR1 "Heksor: The CNS Substrate of an Adaptive Behavior" by Jonathan R. Wolpaw and Adam Kamesar

I am pleased to tell you that your Topical Review article has been accepted for publication in The Journal of Physiology, subject to any modifications to the text that may be required by the Journal Office to conform to House rules.

NEW POLICY: In order to improve the transparency of its peer review process The Journal of Physiology publishes online as supporting information the peer review history of all articles accepted for publication. Readers will have access to decision letters, including all Editors' comments and referee reports, for each version of the manuscript and any author responses to peer review comments. Referees can decide whether or not they wish to be named on the peer review history document.

The last Word version of the paper submitted will be used by the Production Editors to prepare your proof. When this is ready you will receive an email containing a link to Wiley's Online Proofing System. The proof should be checked and corrected as quickly as possible.

All queries at proof stage should be sent to tjp@wiley.com

The accepted version of the manuscript will be published online, prior to copy editing in the Accepted Articles section.

Are you on Twitter? Once your paper is online, why not share your achievement with your followers. Please tag The Journal (@jphysiol) in any tweets and we will share your accepted paper with our 22,000+ followers!

Yours sincerely,

Ian D. Forsythe
Deputy Editor-in-Chief
The Journal of Physiology
<https://jp.msubmit.net>
<http://jp.physoc.org>
The Physiological Society
Hodgkin Huxley House
30 Farringdon Lane
London, EC1R 3AW
UK
<http://www.physoc.org>
<http://journals.physoc.org>

*** IMPORTANT NOTICE ABOUT OPEN ACCESS ***

To assist authors whose funding agencies mandate public access to published research findings sooner than 12 months after publication The Journal of Physiology allows authors to pay an open access (OA) fee to have their papers made freely available immediately on publication.

You will receive an email from Wiley with details on how to register or log-in to Wiley Authors Services where you will be able to place an OnlineOpen order.

You can check if your funder or institution has a Wiley Open Access Account here <https://authorservices.wiley.com/author-resources/Journal-Authors/licensing-and-open-access/open-access/author-compliance-tool.html>

Your article will be made Open Access upon publication, or as soon as payment is received.

If you wish to put your paper on an OA website such as PMC or UKPMC or your institutional repository within 12 months of publication you must pay the open access fee, which covers the cost of publication.

OnlineOpen articles are deposited in PubMed Central (PMC) and PMC mirror sites. Authors of OnlineOpen articles are permitted to post the final, published PDF of their article on a website, institutional repository, or other free public server, immediately on publication.

Note to NIH-funded authors: The Journal of Physiology is published on PMC 12 months after publication, NIH-funded authors DO NOT NEED to pay to publish and DO NOT NEED to post their accepted papers on PMC.

EDITOR COMMENTS

Reviewing Editor:

The careful additional work undertaken by the authors, in response to the most recent reviews, is acknowledged.

Senior Editor:

Thank you for a careful consideration of the referee's concerns; while these are not wholly resolved your final article is interesting and gives a novel perspective. I hope it generates as much interest from the Journal readers, and I look forward to seeing it in Press.

REFEREE COMMENTS

Referee #5:

The authors have revised their manuscript in line with the matters raised in the first round of reviews. Although I find myself in disagreement with the authors with respect to one or two matters, these disagreements are really matters of opinion or relate to the meanings of words rather than to substantive scientific content. For example, the term 'conceptual' seems to be a source of disagreement. In the context of the manuscript, I referred to all 'models' that are not formulated in mathematical or algorithmic terms as 'conceptual' and that such 'conceptual models' (if that is the right term for them) are verbal descriptions that cannot be implemented or simulated in order to determine their quantitative predictions and expected behaviour (for to do so requires mathematical or algorithmic formulation). As such, conceptual models can reasonably be described as 'vague'. I don't think that the authors really disagree with this given their responses to comments, but they do seem to take issue with my claim that their 'models' are 'conceptual', even though they are not mathematically formulated or 'simulate-able'. This must, therefore, be regarded as a quibble about the meanings of words. I do think that conceptual models can be valuable first steps towards formulation of formal models and as guides to experimentation, so they have an important role to play. Thus, I don't really think that the apparent disagreement is substantive. I would have liked to see an acknowledgment that some of the problems identified by the authors (such as the stability-plasticity dilemma) lie at the heart of some formal modelling efforts, but I see no reason to insist upon it (even if I could). In sum, I think that the revised manuscript represents a satisfactory revision given the issues raised.

2nd Confidential Review

03-Jun-2022